UWThPh 2025-9

# Vacua, Symmetries, and Higgsing of Chern–Simons Matter Theories

Fabio Marino and Marcus Sperling

*Fakultät für Physik, Universität Wien,*
*Boltzmanngasse 5, 1090 Wien, Austria,*
Email: `fabio.marino@univie.ac.at`,
`marcus.sperling@univie.ac.at`

## Abstract

Three-dimensional supersymmetric Chern–Simons Matter (CSM) theories typically preserve $\mathcal{N} = 3$ supersymmetry but can exhibit enhanced $\mathcal{N} = 4$ supersymmetry under special conditions. A detailed understanding of the moduli space of CSM theories, however, has remained elusive. This paper addresses this gap by systematically analysing the maximal branches of the moduli space of $\mathcal{N} = 3$ and $\mathcal{N} = 4$ CSM realised via Type IIB brane constructions.

Firstly, for $\mathcal{N} = 4$ theories with Chern–Simons levels equal 1, the $\mathrm{SL}(2, \mathbb{Z})$ dualisation algorithm is employed to construct dual Lagrangian 3d $\mathcal{N} = 4$ theories without CS terms. This allows the full moduli space to be determined using quiver algorithms that compute Higgs and Coulomb branch Hasse diagrams and associated RG flows.

Secondly, for $\mathcal{N} = 4$ theories with CS-levels greater 1, where $\mathrm{SL}(2, \mathbb{Z})$ dualisation does not yield CS-free Lagrangians, a new prescription is introduced to derive two magnetic quivers, $\mathsf{MQ}_A$ and $\mathsf{MQ}_B$, whose Coulomb branches capture the maximal A and B branches of the original $\mathcal{N} = 4$ CSM theory. Applying the decay and fission algorithm to $\mathsf{MQ}_{A/B}$ then enables the systematic analysis of A/B branch RG flows and their geometric structures.

Thirdly, for $\mathcal{N} = 3$ CSM theories, one magnetic quiver for each maximal (hyper-Kähler) branch is derived from the brane system. This provides an efficient and comprehensive characterisation of these previously scarcely studied features.

# 1 Introduction

Three-dimensional Chern–Simons Matter (CSM) theories with varying amounts of super-symmetry have been studied from multiple perspectives, ranging from field theory to string/M-theory. In general, the maximal supersymmetry a CSM theory can exhibit is $\mathcal{N} = 3$, due to constraints on supersymmetric Chern–Simons terms [1–3]. However, in special cases, supersymmetry enhancement can occur, leading to theories with $\mathcal{N} = 4$ [4, 5], $\mathcal{N} = 5$ [6], $\mathcal{N} = 6$ [7, 8], or even $\mathcal{N} = 8$ [9, 10]. These theories have led to significant developments in infrared dualities, mirror symmetry, and supersymmetry enhancement, while also providing key examples in holography and M2-brane dynamics.

As in any quantum field theory, a central question concerns the structure of the vacuum configurations and the nature of low-energy excitations around them. In supersymmetric theories, the set of vacua is typically not discrete but forms a continuous moduli space $\mathcal{M}$. This moduli space encodes both the vacua and the associated infrared (IR) dynamics. Understanding its structure is essential for characterising residual theories after Higgs mechanisms, as well as for tracking renormalisation group (RG) flows triggered by expectation values of local operators.

However, studying $\mathcal{M}$ is often challenging, as quantum corrections can lift classical vacua or significantly deform the moduli space. These difficulties are mitigated in the presence of extended supersymmetry. In three-dimensional theories with $\mathcal{N} = 3$ or $\mathcal{N} = 4$ supersymmetry, the moduli space $\mathcal{M}$ is typically a stratified singular space composed of hyper-Kähler branches. The singularities signal the appearance of additional massless states and may indicate the presence of strongly interacting IR fixed points. Moreover, the effective action at low energies is governed by the metric on $\mathcal{M}$, which, however, remains sensitive to quantum corrections.

To make progress, it is fruitful to reinterpret $\mathcal{M}$ as a complex algebraic variety. In this picture, the space of vacua is described by the spectrum of gauge-invariant, local chiral operators — that is, the chiral ring. The moduli space is then parametrised by the VEVs of a finite set of such operators subject to polynomial relations. This algebraic approach not only bypasses the difficulties associated with quantum-corrected metrics but also provides a concrete and computable framework for characterising the geometry of $\mathcal{M}$. The insights gained include: (i) (0-form) symmetries of the QFT are manifest as isometries of $\mathcal{M}$. (ii) The strata of $\mathcal{M}$ are indicative of the residual theories that can be reached via RG-flows triggered by VEVs of local operators. (iii) Matching the algebro-geometric description of $\mathcal{M}$ between allegedly dual theories provides strong evidence for the duality beyond the matching of moduli space dimensions and symmetries alone.

For 3d $\mathcal{N} = 4$ theories without CS-terms, $\mathcal{M}$ contains two distinguished components: the Higgs branch $\mathcal{H}$, parameterised by gauge-invariant chiral operators built from matter fields, and the Coulomb branch $\mathcal{C}$, parameterised by monopole operators. Both branches are symplectic singularities [11], and a feature of three-dimensional $\mathcal{N} = 4$ mirror symmetry [12] is the exchange of these two branches between mirror dual pairs.

While generic CSM theories preserve only $\mathcal{N} = 3$ supersymmetry, certain configurations with appropriately chosen matter content and Chern–Simons levels exhibit enhanced $\mathcal{N} = 4$ supersymmetry [4, 5, 7, 13–17]. In $\mathcal{N} = 4$ CSM theories, the presence of Chern–Simons couplings modifies the description of $\mathcal{M}$. The superconformal fixed point exhibits an $\mathrm{SU}(2)_A \times \mathrm{SU}(2)_B$ R-symmetry, and the moduli space still features two maximal quantum branches, referred to as the A and B branches. In contrast to non-CS theories, neither of these two branches is solely captured by mesons or monopole operators alone. Instead, due to the non-trivial CS couplings bare monopole operators are not gauge-invariant on their own and must be dressed with (charged) matter fields to ensure gauge invariance.

Understanding these quantum branches is crucial not just from a geometric standpoint, but also because they govern RG flows triggered by A/B-branch operators.

For $\mathcal{N} = 3$ CSM theories, the R-symmetry is the diagonal SU(2) of the $\mathcal{N} = 4$ R-symmetry, which implies that the maximal branches are all hyper-Kähler, but not separated in the usual manner. In general, the number of maximal branches is greater than two.

The first goal of this work is to provide a systematic description of the A and B branches of $\mathcal{N} = 4$ CSM theories realised on the world-volume of D3-branes suspended between NS5 and $(1, \kappa)$ 5-branes in Type IIB string theory [18–21]. The second goal is to initiate the analysis of the maximal branches in $\mathcal{N} = 3$ CSM theories realised in brane systems with NS5, D5, and various $(p, q)$ 5-branes.

In this setup, the brane perspective offers a unified, geometric interpretation of the moduli space: the maximal branches correspond to D3-brane segments along distinct 5-branes[1]. For $\mathcal{N} = 4$ systems, the A branch corresponds to the motion of D3-brane segments along NS5-branes, while the B branch arises from motion along $(1, \kappa)$ 5-branes. In contrast, for $\mathcal{N} = 3$ configurations, there are as many maximal branches as there are distinct types of 5-branes. Despite this uniform brane realisation, the cases (i) $\mathcal{N} = 4$ with $|\kappa| = 1$, (ii) $\mathcal{N} = 4$ with $|\kappa| > 1$, and (iii) $\mathcal{N} = 3$ require different techniques, and this work develops a systematic framework based on magnetic quivers to capture all regimes.

**Plan for CS-level $|\boldsymbol{\kappa}| = 1$.** For CS-level $|\kappa| = 1$, a 3d $\mathcal{N} = 4$ CSM theory can be dualised into a standard 3d $\mathcal{N} = 4$ $T_\sigma^\rho[\mathrm{SU}(N)]$ theory (linear case) or an affine A-type Kronheimer-Nakajima quiver (circular case) via the SL(2, $\mathbb{Z}$) transformation $\mathcal{T}^T = \mathcal{T}\mathcal{S}^{-1}\mathcal{T}$ [24, 25], where $\mathcal{S}$ and $\mathcal{T}$ are the SL(2, $\mathbb{Z}$) generators (see Appendix C.1 for conventions). Hence, the strategy employed in Section 2 (resp. 4) for linear (resp. circular) CSM quivers is the following, see also Figure 1a:

- Dualise the $\mathrm{CSM}_{|\kappa|=1}$ explicitly into a 3d $\mathcal{N} = 4$ theory $\mathsf{Q}$ via the *dualisation algorithm* [26–28]. Compared to dualising just the brane system, this has the advantage that the symmetry fugacities are tracked across the SL(2, $\mathbb{Z}$) duality. This facilitates matching of 3d $\mathcal{N} = 2$ superconformal indices [29–34] and operator spectroscopy.

- On $\mathsf{Q}$ one has full control over the Higgs and Coulomb branch. In addition to Hilbert series [35–38], the Higgs/Coulomb branch Hasse diagrams [39] can be worked out from the quiver description either via the decay and fission algorithm [40, 41] on the Coulomb branch or the Higgs branch subtraction [42] on the Higgs branch. Due to the dualisation algorithm, each Coulomb/Higgs branch RG-flow can be traced back into a A/B branch Higgsing for $\mathrm{CSM}_{|\kappa|=1}$.

- Of course, $\mathsf{Q}$ can be $\mathcal{S}$-dualised into $\mathsf{Q}^\vee$ and the map of Higgs/Coulomb branch to A/B branch of $\mathrm{CSM}_{|\kappa|=1}$ is swapped.

- Interestingly, one can complete the SL(2, $\mathbb{Z}$) duality map between $\mathrm{CSM}_{|\kappa|=1}$, $\mathsf{Q}$, and $\mathsf{Q}^\vee$ via another Chern–Simons Matter theory $\mathrm{CSM}'_{|\kappa|=1}$ that turns out to be the $\mathcal{T}$-dual of $\mathrm{CSM}_{|\kappa|=1}$. In the brane system, this is rather transparent as one simply swaps NS5 and $(1, 1)$ 5-branes via the $\mathcal{T}$ transformation.

**Plan for CS-level $|\boldsymbol{\kappa}| > 1$.** For a 3d $\mathcal{N} = 4$ theory $\mathrm{CSM}_{|\kappa|>1}$ with CS-levels larger than 1, there exists no SL(2, $\mathbb{Z}$) duality transformation into a pure 3d $\mathcal{N} = 4$ Lagrangian theory without CS-terms. Recalling that the A/B branches are affected by quantum relations, the well-suited technique are the *magnetic quivers* [43, 44] — i.e. symplectic singularities that

---

[1]Identifying branches of vacua from brane realisations builds upon the techniques developed in [22–24]

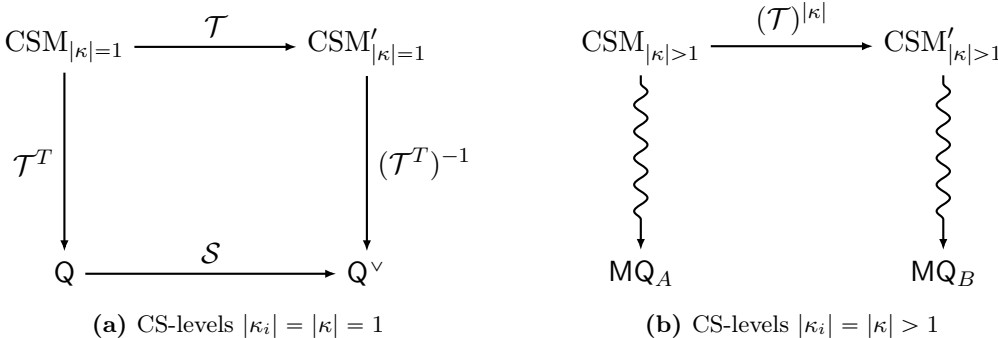

**(a)** CS-levels $|\kappa_i| = |\kappa| = 1$                 **(b)** CS-levels $|\kappa_i| = |\kappa| > 1$

**Figure 1:** Chern–Simons matter theories $\mathrm{CSM}_\kappa$ and their $\mathcal{T}^T$-duals or magnetic quivers. (a): for CS-levels $|\kappa_i| = 1$, there is a $\mathrm{SL}(2, \mathbb{Z})$ duality web at play. The $\mathcal{T}^T$ map yields standard $3d$ $\mathcal{N} = 4$ quiver theory $\mathsf{Q}$, on which another $\mathcal{S}$ yields the mirror dual $\mathsf{Q}^\vee$. The original $\mathrm{CSM}_{|\kappa|=1}$ theory is mapped via $\mathcal{T}$ into another $\mathrm{CSM}'_{|\kappa|=1}$ theory whose A and B branch are swapped with respect to those of $\mathrm{CSM}_{|\kappa|=1}$. Applying another $(\mathcal{T}^T)^{-1}$ transformation completes a closed $\mathrm{SL}(2, \mathbb{Z})$ web into $\mathsf{Q}^\vee$. (b): for CS-levels $|\kappa_i| > 1$, there is no $\mathrm{SL}(2, \mathbb{Z})$ transformation into Lagrangian non-CS theories. Instead, one observes that a $(\mathcal{T})^{|\kappa|}$ transformation maps $\mathrm{CSM}_{|\kappa|>1}$ into $\mathrm{CSM}'_{|\kappa|>1}$; again, both theories are dual and have swapped A/B branches. The proposal is to map the A/B branches of both $\mathrm{CSM}_{|\kappa|>1}$ and $\mathrm{CSM}'_{|\kappa|>1}$ into a magnetic quiver, called $\mathsf{MQ}_A$ and $\mathsf{MQ}_B$, respectively. The two magnetic quivers are sufficient to analyse the moduli space and the RG-flows of $\mathrm{CSM}_{|\kappa|>1}$.

are realised by a 3d $\mathcal{N} = 4$ Coulomb branch of an auxiliary quiver theory. Concretely, in Section 3 (and also Section 4) a simple prescription is proposed that derives two magnetic quivers $\mathsf{MQ}_{A/B}$, one for the A and one for the B branch; see Figure 1b.

- Building on the brane realisation of $\mathrm{CSM}_{|\kappa|>1}$, the A branch magnetic quiver $\mathsf{MQ}_A$ is derived via an auxiliary brane configuration that isolates the A branch moduli of the CSM system.

- Recall the manifestation of the two branches in the CSM brane system, one swaps the NS5 and $(1, \kappa)$ 5-branes via a $(\mathcal{T})^{|\kappa|}$ $\mathrm{SL}(2, \mathbb{Z})$ transformation. The resulting $\mathrm{CSM}'_{|\kappa|>1}$ can now be analysed as before: deriving an auxiliary brane system that isolates its A branch moduli allows to derive a B branch magnetic quiver $\mathsf{MQ}_B$ for $\mathrm{CSM}_{|\kappa|>1}$.

- Having established $\mathsf{MQ}_{A/B}$, one can validate the proposal via matching Coulomb branch Hilbert series [38] of $\mathsf{MQ}_{A/B}$ with the A/B branch limits of the superconformal index of $\mathrm{CSM}_{|\kappa|>1}$, à la [45]. Moreover, by using the decay and fission algorithm [40, 41] on $\mathsf{MQ}_{A/B}$ one then predicts the A/B branch Hasse diagrams. In particular, this gives access to the A/B branch RG-flows of $\mathrm{CSM}_{|\kappa|>1}$.

As an appetiser of the results, Table 1 showcases the $\mathsf{MQ}_{A/B}$ from frequently studied linear CSM quiver theories. For the abelian cases (Table 1a) the magnetic quivers elegantly reproduce and extend previous piecewise results, see for example [14, 15, 46, 47]. For non-abelian CSM quivers (Table 1b), most known data on A/B branches were conjectured from limits of the index [47, 48] (or explicit Hilbert series [49]). The techniques presented here not only allow for a much easier derivation of the branch data via $\mathsf{MQ}_{A/B}$ (for a much more vast class of theories), but also provide more insights on symmetries and RG-flows.

**Extension to $\mathcal{N} = 3$ CSM.** Adding extra $(p, q)$ 5-branes to the brane system breaks $\mathcal{N} = 4$ down to $\mathcal{N} = 3$. Nonetheless, the maximal branches of the moduli space of such $\mathcal{N} = 3$ CSM theories are all symplectic singularities as well, see for instance [15]. Fortunately, from the brane system perspective, there is no essential difference between $\mathcal{N} = 4$ and $\mathcal{N} = 3$ configurations: maximal branches correspond to D3-segments moving

between distinct $(p, q)$ 5-branes. Therefore, the proposed magnetic quiver prescription is the suitable framework to capture the geometry of the maximal branches in $\mathcal{N} = 3$ CSM theories as well, as demonstrated in Section 5.

To extract the magnetic quiver corresponding to each maximal branch, one performs an $\mathrm{SL}(2, \mathbb{Z})$ transformation on the original brane configuration such that the $(p, q)$ 5-branes of interest are exchanged with NS5-branes. This procedure enables a reinterpretation of the moduli associated with D3-brane segments stretched between $(p, q)$ 5-branes in terms of D3-branes moving between NS5-branes in the dual configuration. The magnetic quiver can then be read off using well-established rules. This proposal is validated against known maximal branches for Abelian $\mathcal{N} = 3$ CSM theories. For non-Abelian $\mathcal{N} = 3$ CSM theories, the proposal offers an efficient and fast derivation of new predictions for the maximal branches. This includes in particular, the non-Lagrangian $\mathcal{N} = 3$ Chern–Simons theories realised by $(p, q)$ 5-branes with $p > 1$ and $q \neq 0$.

**Outline.**   Section 2 is focused on linear CSM quiver theories with CS-levels $|\kappa| = 1$; which benefits for an explicit $\mathrm{SL}(2, \mathbb{Z})$ dualisation and subsequent analysis of standard 3d $\mathcal{N} = 4$ linear quiver theories. Thereafter, Section 3 is devoted to linear CSM quiver theories with $|\kappa| > 1$. Therein, the magnetic quiver derivation is explained and cross-checked. The setup is then extended to circular CSM quivers in Section 4. In Section 5, the maximal branches of $\mathcal{N} = 3$ CSM theories are subjected to the magnetic quiver approach. Finally, Section 6 provides conclusions and an outlook. Several appendices complement the main body with relevant background material and computational details.

**Notation.**   Several methods in this paper have a graphical representation.

- Firstly, Type IIB brane configurations are drawn by using coloured lines of different angles: D3 branes ( $-$ ), NS5 branes ( $|$ ), $(p, q)$ 5-branes ( $\diagup$ ), and D5 branes ( $\otimes$ ), see Appendix A.2.

- Secondly, the supersymmetric QFTs are represented by quiver diagrams: round nodes ( $\bigcirc$ ) encode vector multiplets with or without CS-level; edges encode a pair of chiral multiplets in conjugate representations; loops/arcs on round nodes represent the $\mathcal{N} = 2$ adjoint chiral in the $\mathcal{N} = 4$ vector multiplet; square nodes ( $\square$ ) encode flavour symmetries.

- Thirdly, Hasse diagrams [39] or the phase diagram of the RG-flows, are graphs composed of (i) vertices: labelling residual theories, (ii) edges: encoding the geometric type of the transition. Most relevant are: Kleinian/Du-Val singularity $A_k \cong \mathbb{C}^2/\mathbb{Z}_{k+1}$, and the closure of the minimal nilpotent orbit of $\mathfrak{sl}(n + 1)$, denoted by $a_n = \overline{\mathcal{O}}_{\min}^{\mathfrak{sl}(n+1)}$.

**Table 1 (a): Exemplary Abelian Chern-Simons Matter theories**

| CSM theory | MQ$_A$ | Hasse diag. A branch | MQ$_B$ | Hasse diag. B branch |
|---|---|---|---|---|
| $+\kappa$ — $-\kappa$, nodes $1,1$ | $\square\,|\kappa|$ — $\bigcirc\,1$ | $A_{|\kappa|-1}$ | trivial | — |
| $+\kappa,-\kappa,+\kappa$, nodes $1,1,1$ | $\square\,|\kappa|+1$ — $\bigcirc\,1$ | $A_{|\kappa|}$ | $\square\,|\kappa|+1$ — $\bigcirc\,1$ | $A_{|\kappa|}$ |
| $+\kappa,-\kappa,+\kappa,-\kappa$, nodes $1,1,1,1$ | $\square\,|\kappa|$, $\square\,|\kappa|$ — $\bigcirc\,1$, $\bigcirc\,1$ | $|\kappa|>1:$ $A_{|\kappa|-1}$, $A_{|\kappa|}$ branches; $|\kappa|=1:$ $a_2$ | $\square\,|\kappa|+2$ — $\bigcirc\,1$ | $A_{|\kappa|+1}$ |

(a) Exemplary Abelian Chern-Simons Matter theories. For $|\kappa|=1$, the theories are dual to 3d $\mathcal{N}=4$ theories, cf. [14, 46]. All MQ$_{A/B}$ are consistent with the Hilbert series limits of the index derived in [47]. The 3-node (resp. 4-node) quivers were denoted Model III (resp. IV) in [14], where their A/B branches have been studied. The 3-node CSM quiver is known to have isomorphic A and B branch, see e.g. [14, 15]: indeed MQ$_A$ = MQ$_B$. This is a sign of the $\mathcal{T}^{|\kappa|}$ duality that exchanges NS5 and $(1,\kappa)$ 5-branes.

**Table 1 (b): Exemplary non-Abelian Chern-Simons Matter theories**

| CSM theory | MQ$_A$ | MQ$_B$ | Parameter range |
|---|---|---|---|
| $+\kappa$ — $-\kappa$, $N_1,N_2$ | $\square\,|\kappa|-|N_1-N_2|$ — $\bigcirc\,\min(N_1,N_2)$ | trivial | $|N_1-N_2|\leq|\kappa|$, good-ness of MQ$_A$ |
| $+\kappa,-\kappa,+\kappa$, $N_1,N_2,N_3$ | $\square\,|\kappa|-|N_1-N_2|+N_3$ — $\bigcirc\,\min(N_1,N_2)$ | $\square\,|\kappa|-|N_2-N_3|+N_1$ — $\bigcirc\,\min(N_2,N_3)$ | $|N_1-N_2|\leq|\kappa|,\ N_3\leq|\kappa|$, $|N_2-N_3|\leq|\kappa|,\ N_1\leq|\kappa|$, good-ness of MQ$_{A/B}$ |
| $+\kappa,-\kappa,+\kappa,-\kappa$, $N_1,N_2,N_3,N_4$ | $\square\,|\kappa|-|N_1-N_2|+\max(0,N_3-N_4)$, $\square\,|\kappa|-|N_3-N_4|+\max(0,N_2-N_1)$ — $\bigcirc\,\min(N_1,N_2)$, $\bigcirc\,\min(N_3,N_4)$ | $\square\,|\kappa|-|N_2-N_3|+N_3+N_4$, $\square\,|\kappa|-|N_2-N_3|+N_3+N_4$ — $\bigcirc\,\min(N_2,N_3)$, $\bigcirc\,\min(N_2,N_3)$ | $|N_1-N_2|\leq|\kappa|,\ |N_3-N_4|\leq|\kappa|$, $N_1\leq|\kappa|,\ N_4\leq|\kappa|$, good-ness of MQ$_{A/B}$ |

(b) Exemplary non-Abelian Chern-Simons Matter theories. The 2-node quiver belongs to the 3d $\mathcal{N}=4$ CSM theories introduced in [4]. It was argued in [16, 50] that this CSM theory is good if $|\kappa|-N_1-N_2>-1$, which agrees with the condition for a good MQ$_A$. Moreover, MQ$_A$ is consistent with the Hilbert series limit of the index in [47]. The 3-node quiver with $N_1=N_2=N_3$ is known to have isomorphic A and B branch, which is manifest in MQ$_{A/B}$. Again, this follows from $\mathcal{T}^{|\kappa|}$, which swaps NS5 and $(1,\kappa)$ 5-branes. The 4-node quiver is the 3d $\mathcal{N}=4$ CSM theory constructed in [5]. The MQ$_{A/B}$ for the 4-node quiver generalise and refine conjectures based on HS-limits [47].

**Table 1:** Overview of magnetic quivers MQ$_{A/B}$ applied to well-studied CSM theories. Results are displayed for a suitable parameter range; based on brane systems in a suitable GK-duality frame (see Section 2 and Appendix D). The criteria for *good* MQ$_{A/B}$ imply further constraints on the CSM theory to be *good*.

## 2 Linear $\mathcal{N}$=4 Chern–Simons Matter theories, CS-levels =1

In this section examples of linear $\mathcal{N} = 4$ Chern–Simons Matter theories with CS-levels $|\kappa_i| = |\kappa| = 1$ (or $\text{CSM}_{|\kappa|=1}$ for short) that are realised via D3, NS5, and $(1, \pm 1)$ 5-brane configurations in Type IIB superstring theory are considered (see Appendix A.2 for a brief summary). This class of models is special because one can $\text{SL}(2,\mathbb{Z})$ dualise them into non-CS theories. This is particularly manifest in the D3, NS5, and $(1, \pm 1)$ 5-brane system, which is mapped into a standard D3-NS5-D5 brane system via the $\mathcal{T}^T = -\mathcal{T}\mathcal{S}\mathcal{T}$ transformation [24, 25], see Figure 2. Field-theoretically, one may use the dualisation algorithm [28], as reviewed in Appendix C. The dualisation algorithm allows one not only to obtain the $\mathcal{T}^T$-dual theory Q, which is a standard $T_\sigma^\rho[\text{SU}(N)]$ theory [24], but also to map the fugacities (and hence the symmetries) and to assign the correct charges to the fields. On the 3d $\mathcal{N} = 4$ non-CS theory Q, one can act with the $\mathcal{S}$ generator to deduce the 3d mirror dual theory $Q^\vee$ (as well as the brane system), see Figure 2. One may then ask, what CSM theory is associated to this mirror via the analogue of $\mathcal{T}^T$?

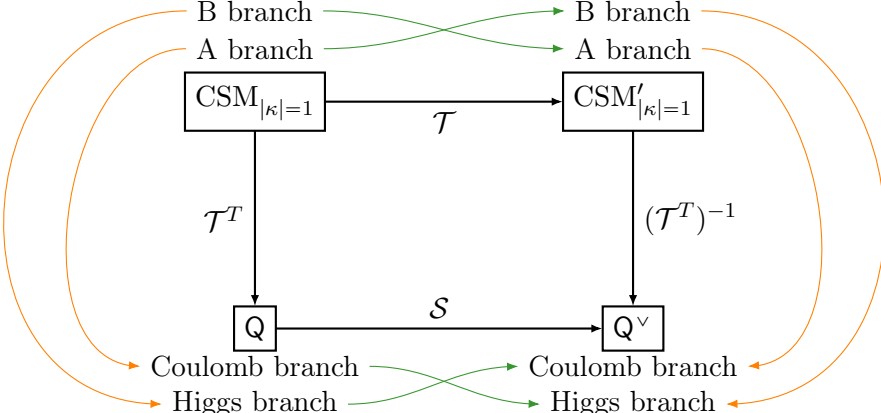

**Figure 2:** The $\text{SL}(2,\mathbb{Z})$ duality web employed to study $\text{CSM}_{|\kappa|=1}$ theories. All the computations carried out in this section involve exclusively the leftmost part of the diagram, namely the starting theory $\text{CSM}_{|\kappa|=1}$ and its $\mathcal{T}^T$-dual Q, as the rightmost sector of the web is equivalent. To match the index of these theories one has to properly transform of the $\text{U}(1)_{\text{axial}}$ fugacity $t$ (see eq. (A.1)).

The $\text{SL}(2,\mathbb{Z})$ web in Figure 2 can be written as[2]

$$\mathcal{S}\mathcal{T}^T = (\mathcal{T}^T)^{-1}\mathcal{T}. \tag{2.1}$$

One can use the $(p,q)$-branes notation and write $\text{NS5} = (\pm 1, 0)$ and $\text{D5} = (0, \pm 1)$. Then, using the fact [18] that

$$(p,q) \xrightarrow{\mathcal{S}} (q, -p) \qquad :\Leftrightarrow \quad (p,q) = \mathcal{S}(q, -p), \tag{2.2a}$$

$$(p,q) \xrightarrow{\mathcal{T}} (p, q-p) \qquad :\Leftrightarrow \quad (p,q) = \mathcal{T}(p, q-p), \tag{2.2b}$$

$$(p,q) \xrightarrow{\mathcal{T}^T} (p-q, q) \quad :\Leftrightarrow \quad (p,q) = \mathcal{T}^T(p-q, q), \tag{2.2c}$$

the $\text{SL}(2,\mathbb{Z})$ web in Figure 2 taking $\kappa = +1$ can be written in terms of branes as follows:

$$
\begin{array}{ccc}
(1,0) & \xrightarrow{\ \mathcal{T}\ } & (1,-1) \\
(1,1) & & (1,0) \\
\mathcal{T}^T \downarrow & & \downarrow (\mathcal{T}^T)^{-1} \\
(1,0) & \xrightarrow{\ \mathcal{S}\ } & (0,-1) \\
(0,1) & & (1,0)
\end{array} \tag{2.3}
$$

---

[2]Using Appendix C.1, one verifies $(\mathcal{T}^T)^{-1}\mathcal{T}(\mathcal{T}^T)^{-1} = (-\mathcal{S}\mathcal{T}\mathcal{S})\mathcal{T}(-\mathcal{S}\mathcal{T}\mathcal{S}) = +(\mathcal{S}\mathcal{T})^3\mathcal{S} = \mathcal{S}$.

Therefore, the appearing $\mathrm{SL}(2, \mathbb{Z})$ transformations can be summarised as follows:

- To map the CSM brane system to a standard D3-D5-NS5 configuration for the 3d $\mathcal{N} = 4$ theory Q: keep NS5 and map $(1, 1)$ into D5. This is achieved by $\mathcal{T}^T$.

- To map the brane configuration for Q to that of its 3d $\mathcal{N} = 4$ mirror dual theory $Q^\vee$: utilise the standard $\mathcal{S}$ transformation [22], i.e. swap D5 and NS5.

- To go from CSM to CSM′, use same logic as with Q to $Q^\vee$: swap NS5 and $(1, \pm 1)$ branes. This is realised by $\mathcal{T}$.

- Lastly, close the $\mathrm{SL}(2, \mathbb{Z})$ diagram, via a map from CSM′ to $Q^\vee$; this yields $(\mathcal{T}^T)^{-1}$.

At this point, it might not be clear why the web of theories in Figure 2 is helpful. One useful (though fairly standard for $|\kappa| = 1$) relation that can be extracted is

$$\text{A-branch}(\text{CSM}_{|\kappa|=1}) \cong \mathcal{C}(\mathsf{Q}) , \qquad \text{B-branch}(\text{CSM}_{|\kappa|=1}) \cong \mathcal{C}(\mathsf{Q}^\vee) , \qquad (2.4)$$

with an analogous statement for Higgs branches. However, for $|\kappa| > 1$, there *does not exist* an $\mathrm{SL}(2, \mathbb{Z})$ transformation to a D3-D5-NS5 system. Nevertheless, one can define a prescription to an auxiliary brane system that yields a magnetic quiver for the A-branch, see Section 3. Then, even for $|\kappa| > 1$, one *can* swap NS5 and $(1, \kappa)$ 5-branes, analogous to Figure 2. On this system, one applies the *same prescription* for an auxiliary brane system and defines a second magnetic quiver that now captures the B-branch of the CSM theory. This motivates the web in Figure 2 for $|\kappa| = 1$; it generalises to the $|\kappa| > 1$ case.

To start this section, an Abelian theory with alternating CS-levels is considered as warm-up example in Section 2.1. The dualisation process, which is detailed in Appendix C, is directly applied to derive the dual theories. A number of aspects are then considered: evaluation of indices and matching gauge-invariant operators across the duality. Thereafter, the Higgsing pattern is studied: for the non-CS dual theories, the Higgs/Coulomb branch RG-flows are well-known. In contrast, the CS theory has highly non-trivial RG-flows. By dualising back each theory in the Hasse diagram of the non-CS dual, namely by applying the $(\mathcal{T}^T)^{-1}$ transformation, one can produce all the theories in the Hasse diagram of the electric CS model for both branches. In this work, the *A branch* of the CS-theory corresponds to the Coulomb branch of its non-CS $\mathcal{T}^T$-dual, and the *B branch* of the CS-theory corresponds to the Higgs branch of its non-CS $\mathcal{T}^T$-dual. Moreover, via the operator spectroscopy, one can also recognise the operators taking a VEV in the CSM theory.

This strategy is then extended to an non-Abelian example in Section 2.2 and subsequently generalised in Section 2.3. As a remark, the focus lies on good theories, in the sense of [24]. A intuitive way to determine such a property for a $\text{CSM}_{|\kappa|=1}$ theory is as follows: a $\text{CSM}_{|\kappa|=1}$ theory is *good* if its $\mathcal{T}^T$-dual is *good*.

## 2.1 Example: Abelian CSM theory

To begin with, consider the Abelian theory shown in Figure 3a, wherein also the brane realisation is provided. Upon Giveon–Kutasov (GK) duality [51] one can derive the equivalent theory shown Figure 3b (see Appendix D for more details). In addition, using the dualisation algorithm one can derive a $\mathcal{T}^T$-dual theory, as shown in Figure 3c, which is a standard $T^\rho_\sigma[\mathrm{SU}(N)]$ theory (see Appendix C.6 for the explicit dualisation of this example).

**Index.** First, consider the superconformal index analysis. The global symmetry algebra of the theories in Figure 3 is

$$[\mathfrak{u}(1)_{q_1} \oplus \mathfrak{u}(1)_{q_2}]_A \oplus [\mathfrak{su}(2)_u \oplus \mathfrak{su}(2)_v \oplus \mathfrak{u}(1)_s]_B , \qquad (2.5)$$

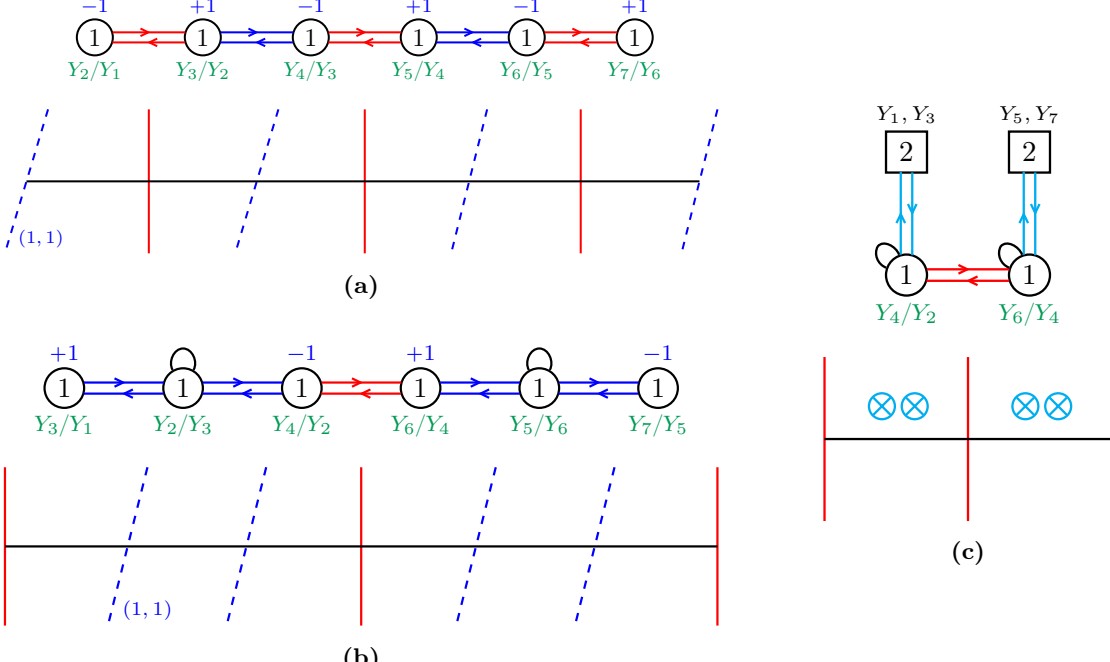

**Figure 3:** (a): The electric theory, namely an Abelian $\mathrm{CSM}_{|\kappa|=1}$ quiver with 6 gauge nodes. For the quiver diagram the 3d $\mathcal{N}=2$ language is employed. In particular, hypermultiplets (red) and twisted hypermultiplets (blue) are represented using pairs of arrows (namely chiral multiplets). The CS-levels are denoted in blue above the corresponding gauge nodes, while the topological fugacities are denoted in green. Below the quiver, the corresponding brane system is shown. Brane configurations are drawn by using coloured lines of different angles, see Appendix A.2. (b): The result of the GK duality on the starting electric theory above. The arcs on the gauge nodes represent the $\mathcal{N}=2$ massless adjoint chiral multiplets (they are present only on gauge nodes without CS-level since the CS term induces a mass for the adjoint chiral multiplet, see Appendix A). Below the quiver, the corresponding colour coded brane system is shown. (c): The $\mathcal{T}^T$-dual theory, where the flavour Cartans have been explicitly written in black to show the parameters map across the duality. Below the quiver, the corresponding colour coded brane system is shown.

where A/B denotes the branches of the CSM theories, which match with the Coulomb/Higgs branches of their $\mathcal{T}^T$-dual. The refined 3d $\mathcal{N}=2$ superconformal index[3] of the theories in Figure 3 is perturbatively evaluated to

$$
\begin{aligned}
\mathcal{I} = 1 &+ x(t^2([2]_u + [2]_v + 1) + 2t^{-2}) \\
&+ x^{3/2}(t^3([1]_u[1]_v(s + s^{-1})) + t^{-3}(q_1 + q_1^{-1} + q_2 + q_2^{-1})) \\
&+ x^2(t^4([2]_u[2]_v + [4]_u + [2]_u + [4]_v + [2]_v + 1) + t^{-4}(q_1q_2 + q_1^{-1}q_2^{-1} + 3) - 4) \\
&+ O(x^{5/2})
\end{aligned}
\tag{2.6}
$$

wherein the appearing irreducible representations (irreps) of (2.5) have been denoted by Dynkin labels for non-Abelian factors and fugacities for Abelian parts. The map between the above variables and the fugacities in the quiver (Figure 3a) description is given by

$$
\frac{Y_1}{Y_3} = u^2\,, \qquad \frac{Y_2}{Y_4} = q_1\,, \qquad \frac{Y_3}{Y_5} = \frac{s}{u\,v}\,, \qquad \frac{Y_4}{Y_6} = q_2\,, \qquad \frac{Y_5}{Y_7} = v^2\,.
\tag{2.7}
$$

Analogously, one can build the fugacity map for the GK dual (Figure 3b) of the electric theory and for the $\mathcal{T}^T$-dual (Figure 3c). The index expansion written in terms of flavour

---

[3]For the 3d $\mathcal{N}=2$ supersymmetric index conventions, the reader is referred to [52]. In particular, using $x$ and $t$ as the fugacities for the R-symmetry and the axial symmetry respectively, the $\mathcal{N}=4$ hypermultiplet has charges $x^{1/2}t^{+1}$, while the $\mathcal{N}=4$ twisted hypermultiplet has $x^{1/2}t^{-1}$.

characters allows to read the following information [53, 54]. Order $O(x)$ contains the flavour currents: The A/Coulomb branch has an $\mathfrak{u}(1)_{q_1} \oplus \mathfrak{u}(1)_{q_2}$ isometry algebra, while the B/Higgs branch has an $\mathfrak{su}(2)_u \oplus \mathfrak{su}(2)_v \oplus \mathfrak{u}(1)_s$ isometry algebra. This is reflected in (2.5). Next, order $O(x^{3/2})$ contains the first gauge invariant operators that are non-trivially charged under the Abelian symmetry factors, see Table 2. Very insightful is order $O(x^2)$, which contains positive terms reflecting marginal operators and negative terms which are the symmetry currents. More specifically, one has the following:

- $O(t^4)$ terms contain pure B/Higgs branch operators that transform in

$$\mathrm{Sym}^2([2]_u + [2]_v + 1) = [2]_u[2]_v + [4]_u + [2]_u + [4]_v + [2]_v + 3. \qquad (2.8)$$

  Note, however, that not all three singlets coming from $\mathrm{Sym}^2([2])$ are independent. In fact, an F-term analysis of the $\mathcal{T}^T$-dual shows that $\mathrm{Tr}\left[\mu^2_{\mathrm{SU}(2)_u}\right] = \mathrm{Tr}\left[\mu^2_{\mathrm{SU}(2)_v}\right] = (Q_{1,2}\tilde{Q}_{1,2})^2$, where $\mu_{\mathrm{SU}(2)_u} = P_1\tilde{P}_1$ and $\mu_{\mathrm{SU}(2)_v} = P_2\tilde{P}_2$ are the moment maps for the non-abelian symmetries (using the field names of Table 2). Consequently, the singlets are identified by two relations, and only one singlet survives as shown in (2.6).

- $O(t^{-4})$ terms contain pure A/Coulomb branch operators for which there are $\mathrm{Sym}^2(2) = 3$. In addition, there are two non-trivially charged new gauge-invariant operators, which can be analogously identified as the operators in Table 2.

- $O(t^0)$ positive terms are mixed branch operators. In fact, based on the honest dual theory, one expects such operators to transform as

$$1 \cdot [2]_u + 1 \cdot [2]_v = t^2(1) \cdot t^{-2}([2]_u + [2]_v) \qquad (2.9)$$

  i.e. under the non-Abelian factors of the A/Coulomb branch and under one Abelian factor of the B/Higgs branch. This can be seen from the Hasse diagrams in Figure 4. On each branch, there are non-Abelian Higgsings, after which on the other branch only an Abelian Higgsing is left. Therefore, using (2.9) the whole $x^2$ coefficient of (2.6) can be rewritten as

$$t^4([2]_u[2]_v + [4]_u + [2]_u + [4]_v + [2]_v + 1) + t^{-4}(q_1q_2 + q_1^{-1}q_2^{-1} + 3)$$
$$+ \,t^2(1) \cdot t^{-2}(\cancel{[2]_u + [2]_v}) - (1 + \cancel{[2]_v + [2]_u} + 2 + 1) \qquad (2.10)$$

- $O(t^0)$ negative terms are the $(1 + [2]_v + [2]_u)_A + (2)_B$ flavour currents of (2.5) plus potential extra SUSY currents. By comparison, in (2.10) one extra SUSY current appears — the axial U(1) — signalling the supersymmetry enhancement to $\mathcal{N} = 4$.

The index expansion (2.6) also suggests that the global form of the symmetry group is

$$\left[\frac{\mathrm{SU}(2)_u \times \mathrm{SU}(2)_v \times \mathrm{U}(1)_s}{\mathbb{Z}_2 \times \mathbb{Z}_2}\right]_A \times [\mathrm{U}(1)_{q_1} \times \mathrm{U}(1)_{q_2}]_B, \qquad (2.11)$$

where the $\mathrm{SU}(2)_u \times \mathrm{SU}(2)_v$ centres $\mathbb{Z}_2 \times \mathbb{Z}_2$ act on the $\mathrm{U}(1)_s$ with charges $(1,1)$.

**Table 2.** Operator spectroscopy

| $O(x)$ | $O(t)$ | Character | Operators (Fig. 3a) | Operators (GK dual, Fig. 3b) | Operators ($\mathcal{T}^T$-dual, Fig. 3c) |
|---|---|---|---|---|---|
| $x^1$ | $t^{-2}$ | $(q_1)^0 \equiv 1$ | Tr$\left[Q_{2,3}\tilde{Q}_{2,3}\right]$ | Tr$\left[Q_{1,2}\tilde{Q}_{1,2}\right] \overset{\text{F-terms}}{=}$ Tr$\left[Q_{2,3}\tilde{Q}_{2,3}\right]$ | Tr$[A_1]$ |
| | | $(q_2)^0 \equiv 1$ | Tr$\left[Q_{4,5}\tilde{Q}_{4,5}\right]$ | Tr$\left[Q_{4,5}\tilde{Q}_{4,5}\right] \overset{\text{F-terms}}{=}$ Tr$\left[Q_{5,6}\tilde{Q}_{5,6}\right]$ | Tr$[A_2]$ |
| | $t^2$ | $[2]_u$ | $\mathfrak{M}^{(++0000)}\tilde{Q}_{1,2}$, $\mathfrak{M}^{(--0000)}Q_{1,2}$, Tr$\left[Q_{1,2}\tilde{Q}_{1,2}\right]$ | $\mathfrak{M}^{(0+0000)}$, $\mathfrak{M}^{(0-0000)}$, Tr$[A_2]$ | Tr$\left[P_1\tilde{P}_1\right]$ |
| | | $[2]_v$ | $\mathfrak{M}^{(0000++)}\tilde{Q}_{5,6}$, $\mathfrak{M}^{(0000--)}Q_{5,6}$, Tr$\left[Q_{5,6}\tilde{Q}_{5,6}\right]$ | $\mathfrak{M}^{(0000+0)}$, $\mathfrak{M}^{(0000-0)}$, Tr$[A_5]$ | Tr$\left[P_2\tilde{P}_2\right]$ |
| | | $(s)^0 \equiv 1$ | Tr$\left[Q_{3,4}\tilde{Q}_{3,4}\right]$ | Tr$\left[Q_{3,4}\tilde{Q}_{3,4}\right]$ | Tr$\left[Q_{1,2}\tilde{Q}_{1,2}\right]$ |
| $x^{\frac{3}{2}}$ | $t^{-3}$ | $q_1 + \frac{1}{q_1}$ | $\mathfrak{M}^{(0++000)}\tilde{Q}_{2,3}$, $\mathfrak{M}^{(0--000)}Q_{2,3}$ | $\mathfrak{M}^{(+++000)}\tilde{Q}_{1,2}\tilde{Q}_{2,3}$, $\mathfrak{M}^{(---000)}Q_{1,2}Q_{2,3}$ | $\mathfrak{M}^{(+0)}$, $\mathfrak{M}^{(-0)}$ |
| | | $q_2 + \frac{1}{q_2}$ | $\mathfrak{M}^{(000++0)}\tilde{Q}_{4,5}$, $\mathfrak{M}^{(000--0)}Q_{4,5}$ | $\mathfrak{M}^{(000+++)}\tilde{Q}_{4,5}\tilde{Q}_{5,6}$, $\mathfrak{M}^{(000---)}Q_{4,5}Q_{5,6}$ | $\mathfrak{M}^{(0+)}$, $\mathfrak{M}^{(0-)}$ |
| | $t^3$ | $([1]_u \cdot [1]_v) \cdot (s + \frac{1}{s})$ | $\mathfrak{M}^{(00++00)}\tilde{Q}_{3,4}$, $\mathfrak{M}^{(00--00)}Q_{3,4}$ | $\mathfrak{M}^{(00++00)}\tilde{Q}_{3,4}$, $\mathfrak{M}^{(00--00)}Q_{3,4}$ | Tr$\left[P_1Q_{1,2}\tilde{P}_2\right]$ |
| | | | $\mathfrak{M}^{(+++00)}\tilde{Q}_{1,2}\tilde{Q}_{3,4}$, $\mathfrak{M}^{(---00)}Q_{1,2}Q_{3,4}$ | $\mathfrak{M}^{(0+++00)}\tilde{Q}_{3,4}$, $\mathfrak{M}^{(0---00)}Q_{3,4}$ | Tr$\left[\tilde{P}_1\tilde{Q}_{1,2}P_2\right]$ |
| | | | $\mathfrak{M}^{(00+++)}\tilde{Q}_{3,4}\tilde{Q}_{5,6}$, $\mathfrak{M}^{(00---)}Q_{3,4}Q_{5,6}$ | $\mathfrak{M}^{(00+++0)}\tilde{Q}_{3,4}$, $\mathfrak{M}^{(00---0)}Q_{3,4}$ | |
| | | | $\mathfrak{M}^{(+++++)}\tilde{Q}_{1,2}\tilde{Q}_{3,4}\tilde{Q}_{5,6}$, $\mathfrak{M}^{(-----)}Q_{1,2}Q_{3,4}Q_{5,6}$ | $\mathfrak{M}^{(0+++0)}\tilde{Q}_{3,4}$, $\mathfrak{M}^{(0---0)}Q_{3,4}$ | |

**Table 2:** Operator spectroscopy for the theory of Figure 3a, for its GK dual (Figure 3b) and for the $\mathcal{T}^T$-dual theory (Figure 3c), represented in the first, second and third quiver respectively, together with the fields' names. In this table and throughout the whole paper, the following notation for monopole operators is used: given a theory with gauge nodes $U(N_1) \times U(N_2) \times \ldots$, $\mathfrak{M}^{(0+\ldots)}$ denotes the monopole operator with $U(N_1)$ flux $\{0,0,\ldots,0\}$, $U(N_2)$ flux $\{+1,0,\ldots,0\}$ and so on. Moreover, note that traces are taken over gauge indices, and hence the multiplicity of the operators. In particular, considering the $\mathcal{T}^T$-dual theory in the last column, the operators in the third and fourth lines of the table are in the adjoint of one of the two SU(2)s, while the operators in the last line of the table are in the fundamental of both the two SU(2)s.

**Operators.** The operators appearing in the first orders of the superconformal index (2.6) can be identified explicitly in each duality frame in Figure 3. The results for the starting frame are summarised in Table 2. As anticipated above, one observes that the Coulomb and Higgs branch operators in the $\mathcal{T}^T$-dual theory correspond to operators in the CSM theories which are not pure monopole operators or mesons since dressed monopole operators appear. Indeed a monopole operator with non-zero flux for certain gauge factors having a CS-level must be dressed with bifundamental fields to ensure gauge invariance. Therefore, instead of the usual Coulomb branch $\leftrightarrow$ Higgs branch swap operated by mirror symmetry (via the $\mathcal{S}$ generator of the $\mathrm{SL}(2,\mathbb{Z})$ duality group) on $T_\sigma^\rho[\mathrm{SU}(N)]$ theories, one now has the map

$$\mathrm{CSM}_{|\kappa|=1} \xrightarrow{\ \mathcal{T}^T\ } T_\sigma^\rho[\mathrm{SU}(N)]$$

$$\text{A branch} \longrightarrow \text{Coulomb branch}\,, \tag{2.12a}$$

$$\text{B branch} \longrightarrow \text{Higgs branch}\,, \tag{2.12b}$$

operated by the $\mathcal{T}^T$ generator of the $\mathrm{SL}(2,\mathbb{Z})$ duality group.

**Higgsing pattern.** The dual 3d linear quiver theory exhibits Higgs/Coulomb branch Higgsings as shown in Figures 4d and 4b. This has to be compared to the B/A branch Higgsings of the electric CSM shown in Figures 4a and 4c.

Note that such Higgs mechanisms are manifest in the CSM brane systems in the analogous fashion as in the standard D3-D5-NS5 configurations, see for example [15, 22–24].

- Motion of D3-branes between NS5-branes yields Higgsing along A branch. (Or dually, D3-branes moving between NS5-branes, which are Coulomb branch motions.)

- Motion of D3-branes between $(1,1)$ 5-branes yields Higgsing along B branch. (Dually, D3-branes moving between D5-branes, which are Higgs branch motions.)

Therefore, one arrives at the following lessons: The A branch Higgsing does not change the position of the CS-levels, while B branch Higgsing moves them along the quiver (this becomes clearer when the brane configurations are drawn in Section 2.2).

Furthermore, note that in Figures 4a and 4c the GK dual of the original CSM theory has been used to draw the Hasse diagrams, the reason being that their brane motions are exactly the analogous of those happening in the $\mathcal{T}^T$-dual. Indeed, as argued in Appendix D, the GK duality is the $\mathcal{T}^T$-analogous of the brane creation/annihilation [22]. This means that, as the stable $\mathcal{T}^T$-dual theory is reached after a set of brane creation/annihilation moves, the most natural electric theory to which its brane motions can be compared to is the GK dual of the CSM model.

**Conclusion.** This abelian example has shown the following: (i) one can map operators exactly between the $\mathrm{SL}(2,\mathbb{Z})$ dual theories. (ii) The A/B branches are understood in the brane system via D3-segments moving between distinct types of 5-branes, see also [15]. (iii) The Higgsing transition geometries are straightforwardly extracted.

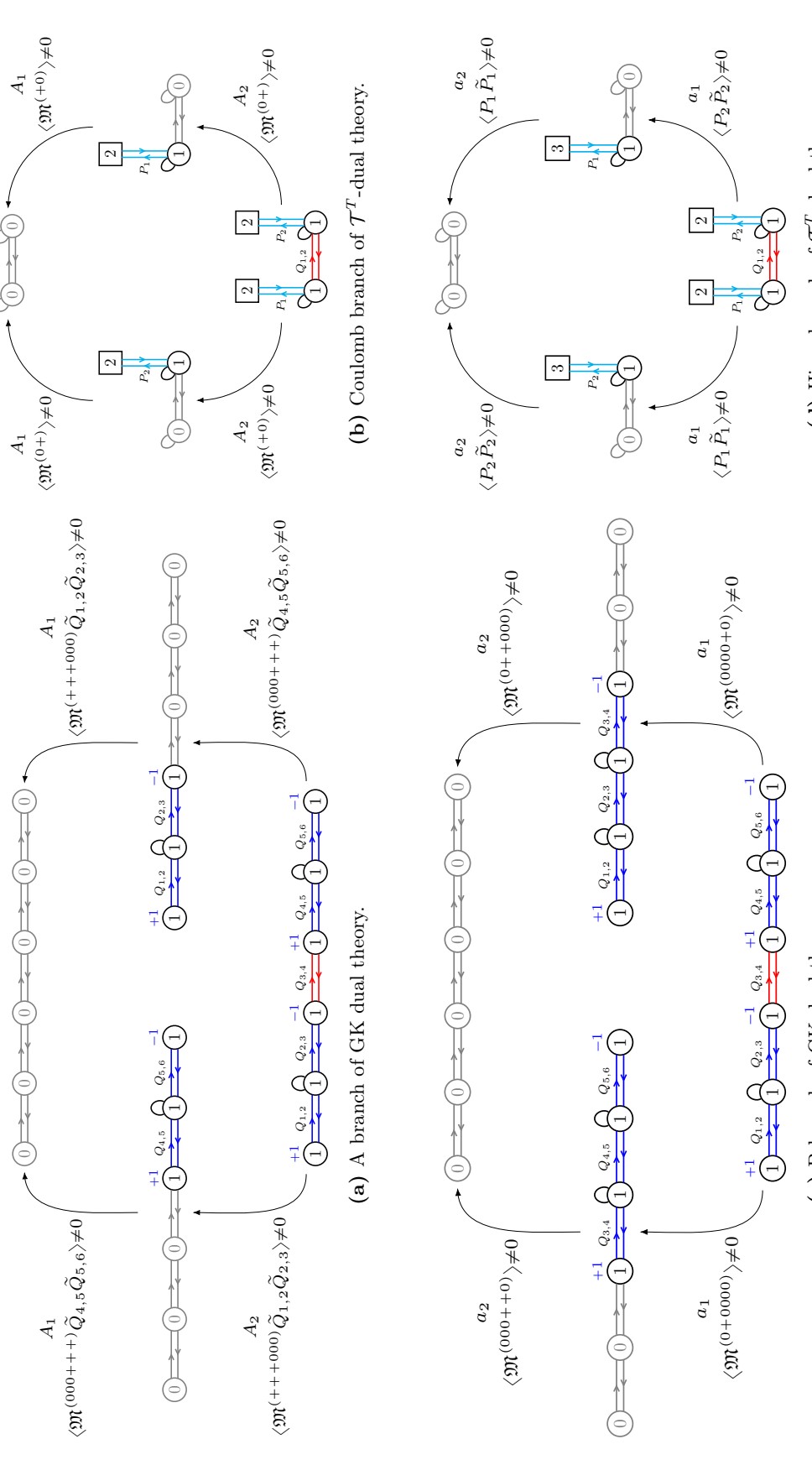

**Figure 4:** Hasse diagrams of maximal branches of moduli space. In each step the names assigned to the fields have nothing to do with the names of the fields of the prior/subsequent step. Close to the arrows, both the geometric transition and the operator taking VEV are written. The grey pieces are the trivial parts of the quiver, and are depicted just to ease tracking the changes. (a): A branch of the GK-dual theory, which agrees with the Coulomb branch Hasse diagram (b) of the $\mathcal{T}^T$-dual. (c): B branch Hasse diagram of the GK-dual theory, which matches the Higgs branch Hasse diagram (d) of the $\mathcal{T}^T$-dual.

## 2.2 Example: non-Abelian CSM theory

Now it is time to approach non-Abelian CSM theories. To begin with consider the theory in Figure 5a. Equivalently, via GK duality, one can also consider theory in Figure 5b; moreover, one has the $\mathcal{T}^T$-dual theory depicted in Figure 5c (with the parametrisation dictated by the dualisation algorithm).

As a remark, it is crucial to notice that the quiver in Figure 5b, which contains a plateau of $n = 3$ blue links connecting gauge nodes of rank $N = 3$ with CS-levels $\pm 1$ on the sides, is a good theory because the condition $n \geqslant N$ is satisfied. This is clear from the $\mathcal{T}^T$-dual theory perspective in Figure 5c. This fact is further generalized through the notion of *plateau balancing* in Section 2.3. On the other hand, the setup in Figure 5a has a shorter plateau but it also has CS-levels on the external nodes of the quiver, meaning that some GK moves can be used to move the $(1, 1)$ 5-branes towards the centre of the quiver and apply the above logic. This is a further reason why working with the GK dual is somewhat more convenient.

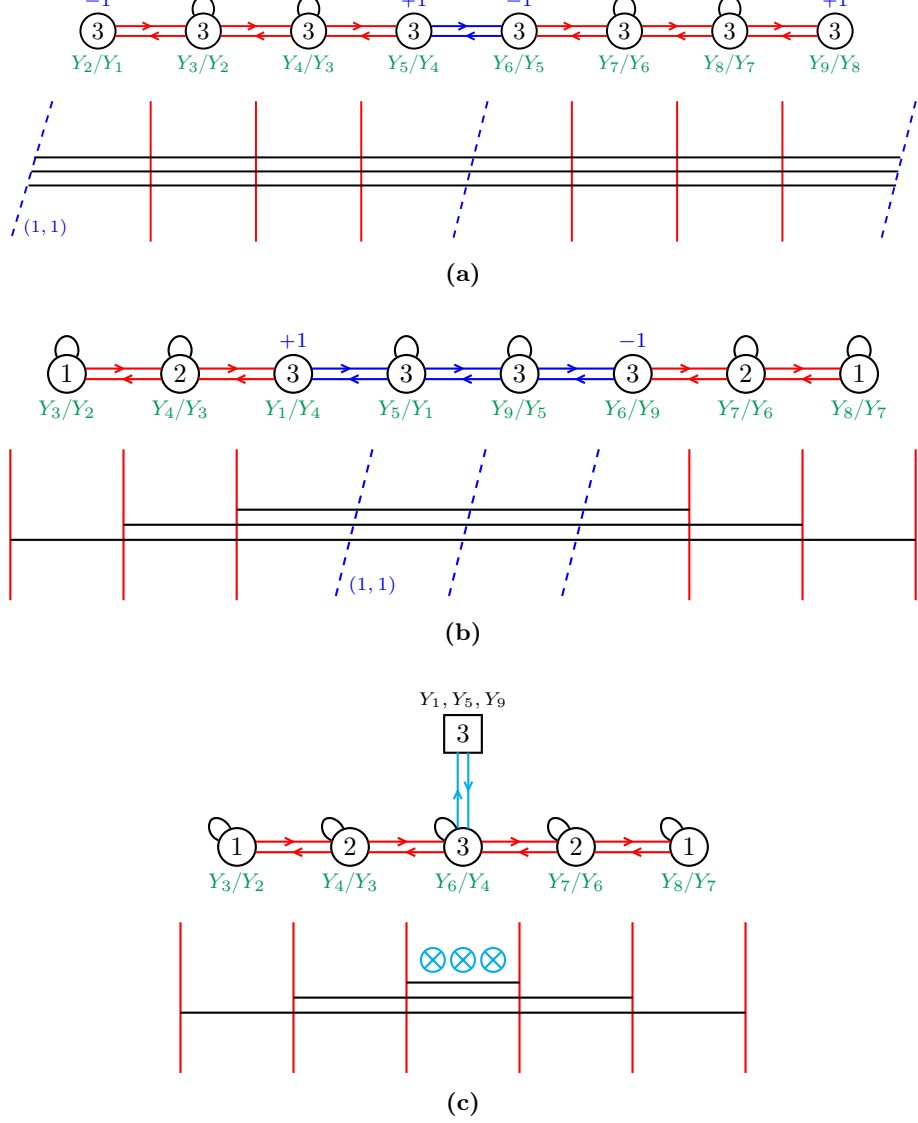

**Figure 5:** (a): The electric theory, together with the associate brane system. (b): The result of the GK duality on the starting electric theory above, together with the associate brane system. (c): The $\mathcal{T}^T$-dual theory, together with the associate brane system.

**Index.** The global symmetry algebra of all the theories in Figure 5 is

$$[\mathfrak{su}(3)_z \oplus \mathfrak{su}(3)_w \oplus \mathfrak{u}(1)_q]_A \oplus [\mathfrak{su}(3)_v]_B \,, \tag{2.13}$$

and the perturbative expansion of the refined index is evaluated to[4]

$$
\begin{aligned}
\mathcal{I} =&\, 1 + x(t^2[1,1]_v + t^{-2}(1 + [1,1]_w + [1,1]_z)) \\
&+ x^{3/2}(t^{-3}([0,1]_w[1,0]_z q + [1,0]_w[0,1]_z q^{-1})) \\
&+ x^2(t^4(2[1,1]_v + [2,2]_v + 1) \\
&\quad + t^{-4}([2,2]_w + [2,2]_z + 2[1,1]_w + 2[1,1]_z + [1,1]_w[1,1]_z + 2) \\
&\quad + [1,1]_v[1,1]_w + [1,1]_v[1,1]_z - [1,1]_w - [1,1]_z - 2) + O(x^{5/2}) \,.
\end{aligned}
\tag{2.14}
$$

The employed fugacity map for the theory in Figure 5a reads

$$
\begin{aligned}
Y_1/Y_5 &= \frac{v_1^2}{v_2}\,, & Y_5/Y_9 &= \frac{v_2^2}{v_1}\,, & Y_2/Y_3 &= \frac{z_1^2}{z_2}\,, & Y_3/Y_4 &= \frac{z_2^2}{z_1}\,, \\
Y_6/Y_7 &= \frac{w_1^2}{w_2}\,, & Y_7/Y_8 &= \frac{w_2^2}{w_1}\,, & Y_3/Y_7 &= q\frac{z_2}{z_1}\frac{w_1}{w_2}\,.
\end{aligned}
\tag{2.15}
$$

Analogously, one can build the fugacity map for the GK dual (Figure 5b) and for the $\mathcal{T}^T$-dual (Figure 5c). In these cases the fugacity map differs from the one in (2.15). From the index expansion (2.14), one also finds the global from of the symmetry group:

$$\left[\frac{\mathrm{SU}(3)_z \times \mathrm{SU}(3)_w \times \mathrm{U}(1)_q}{\mathbb{Z}_3 \times \mathbb{Z}_3}\right]_A \times [\mathrm{PSU}(3)_v]_B \,, \tag{2.16}$$

wherein the $\mathrm{SU}(3)_z \times \mathrm{SU}(3)_w$ centres $\mathbb{Z}_3 \times \mathbb{Z}_3$ act on $\mathrm{U}(1)_q$ with charges $(-1,1)$.

**B branch Higgsing.** Consider the B branch Hasse diagram of the CSM in Figure 7, and analyse the flow between the different "levels" (lvs) on the leftmost part of the diagram via brane motions and field theory. One finds the following:

lv 0) A VEV is given to the $\mathfrak{su}(3)$ moment map produced by the balanced nodes in the centre of the blue hypermultiplets' plateau. Indeed, one finds 8 operators at order $xt^2$ forming the adjoint representation of $\mathfrak{su}(3)$:

| | | | | |
|---|---|---|---|---|
| positive roots | $\mathfrak{M}^{(000+0000)}$, | $\mathfrak{M}^{(0000+000)}$, | $\mathfrak{M}^{(000++000)}$, | (2.17a) |
| negative roots | $\mathfrak{M}^{(000-0000)}$, | $\mathfrak{M}^{(0000-000)}$, | $\mathfrak{M}^{(000--000)}$, | (2.17b) |
| Cartan elements | $\mathrm{Tr}\,[A_4]$, | $\mathrm{Tr}\,[A_5]$ . | | (2.17c) |

In the brane system, there are three $(1,1)$ 5-branes in the centre that have the same linking numbers[5]; hence, they give rise to an $\mathfrak{su}(3)$ symmetry. It is then no surprise that the flavour current is generated by standard monopole operators.

lv 1) There is a $\mathfrak{u}(1)$ global symmetry resulting from the red twisted hypermultiplet[6] $(3-4)$ such that the operator acquiring a VEV is at order $xt^2$.

This $\mathfrak{u}(1)$ factor is manifest in the branes due to a single NS5 in between two $(1,1)$ 5-branes.

---

[4]From the index expansion (2.14), it is possible to recognise the extra SUSY currents, which signals the enhancement to $\mathcal{N} = 4$. This is analogous to Section 2.1.

[5]In [22], the linking numbers for D5 and NS5 branes are defined. Here, the argument relies on the known $\mathcal{T}^T$ transformation of the brane system with D3, NS5, and $(1,1)$ 5-branes to that with D3, NS5, D5 branes. Hence, one may use those linking numbers here, but only for CS-level $|\kappa| = 1$.

[6]In this discussion, $(i - j)$ is the hypermultiplet connecting the $i$-th and the $j$-th gauge nodes.

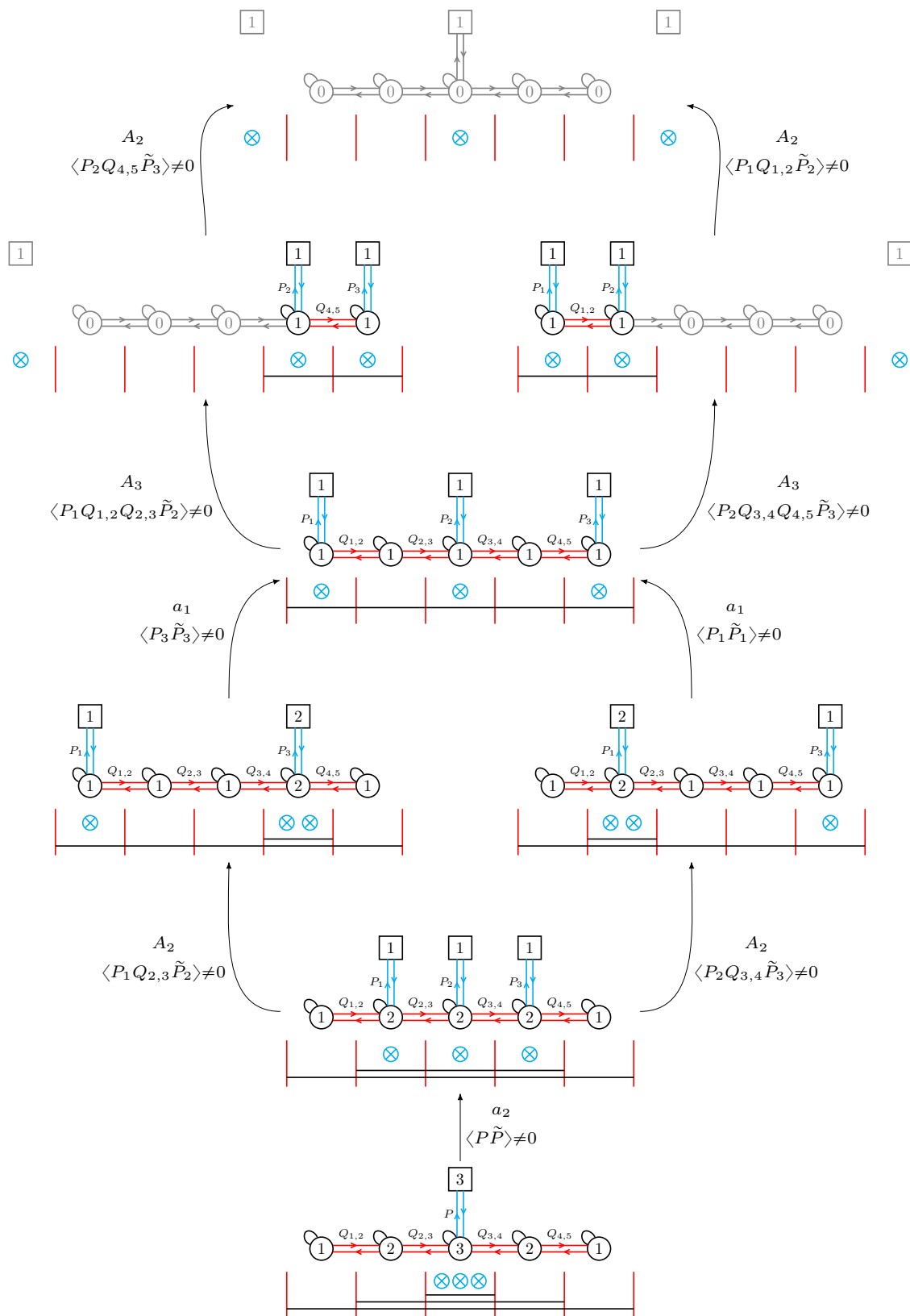

**Figure 6:** The Higgs branch transitions for the $\mathcal{T}^T$-dual theory (Figure 5c). In each step the field names have noting to do with the names of the fields of the prior/subsequent step.

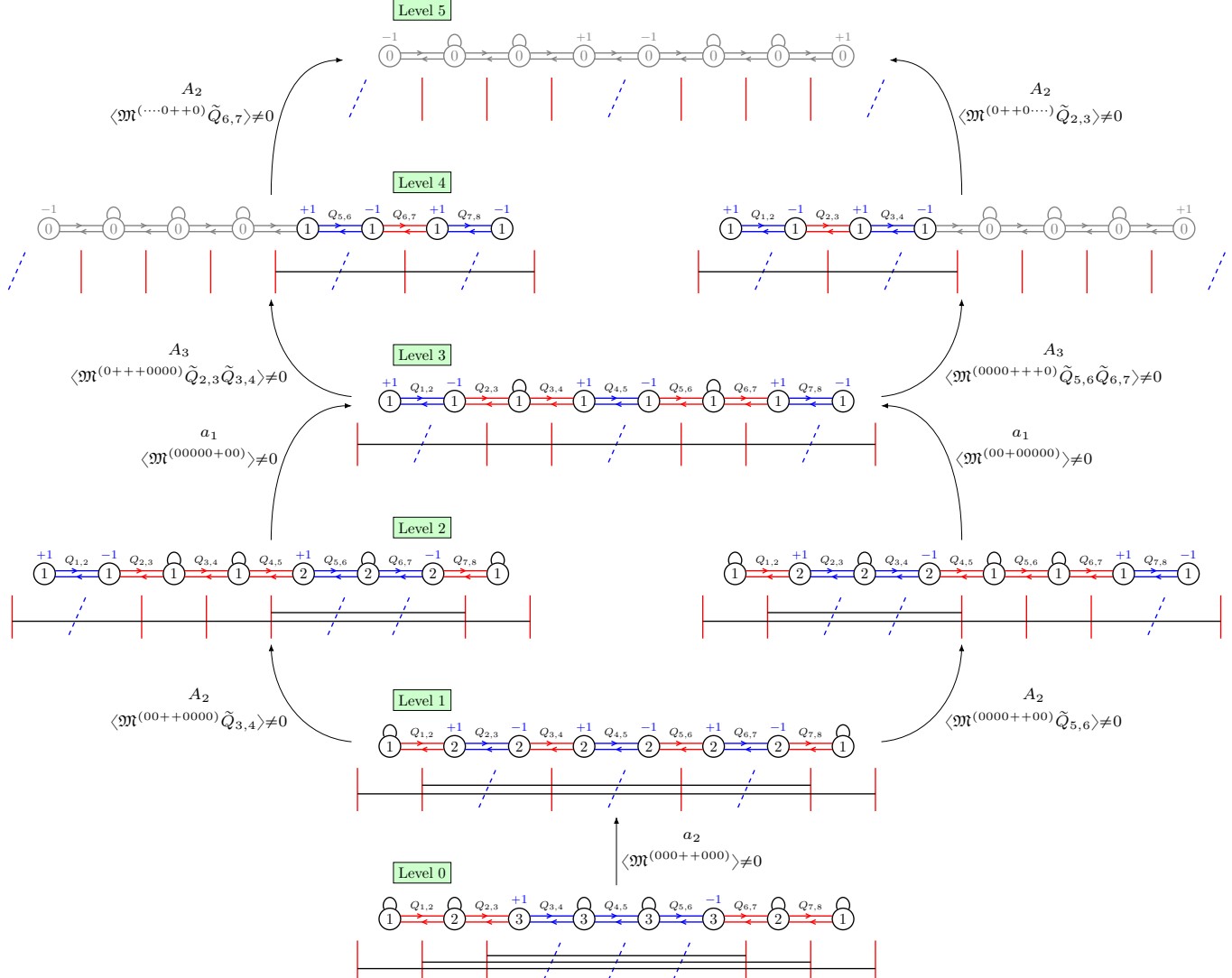

**Figure 7:** The B branch Higgsing for the GK-dual CS theory (5b). In each step the names assigned to the fields have noting to do with the names of the fields of the prior/subsequent step. Recall the notation for monopole operators: for a theory with gauge nodes $U(N_1) \times U(N_2) \times \ldots$, denote by $\mathfrak{M}^{(0+\cdots)}$ the monopole operator with $U(N_1)$ flux $\{0,0,\ldots,0\}$, $U(N_2)$ flux $\{+1,0,\ldots,0\}$ and so on. When considering a monopole operator in a quiver where some nodes have zero rank (depicted in gray only for convenience), the empty slots in the flux are represented by dots.

lv 2) A VEV is given to the $\mathfrak{su}(2)$ moment map produced by the balanced node in the centre of the blue hypermultiplets' plateau $(5-6)$ and $(6-7)$. Indeed, one finds 3 operators at order $xt^2$ forming the adjoint representation of $\mathfrak{su}(2)$:

$$\text{postive root} \qquad \mathfrak{M}^{(00+00000)}, \qquad (2.18a)$$

$$\text{negative root} \qquad \mathfrak{M}^{(00-00000)}, \qquad (2.18b)$$

$$\text{Cartan element} \qquad \text{Tr}\,[A_6]. \qquad (2.18c)$$

Again, this B-branch symmetry factor stems from two adjacent $(1,1)$ 5-branes that identical linking numbers.

lv 3) There is a $\mathfrak{u}(1)$ symmetry induced by the plateau of two red twisted hypermultiplets $2, 3$ and $3, 4$ such that the operator that acquires a VEV is at order $x^2t^4$. This agrees

with the slice being $A_3$ for which the highest degree generator[7] has degree 2.

Here, one finds two $(1,1)$ 5-branes with two NS5 branes in between. Hence, this gives rise to a $\mathfrak{u}(1)$ and the charged operator connects the two hypermultiplets induced by the NS5 brane junctions.

lv 4) There is a $\mathfrak{u}(1)$ due to the red twisted hypermultiplet $6, 7$ such that the operator that acquires a VEV is at order $x^{\frac{3}{2}}t^3$. Again, this agrees with and $A_2$ slice geometry.

The brane system is that of a single NS5 in between two $(1,1)$ 5-branes.

This matches exactly the Higgsing along the Higgs branch of the $\mathcal{T}^T$ dual theory, as shown in Figure 6. This can be workout on the brane system (see for instance [55, 56]) or directly on the $\mathcal{T}^T$ dual quiver (see [42]).

**A branch Higgsing.** For the A branch Hasse diagram of Figure 9, the RG-flow analysis is done analogously. Focusing on the leftmost part, one finds the following:

lv 0) A VEV is given to the $\mathfrak{su}(3)$ moment map that is generated by the two leftmost nodes, which are balanced. The operator acquiring a VEV is at order $xt^{-2}$.

In the brane system, this is traced back to three NS5 branes with the identical linking numbers. Hence, the standard concept of balanced gauge nodes applies.

lv 1) Next, give a VEV to the $\mathfrak{su}(4)$ moment map produced by the two rightmost nodes, which are balanced, and by the plateau of blue hypermultiplets. In fact, one finds 15 operators at order $xt^{-2}$ forming the adjoint representation of $\mathfrak{su}(4)$:

$$\mathfrak{M}^{(\cdot 0++++++)}\widetilde{Q}_{3,4}\widetilde{Q}_{4,5}\widetilde{Q}_{5,6}, \quad \mathfrak{M}^{(\cdot 0------)}Q_{3,4}Q_{4,5}Q_{5,6}, \tag{2.19a}$$

$$\mathfrak{M}^{(\cdot 0+++++0)}\widetilde{Q}_{3,4}\widetilde{Q}_{4,5}\widetilde{Q}_{5,6}, \quad \mathfrak{M}^{(\cdot 0-----0)}Q_{3,4}Q_{4,5}Q_{5,6}, \tag{2.19b}$$

$$\mathfrak{M}^{(\cdot 0++++00)}\widetilde{Q}_{3,4}\widetilde{Q}_{4,5}\widetilde{Q}_{5,6}, \quad \mathfrak{M}^{(\cdot 0----00)}Q_{3,4}Q_{4,5}Q_{5,6}, \tag{2.19c}$$

$$\mathfrak{M}^{(\cdot 00000++)}, \quad \mathfrak{M}^{(\cdot 00000--)}, \tag{2.19d}$$

$$\mathfrak{M}^{(\cdot 00000+0)}, \quad \mathfrak{M}^{(\cdot 00000-0)}, \tag{2.19e}$$

$$\mathfrak{M}^{(\cdot 000000+)}, \quad \mathfrak{M}^{(\cdot 000000-)}, \tag{2.19f}$$

$$\mathrm{Tr}\left[Q_{3,4}\widetilde{Q}_{3,4}\right] \overset{\text{F-terms}}{=} \mathrm{Tr}\left[Q_{4,5}\widetilde{Q}_{4,5}\right] \overset{\text{F-terms}}{=} \mathrm{Tr}\left[Q_{5,6}\widetilde{Q}_{5,6}\right], \tag{2.19g}$$

$$\mathrm{Tr}\left[A_7\right], \quad \mathrm{Tr}\left[A_8\right]. \tag{2.19h}$$

Here, the first six lines are the 6 positive roots (along with the negative roots), while the last two lines display the 3 Cartan elements.

Here, the brane system exhibits a more interesting configuration: the three NS5s on the right-hand-side have the same linking number; but the next NS5 to their left also has the same linking number, because the in-between $(1,1)$ 5-brane compensate for the different net-number of D3s. Therefore, the monopole operators include the standard ones (2.19d)–(2.19f) from the stack of three consecutive NS5; which are complemented by (2.19a)–(2.19c), which need to have identical magnetic fluxes in all the gauge nodes between the two NS5s with three $(1,1)$ 5-branes. This follows, because a single D3 segment moves in between those two NS5 branes.

lv 2) A VEV is given to the $\mathfrak{su}(2)$ moment map produced by the leftmost node, which is balanced. The operator taking a VEV is at order $xt^{-2}$.

This symmetry follows from the two NS5s with identical linking numbers.

---

[7]The Kleinian singularity $A_k$ with $uv = z^{k+1}$ has $\deg(z) = 1$ and $\deg(u) = \deg(v) = \frac{k+1}{2}$.

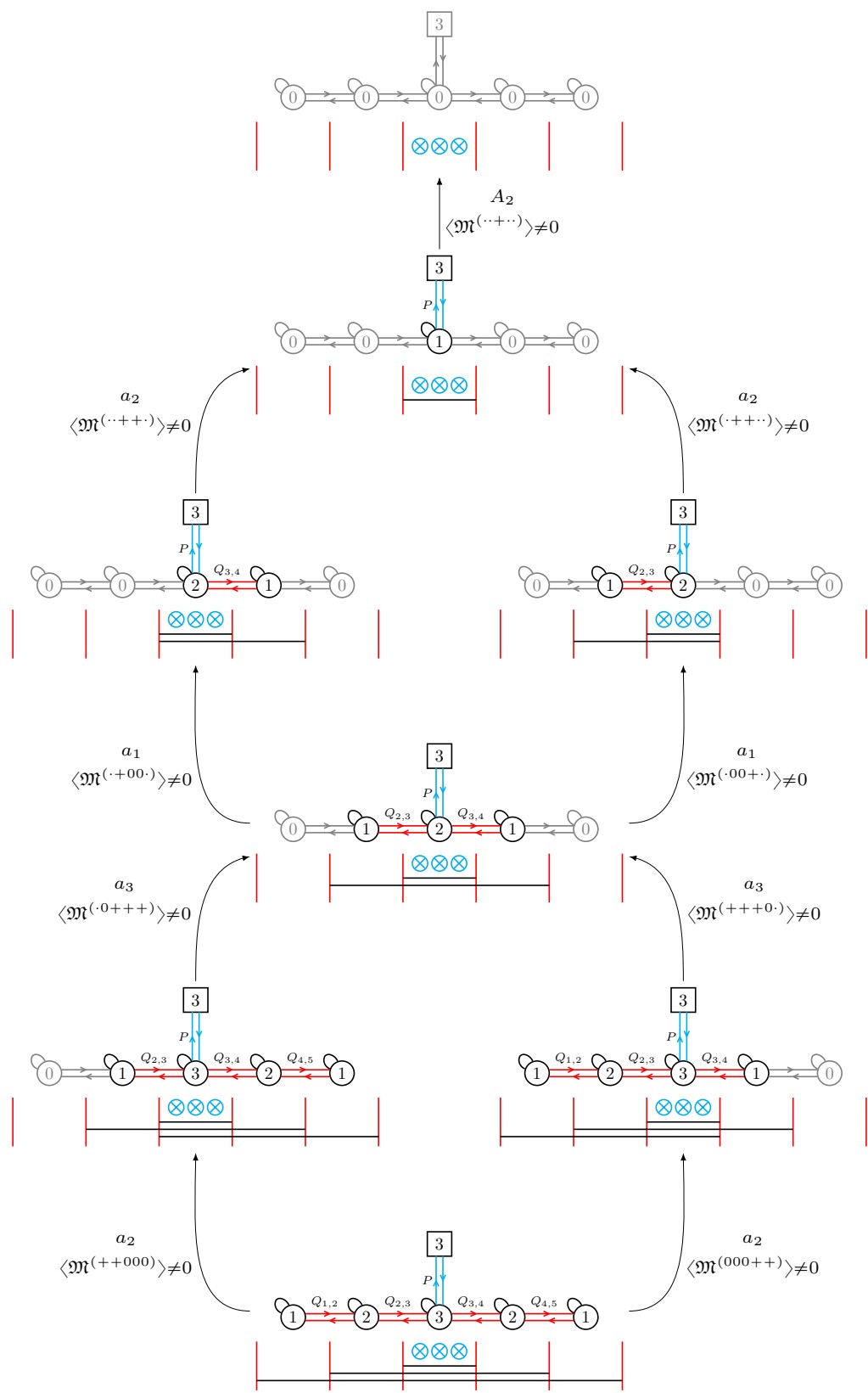

**Figure 8:** The Coulomb branch transitions for the $\mathcal{T}^T$-dual theory (Figure 5c). In each step the assigned field names have noting to do with the names of the fields of the prior/subsequent step. Recall the notation for monopole operators: for a theory with gauge nodes $\mathrm{U}(N_1) \times \mathrm{U}(N_2) \times \ldots$ denote by $\mathfrak{M}^{(0+\cdots)}$ the monopole operator with $\mathrm{U}(N_1)$ flux $\{0, 0, \ldots, 0\}$, $\mathrm{U}(N_2)$ flux $\{+1, 0, \ldots, 0\}$ and so on. When considering a monopole operator in a quiver where some nodes have zero rank (which are depicted for convenience), the empty slots in the monopole flux are represented as dots.

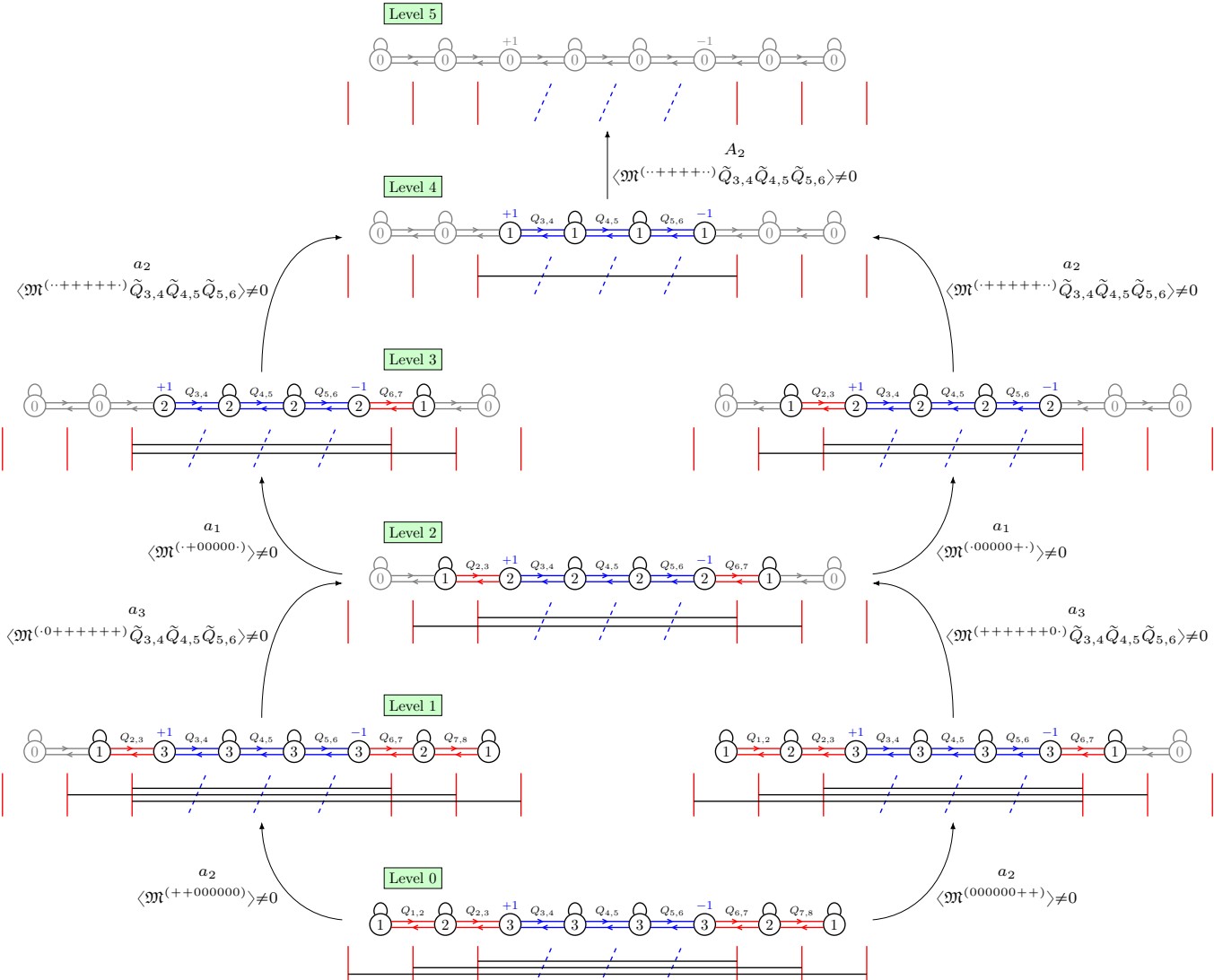

**Figure 9:** The A branch Higgsing for the GK-electric CS theory (Figure 5b). In each step the names assigned to the fields have noting to do with the names of the fields of the prior/subsequent step. Recall the notation for monopole operators: for a theory with gauge nodes $\mathrm{U}(N_1)\times\mathrm{U}(N_2)\times\dots$ denote by $\mathfrak{M}^{(0+\cdots)}$ the monopole operator with $\mathrm{U}(N_1)$ flux $\{0,0,\dots,0\}$, $\mathrm{U}(N_2)$ flux $\{+1,0,\dots,0\}$ and so on. When considering a monopole operator in a quiver where some nodes have zero rank (which are depicted for convenience), the empty slots in the monopole flux are represented as dots.

lv 3) A VEV is given to the $\mathfrak{su}(3)$ moment map produced by the rightmost node, which is balanced, together with the plateau of blue hypermultiplets. In fact, there are 8 operators at order $xt^{-2}$ forming the adjoint representation of $\mathfrak{su}(3)$:

$$\mathfrak{M}^{(\cdot\cdot+++++\cdot)}\widetilde{Q}_{3,4}\widetilde{Q}_{4,5}\widetilde{Q}_{5,6}\,, \quad \mathfrak{M}^{(\cdot\cdot-----\cdot)}Q_{3,4}Q_{4,5}Q_{5,6}\,, \tag{2.20a}$$

$$\mathfrak{M}^{(\cdot\cdot++++0\cdot)}\widetilde{Q}_{3,4}\widetilde{Q}_{4,5}\widetilde{Q}_{5,6}\,, \quad \mathfrak{M}^{(\cdot\cdot----0\cdot)}Q_{3,4}Q_{4,5}Q_{5,6}\,, \tag{2.20b}$$

$$\mathfrak{M}^{(\cdot\cdot0000+\cdot)}\,, \quad \mathfrak{M}^{(\cdot\cdot0000-\cdot)}\,, \tag{2.20c}$$

$$\mathrm{Tr}\left[Q_{3,4}\widetilde{Q}_{3,4}\right] \overset{\text{F-terms}}{=} \mathrm{Tr}\left[Q_{4,5}\widetilde{Q}_{4,5}\right] \overset{\text{F-terms}}{=} \mathrm{Tr}\left[Q_{5,6}\widetilde{Q}_{5,6}\right]\,, \tag{2.20d}$$

$$\mathrm{Tr}\left[A_7\right]\,. \tag{2.20e}$$

which split into the 3 positive and 3 negative roots, alongside the 2 Cartan elements.

This situations is by now familiar. There are two adjacent NS5s with identical linking numbers on the right-hand-side – this gives rise to the standard monopoles (2.20c). The next NS5 to the left, which has three $(1,1)$ 5-branes before it, also has the same linking number; thus, the symmetry is SU(3). The additional roots again come from D3-segments that only span between NS5 branes, implying monopole operators with equal magnetic fluxes in all gauge nodes that are affected, see (2.20a)–(2.20b).

lv 4) There is a $\mathfrak{u}(1)$ due to the plateau of blue hypermultiplets, such that the non-trivially charged operator that acquires a VEV is at order $x^{\frac{3}{2}}t^{-3}$, indicating an $A_2$ transition.

In the brane system, this is manifest in two NS5 branes with different linking numbers.

This agrees with the Coulomb branch Hasse diagram (Figure 8) of the $\mathcal{T}^T$ dual theory, which is deduced from the brane system or directly on the $\mathcal{T}^T$ dual quiver [40, 41].

**Conclusions.** The non-abelian example has served to highlight certain features: (i) the analysis on the brane system is in spirit very similar to the standard D3-NS5-D5 systems. (ii) New features arise from neighbouring NS5 branes that have some $(1,1)$ 5-branes in between. If the NS5s have the same linking number, there is a non-Abelian symmetry generator by monopole operators that have identical fluxes in several gauge nodes. This is a consequence of the fact that the single D3 segment between the two NS5 is just one A-branch moduli, but it induces several gauge nodes in the CSM quiver.

## 2.3 Generalisations

The discussion so far allows one to extract generic features; for example: for linear $\mathrm{CSM}_{|\kappa|=1}$ theory, the *good* condition is identical to that of the 3d $\mathcal{N}=4$ dual. However, in terms of the CSM quiver, the deduction proceeds differently compare to $T^\sigma_\rho[\mathrm{SU}(N)]$ theories. One such instance is the A/B branch isometry (or more generally A/B branch directions) that stem from an entire plateau within the CSM quiver.

To discuss these features systematically, two basic CSM families are introduced below. For these, the $\mathcal{T}^T$ duals, along with global symmetries, and RG-flow Hasse diagrams are analysed. For convenience, one may restrict to theories where GK duality brings all $(1,\pm1)$-branes as close to the system's centre as possible.

### 2.3.1 Family I

Consider the family of CSM theories in Figure 10a, composed of a set of (blue) hypermultiplets connecting a plateau of $n+1$ nodes of rank[8] $N$ (where only the two external nodes

---

[8]Here the case in which no more GK moves are necessary is considered. If one instead has nodes of different ranks, namely different numbers of D3-branes on the two sides of the $(1,\pm1)$ realising the (blue) hypermultiplets, then they can be swapped with NS5-branes via the GK move.

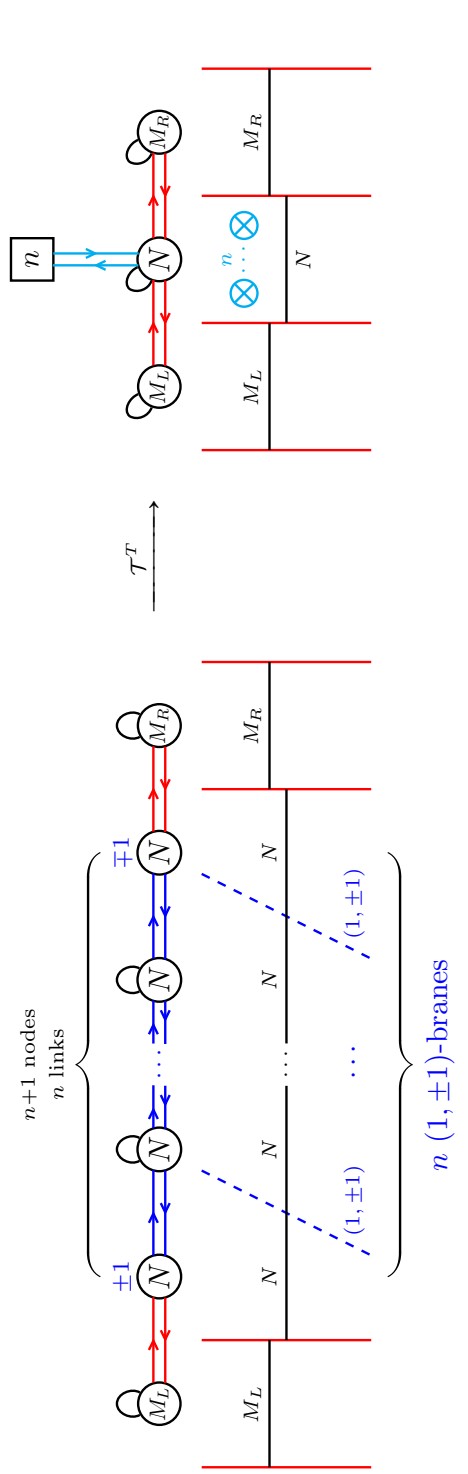

**(a)** The CSM family I (on the left) together with the $\mathcal{T}^{\mathcal{T}}$-dual (on the right).

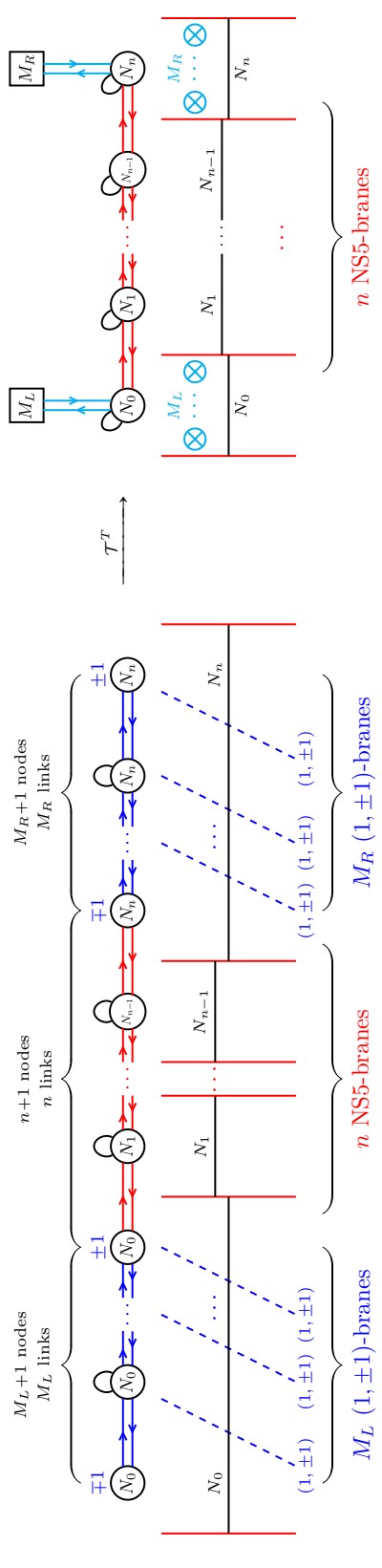

**(b)** The CSM family II (on the left) together with the $\mathcal{T}^{\mathcal{T}}$-dual (on the right).

**Figure 10:** The quiver theories are shown together with the brane configurations. The symbols over D3 and D5 branes denote their multiplicity. Here by assumption all the $(1, \pm 1)$-branes are already as close to the centre of the brane system as possible, meaning that no more GK moves can be performed.

have a CS-level) attached via two (red) twisted hypermultiplets to two extra nodes, one on the left and one on the right, of rank $M_L$ and $M_R$ respectively. The $\mathcal{T}^T$-dual theory is a simple three-node 3d $\mathcal{N} = 4$ quiver shown in Figure 10a as well.

In order for the CSM theory to be good, which is inferred from $\mathcal{T}^T$-dual, the conditions $n + M_L + M_R \geqslant 2N$ and $N \geqslant 2M_{L/R}$ have to be satisfied. Inspired from the $\mathcal{T}^T$-dual theory, it is suggestive to interpret the first condition as a constraint on the entire *plateau of blue hypermultiplets*. One may then define the *plateau balance*[9]

$$b = n + M_L + M_R - 2N \tag{2.21}$$

such that the *the plateau is balanced* if $b = 0$.

Next, the A/B branch of this CSM theory behaves as follows:

**A branch.** The A branch isometry algebra ranges from $\mathfrak{u}(1)^{\oplus 3}$ (for $b > 0$ and $N > 2M_{L/R}$) to at most $\mathfrak{su}(4)$ (for $b = 0$ and $N = 2M_{L/R}$). The $\mathfrak{u}(1)$ factors enhance to non-Abelian factors depending on $b \geqslant 0$ and $N \geqslant 2M_{L/R}$, such that partial enhancements like $\mathfrak{u}(1)^{\oplus 2} \oplus \mathfrak{su}(2)$ are possible.

Next, focus specifically on the blue plateau and its impact on A branch. For $b > 0$, the global symmetry contribution is $\mathfrak{u}(1)$. The Cartan element is the trace of the meson made out of the one of the (blue) hypermultiplets (all the traces are identified by means of the F-terms). For $b > 1$, the global symmetry contribution enhances to $\mathfrak{su}(2)$. The extra two roots are identified as the monopoles with flux $(\pm 1, 0, \ldots, 0)$ for all the nodes inside the blue plateau, dressed with the hypermultiplets in order to be gauge invariant. In the brane system, these monopole operators with identical fluxes in several gauge nodes are attributed to the single D3-segment that can move between the NS5 branes with $n$ $(1, \pm 1)$ 5-branes in between; see for instance [15] for related discussions.

Likewise, in terms of A branch Higgsing, the decay and fission algorithm for the $\mathcal{T}^T$-dual gives rise to a $A_{b+1}$-type Higgsing (where $b$ is the balance, see [40, 41]). Of course, if one (or both) of the outermost gauge nodes are balanced as well, the transition type changes to a minimal nilpotent orbit of $A$-type. From the brane system in Figure 10a, these transitions are readily identified from the D3-segments that can move between the NS5 branes — the defining feature of the A branch.

**B branch.** The B branch has an $\mathfrak{su}(n)$ isometry, which is transparent as the Higgs branch isometry of the $\mathcal{T}^T$-dual theory.

One may ask which operators in the CSM side realise this flavour current. The roots of $\mathfrak{su}(n)$ are the monopole operators with flux $(\pm 1, 0, \ldots, 0)$ for one or more of the nodes without a CS-level inside the blue plateau, while the Cartan elements come from the traces of the adjoints of the same nodes just mentioned. The brane system in Figure 10a manifests this $\mathfrak{su}(n)$ via the stack of $(1, \pm 1)$ branes in the centre.

This non-Abelian isometry gives rise to B branch RG-flow of type $a_{n-1}$, see [55, 56]. The dimensionality, for examples, follows from the $n-1$ D3 brane segments that can move independently in between the stack of $n$ $(1, \pm 1)$ 5-branes.

### 2.3.2   Family II

Consider the CSM quiver family in Figure 10b, composed of a set of (red) twisted hypermultiplets connecting $n+1$ nodes attached to two sets of (blue) hypermultiplets connecting two plateaux of $M_L + 1$ nodes and $M_R + 1$ nodes respectively (having only the two external nodes with CS-level). The $\mathcal{T}^T$-dual is then a simple $(n + 1)$-node 3d $\mathcal{N} = 4$ quiver, as shown in Figure 10b.

---

[9]This is the $\mathcal{T}^T$-dual of the balance notion for central node in the $T_\sigma^\rho[\mathrm{SU}(...)]$ theory of Figure 10a.

Whether the CSM theory is *good* can be judged as follows: The blue plateau (either on the left or right hand side) needs to have non-negative plateau balance (2.21), as discussed above. The other gauge nodes in between red hypermultiplets need to be *good* in the standard sense, which follows from the $\mathcal{T}^T$-dual.

Next, analyse the A/B branch in more detail:

**A branch.** If all the nodes connected by (red) twisted hypermultiplets have the same rank, namely $N_i = N$ for all $i = 0, 1, \ldots, n$, then the A branch has at least a $\mathfrak{su}(n) \oplus \mathfrak{u}(1)^{\oplus 2}$ isometry. Indeed, the $\mathcal{T}^T$-dual theory has all the $n-1$ nodes in the middle of the red plateau balanced. Alternatively, the brane system in Figure 10b has $n$ NS5 branes in the centre that have identical linking for any values of $M_{L,R}$. This then yields the $\mathfrak{su}(n)$ A branch isometry.

Moreover, if $M_L = N \vee M_R = N$ then the A branch isometry enhances to $\mathfrak{su}(n + 1)$, while if $M_L = M_R = N$ it enhances to $\mathfrak{su}(n + 2)$. All of this follows from standard arguments on $\mathcal{T}^T$-dual theory as well as from the brane systems.

On the other hand, if the nodes connected by (red) twisted hypermultiplets form a ramp, namely $N_i = i$ for all $i = 0, 1, \ldots, n$, then the A branch has at least a $\mathfrak{su}(n)$ isometry[10]. Moreover, if $M_R = 2N_n - N_{n-1}$ the A branch isometry enhances to $\mathfrak{su}(n + 1) = a_n$. This conditions ensure that the two right-most NS5 branes in Figure 10b have the same linking numbers, which induces the symmetry enhancement. Of course, the ramp can also be oriented the other way around.

The observation is the following: the red segment of gauge nodes yields a non-Abelian A branch isometry provided the nodes are balanced in the standard sense. The blue plateux on the left (resp. right) can in fact enhance this isometry further if the plateau balance is trivial.

The question is then which operators lead to this red segment $\mathfrak{su}(n)$ symmetry on the CSM side. The roots of the adjoint representation are the monopoles with flux $(\pm 1, 0, \ldots, 0)$ for one or more of the nodes without a CS-level among those connected by (red) twisted hypermultiplets, while the Cartan elements come from the traces of the adjoints of all the nodes connected by (red) twisted hypermultiplets. From the brane perspective, this is the standard scenario for D3-segments moving along NS5 branes, see for instance [24]. In case of the $\mathfrak{su}(n+1)$ symmetry enhancement because of the blue plateau of length $M_L$ (or $M_R$), the extra Cartan element comes from the trace of the meson made out of one of the (blue) hypermultiplets (all the traces are identified by means of the F-terms), while the extra roots come from the monopoles with flux $(\pm 1, 0, \ldots, 0)$ for all the nodes of the blue plateau and with flux $(\pm 1, 0, \ldots, 0)$ for one or more of the nodes without a CS-level among those connected by (red) twisted hypermultiplets. An analogous reasoning can be carried on when the global symmetry gets enhanced to $\mathfrak{su}(n + 2)$.

In terms of A branch Higgsing, this suggests transitions of type $a_{n-1}$ (or the enhanced versions $a_n$ and $a_{n+1}$). This is transparent in the brane system, because a minimal transition is characterised by the number of D3-segments that can move between NS5 branes with identical linking numbers.

**B branch.** Here, one finds an $\mathfrak{su}(M_L) \oplus \mathfrak{u}(1) \oplus \mathfrak{su}(M_R)$ isometry, for which no enhancement takes place.

This also implies that (minimal) B branch RG-flows are triggered by operators from one of the blue plateaux. A non-minimal Higgsing corresponds to an operator connecting both plateaux (i.e. and end-to-end Higgs branch operator in the dual).

---

[10]Necessarily, the part of the system with the $M_L$ $(1, \pm 1)$ 5-brane becomes trivial in this scenario.

# 3 Linear $\mathcal{N}=4$ Chern–Simons Matter theories, CS-levels $>1$

For the theories with CS-levels $|\kappa_i| = |\kappa| = 1$, numerous features were analysed by leveraging the $\mathcal{T}^T$-dual theory. However, for $\mathcal{N} = 4$ theories with CS-levels $|\kappa_i| = |\kappa| > 1$ (or $\mathrm{CSM}_{|\kappa|>1}$ for short), one cannot rely on such a tool since no $\mathrm{SL}(2, \mathbb{Z})$ transformation can generate a Lagrangian non-CS dual from them (see Appendix C.7 for a proof).

**A first magnetic quiver.** However, not all is lost, as one can ask for a *magnetic quiver* $\mathsf{MQ}_A$ instead. In other words, a standard (yet auxiliary) 3d $\mathcal{N} = 4$ quiver whose Coulomb branch $\mathcal{C}$ captures the A branch of the CSM theory:

$$\text{A-branch}\left(\mathrm{CSM}_{|\kappa|>1}\right) \cong \mathcal{C}(\mathsf{MQ}_A) \, . \tag{3.1}$$

If successful, one can then utilise the decay and fission algorithm [40, 41] to trace out all A branch RG-flows. Here, a simple prescription for the derivation of a magnetic quiver for a $\mathrm{CSM}_{|\kappa|>1}$ theory is proposed[11], which is summarised on the left-hand side of Figure 12.

- Firstly, write the associated brane configuration, which contains $(1, \kappa)$ 5-branes. It is convenient to use GK-duality to change the phase of the brane system[12] such that each $(1, \kappa)$ 5-brane locally looks like in Figure 11. That means the difference of D3 branes ending from the left and right is smaller than or equal to $|\kappa|$.

- Secondly, substitute each $(1, \kappa)$-brane with $|\kappa|$ D5-branes. The boundary conditions assigned to the D3s ending on these D5-branes are encoded in Figure 11.

- Thirdly, move D5s across NS5s (paying attention to brane creation or annihilation [22]) such that all D3s are suspended between NS5s — the Coulomb branch phase.

- Finally, read off the magnetic quiver $\mathsf{MQ}_A$ as the 3d $\mathcal{N} = 4$ electric theory from this *auxiliary* (but standard) D3-D5-NS5 system.

Evidence for this proposal's validity comes, for example, from comparing the A-branch limit of the superconformal index of the $\mathrm{CSM}_{|\kappa|>1}$ theory with the Coulomb branch Hilbert series of its magnetic quiver $\mathsf{MQ}_A$. Numerous such checks are detailed in Appendix B.

**Figure 11:** The brane description of the magnetic quiver recipe for a $\mathrm{CSM}_{|\kappa|>1}$ theory. Here, $N \geqslant M$ has been assumed. Moreover, $|\kappa| \geqslant N - M$ is assumed, meaning that the brane system has been transferred to a suitable phase via GK-duality. The integers close to D3 and D5 branes denote their multiplicity.

**A second magnetic quiver.** However, the B-branch of the $|\kappa_i| > 1$ CSM theory remains yet uncaptured. To address this, the $\mathcal{T}$ generator of the $\mathrm{SL}(2, \mathbb{Z})$ group proves useful.

---

[11]At this point it may seem unclear why specifically the A branch is captured. The reason is that in the conventions of this paper, the A branch corresponds to D3s moving between NS5s. Inspired by $\mathrm{SL}(2, \mathbb{Z})$ duality in the $|\kappa| = 1$ case (Figure 2), the proposal isolates the Coulomb branch moduli of D3s between NS5s and converts the bound states $(1, \kappa)$ to mere flavour branes in a standard 3d $\mathcal{N} = 4$ setting.

[12]If one chooses not to do so, one needs to pay attention to the s-rule when generalising the boundary conditions of D3s on D5 branes of Figure 11. See for example Figure 16.

- Firstly, if $\mathcal{T}$ is applied $|\kappa|$ times to a $\mathrm{CSM}_{|\kappa|>1}$ theory, the result is a new $\mathrm{CSM}'_{|\kappa|>1}$ theory whose brane realisation has the NS5 and $(1,\kappa)$-branes interchanged; namely, the A and B branches are swapped. This transformation is implemented on the QFT via the dualisation algorithm [28], as reviewed in Appendix C.5.

- Secondly, by applying the *same* prescription as above to derive the magnetic quiver (now on the brane system with NS5 and $(1,\kappa)$ branes swapped), one obtains a $\mathsf{MQ}_B$ whose Coulomb branch describes the A branch of the $\mathrm{CSM}'_{|\kappa|>1}$ theory. Due to the NS5 $\leftrightarrow (1,\kappa)$ swap operated by $\mathcal{T}^{|\kappa|}$, this A branch, however, is the same as the B branch of the original $\mathrm{CSM}_{|\kappa|>1}$ theory[13]; cf. the right-hand side of Figure 12.

Hence, one arrives at

$$\text{B-branch}\left(\mathrm{CSM}_{|\kappa|>1}\right) \cong \mathcal{C}(\mathsf{MQ}_B) \,, \tag{3.2}$$

which can be cross-checked, for instance, by matching the B branch limit of the superconformal index of the $\mathrm{CSM}_{|\kappa|>1}$ theory with the Coulomb branch Hilbert series of $\mathsf{MQ}_B$.

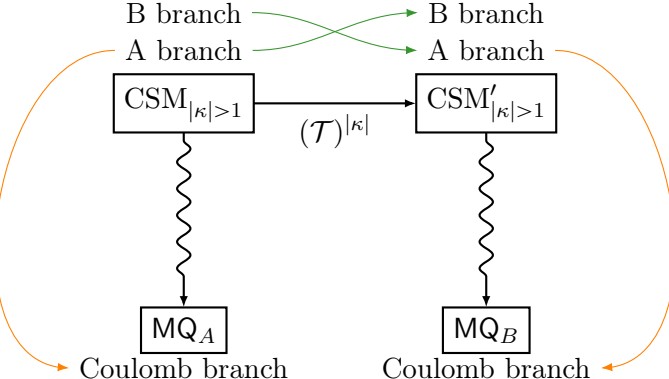

**Figure 12:** The strategy to probe both branches of a $\mathrm{CSM}_{|\kappa|>1}$ theory. The A branch is described by the magnetic quiver $\mathsf{MQ}_A$ obtained via the prescription given in the main text. The B branch is described through the A branch of the $(\mathcal{T})^{|\kappa|}$-dual $\mathrm{CSM}'_{|\kappa|>1}$ theory, which in turn is probed via the magnetic quiver $\mathsf{MQ}_B$. The wiggly arrows encapsulate the proposed recipe to derive the magnetic quivers of a CSM theory.

**Showcasing the proposal.** To summarise, the logic of this magnetic quiver proposal relies on four steps: (i) The CSM theories here are realised by a Type IIB brane system. (ii) The maximal branches of the moduli space correspond to D3 segments moving between distinct types of 5-branes. (iii) The magnetic quivers $\mathsf{MQ}_{A/B}$ capture precisely this motion for a specific branch. (iv) The replacement rule (Figure 11) follows from the identical nature of brane creation/annihilation and GK-duality.

In the remainder of this section and in Appendix B, selected examples of $\mathrm{CSM}_{|\kappa|>1}$ theories are discussed. As above, all considered examples are *good*. For $\mathrm{CSM}_{|\kappa|>1}$ theories this property cannot be inferred from the $\mathcal{T}^T$-dual since, as they do not have a non-CS Lagrangian dual. Instead, one can check that the index expansion of the $\mathrm{CSM}_{|\kappa|>1}$ theory does not show the presence of monopole operators below the unitarity bound, which is the hallmark (and defining property) of a *bad* theory. Alternatively, a more immediate check is to ensure that the associated magnetic quivers $\mathsf{MQ}_{A/B}$ are *good* as (auxiliary) 3d $\mathcal{N}=4$ theories.

---

[13]It is clear that is really the B branch, as $(\mathcal{T})^{|\kappa|}$ swapped NS5 and $(1,\kappa)$, such that the proposal now isolates D3 moduli between $(1,\kappa)$ branes.

## 3.1 Example: CSM theory with 4 nodes

Consider the prototypical CSM theory composed of unitary gauge nodes and hypermultiplets and twisted hypermultiplets introduced in [5]; cf. the 4-node quiver in Table 1b.

Consider the theory $CSM_\kappa$ in Figure 13a, which has alternating CS-levels $(\kappa_1, \kappa_2, \kappa_3, \kappa_4) = (+\kappa, -\kappa, +\kappa, -\kappa)$ with $\kappa > 1$. Next, the two magnetic quivers $MQ_{A/B}$ are derived in turn.

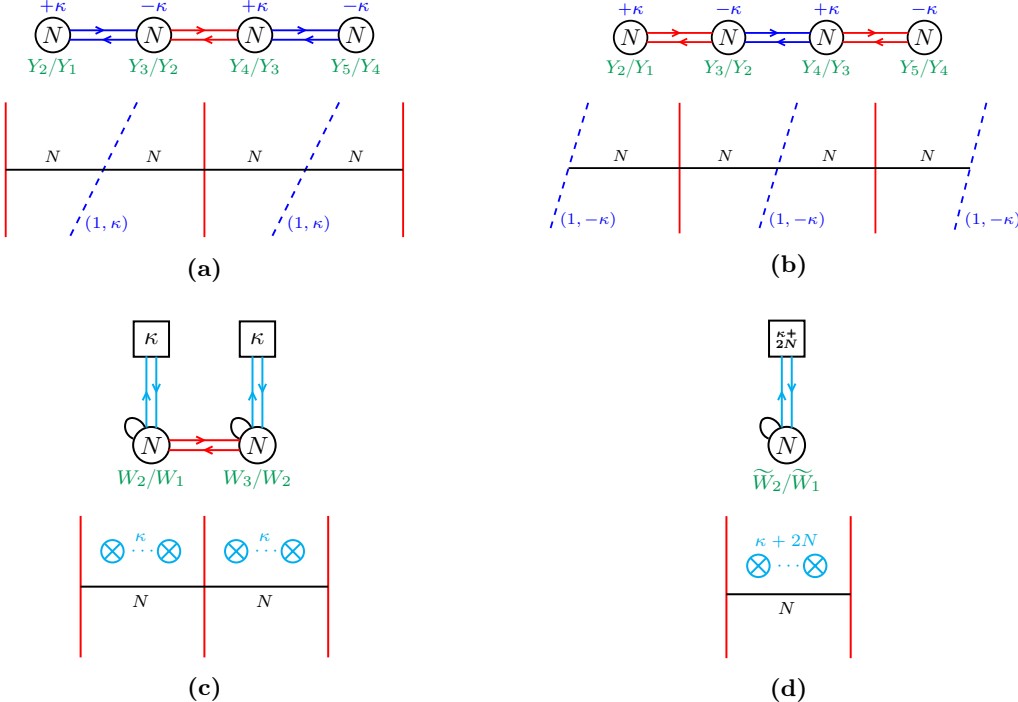

**Figure 13:** (a): Initial $CSM_\kappa$ theory (with $\kappa > 1$). (b): The $CSM'_\kappa$ theory, i.e. the $(\mathcal{T})^\kappa$-dual of $CSM_\kappa$. (c): The magnetic quiver $MQ_A$ for $CSM_\kappa$. (d): The magnetic quiver $MQ_B$ for $CSM_\kappa$. The parameters of the magnetic quivers have different names with respect to those of the CSM theories because they are not related to them by a duality, so no parameters map can be established. The integers over D3 and D5 branes denote their multiplicity.

**A branch $MQ_A$.** The magnetic quiver $MQ_A$ for $CSM_\kappa$ is derived from the brane system by (formally) replacing the $(1, \kappa)$-branes with $\kappa$ D5-branes as in Figure 11; this is shown in Figure 14.

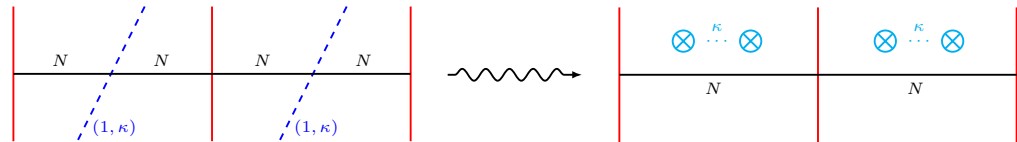

**Figure 14:** The proposed recipe to get $MQ_A$ from $CSM_\kappa$.

The magnetic quiver $MQ_A$ the read off and shown Figure 13c. Requiring $MQ_A$ to be good amounts to

$$\kappa + N \geqslant 2N \quad \Longrightarrow \quad \kappa \geqslant N \,, \tag{3.3}$$

which is now taken as a *good* criterion for the initial $CSM_\kappa$ theory.

**B branch $MQ_B$.** To begin with, dual theory $CSM'_\kappa$ needs to be constructed. In the brane configuration, the $(\mathcal{T})^\kappa$ dualisation of the $CSM_\kappa$ theory into the $CSM'_\kappa$ theory is sketched in Figure 15 and explained in Appendix C.5. The resulting $CSM'_\kappa$ theory is displayed in Figure 13b.

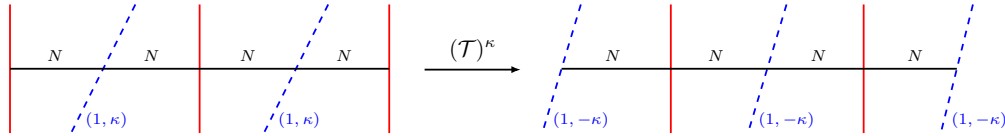

**Figure 15:** The dualisation $CSM_\kappa \to CSM'_\kappa$ at the level of the branes.

Next, the steps for deriving $MQ_B$ from the brane system of $CSM'_\kappa$ are shown in Figure 16, wherein $q \geqslant N$ is assumed. As seen above, this is nothing but the *good* condition of $CSM_\kappa$. The resulting magnetic quiver $MQ_B$ is shown in Figure 13d.

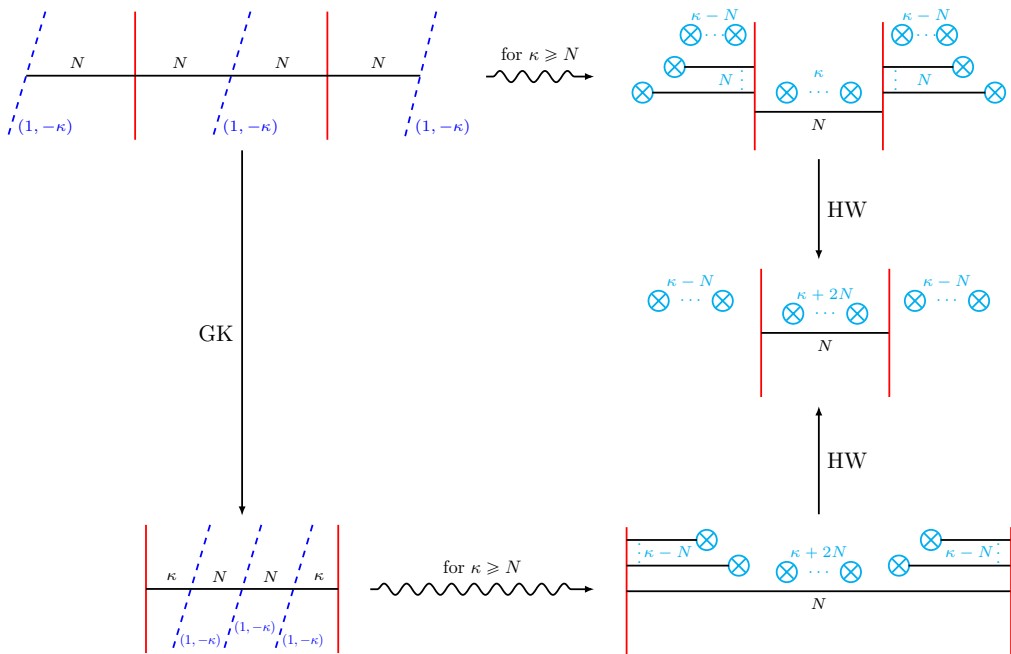

**Figure 16:** Derivation of $MQ_B$ from $CSM'_\kappa$ (assuming the $\kappa \geqslant N$). One use the formal replacement rule (Figure 11) and then take care of brane creation/annihilation or, equivalently, apply GK duality, then formally replace $(1, \pm\kappa)$ branes, and then account for brane creation/annihilation.

Analogously to $MQ_A$, the magnetic quiver $MQ_B$ gives rise to another *good* condition:

$$\kappa + 2N \geqslant 2N \quad \implies \quad \kappa \geqslant 0. \tag{3.4}$$

The reason why condition (3.4) is weaker then (3.3) lies in the fact that, varying $\kappa$, the A branch operators of the $CSM_\kappa$ theory fall below the unitarity bound quicker then those of the B branch. This is reflected in the Coulomb branches of $MQ_{A/B}$ and, ultimately, in the conditions (3.3) and (3.4), among which the strongest has to be taken as the final *good* criterion for the $CSM_\kappa$ theory.

**Global symmetries.** Next consider the global symmetry algebra of $CSM_\kappa$ (Figure 13a):

$$\begin{cases} [\mathfrak{u}(1)_v \oplus \mathfrak{u}(1)_w]_A \oplus [\mathfrak{u}(1)_q]_B, & \kappa > N, \\ [\mathfrak{su}(3)_u]_A \oplus [\mathfrak{u}(1)_q]_B, & \kappa = N. \end{cases} \tag{3.5}$$

Indeed the magnetic quiver $\mathsf{MQ}_A$ in the $\kappa = N$ case has two balanced gauge nodes, which causes the global symmetry enhancement $\mathfrak{u}(1)_v \oplus \mathfrak{u}(1)_w \to \mathfrak{su}(3)_u$ on the A branch of $\text{CSM}_\kappa$. On the other hand, the magnetic quiver $\mathsf{MQ}_B$ gives rise to a non-enhanced $\mathfrak{u}(1)_q$ on the B branch of $\text{CSM}_\kappa$.

**Higgsing patterns.** Using Coulomb branches of the magnetic quivers $\mathsf{MQ}_{A/B}$, one can probe the A/B branch Higgsing pattern of $\text{CSM}_\kappa$ by applying the decay and fission algorithm [40, 41] to $\mathsf{MQ}_{A/B}$ and deduce the RG-flow pattern. Alternatively, the A/B branch Higgsing of a CSM theory can be studied directly in the brane system — maximal branches correspond to D3 segments between distinct 5-branes (see the related discussion in the previous section). For consistency, the two approaches need to produce the same Higgsing pattern for the CSM theories.

In what follows, the arguments are provided for the generic $N$ and $\kappa$ (within the class defined in Figure 13a). In the next section, specific $(N, \kappa)$ values are chosen and analysed: including the magnetic quivers and the specific brane configurations.

**B branch.** The B branch Higgsing of the $\text{CSM}_\kappa$ theory is depicted in Figure 17, showing both the brane systems and the associated quivers. For convenience, the Higgsing is performed on the GK dual of the starting CSM theory. The steps are as follows:

1) Move the two $(1, \kappa)$ 5-branes to the outside via GK duality (see Appendix D, in particular Figure 50).

2) Reconnect four D3 segments to form a single D3-brane that is stretched between the two $(1, \kappa)$ 5-branes. Move this D3 off to infinity along the $(1, \kappa)$ 5-branes — i.e. moving along the B branch. At the level of field theory, this corresponds to the operator[14] $\mathfrak{M}^{(++++)}(\widetilde{Q}_{1,2}\widetilde{Q}_{2,3}\widetilde{Q}_{3,4})^\kappa$ (or equivalently $\mathfrak{M}^{(----)}(Q_{1,2}Q_{2,3}Q_{3,4})^\kappa$) acquiring a VEV. Depending on the choice of $\kappa$ and $N$ one gets a certain value for the charge $p = R[\mathfrak{M}^{(++++)}(\widetilde{Q}_{1,2}\widetilde{Q}_{2,3}\widetilde{Q}_{3,4})^\kappa]$, and the geometric transition is then $A_{p-1}$.

3,4) Iterate the previous step until no more D3 branes are left between the NS5 branes. (This terminates because $\kappa \geqslant N$.) The final theory is completely trivial. Indeed, in the $\kappa = N$ case all the gauge ranks are zero, while in the $\kappa > N$ case what is left are just two copies of SYM with a CS-level, which are indeed trivial (see the related discussion in [18] for more details).

**A branch.** The A branch Higgsing of the $\text{CSM}_\kappa$ theory, depicted in Figure 18, is highly $\kappa$ and $N$ dependent. In the following the three possible cases are discussed:

(i) $\kappa = N$: one expects $(N^2 + 1)$ many Higgsings, the first one being enhanced to $a_2$ (since the starting configuration has both plateaux balanced) and the others being either of the Kleinian type (i.e. $A_\#$, where the number $\#$ depends on $N$ and $\kappa$) or enhanced to $a_1$ (when one of the two plateaux becomes balanced). The operator taking a VEV in the starting $a_2$ Higgsing is taken from the pool of operators at order $xt^2$ that constitute the $\mathfrak{su}(3)$ moment map

$$\mathfrak{M}^{(++++)}(\widetilde{Q}_{1,2}\widetilde{Q}_{3,4})^\kappa, \quad \mathfrak{M}^{(----)}(Q_{1,2}Q_{3,4})^\kappa, \tag{3.6a}$$

$$\mathfrak{M}^{(++00)}(\widetilde{Q}_{1,2})^\kappa, \quad \mathfrak{M}^{(--00)}(Q_{1,2})^\kappa, \tag{3.6b}$$

$$\mathfrak{M}^{(00++)}(\widetilde{Q}_{3,4})^\kappa, \quad \mathfrak{M}^{(00--)}(Q_{3,4})^\kappa, \tag{3.6c}$$

---

[14]Chiral multiplets between node $(i)$ and $(i + 1)$ in the quiver are denoted as $Q_{i,i+1}$ or $\widetilde{Q}_{i,i+1}$. Likewise, $\mathfrak{M}^{(++++)}$ denotes a monopole operator with equal magnetic flux $(1, 0, \ldots, 0)$ in all U($N$) gauge nodes.

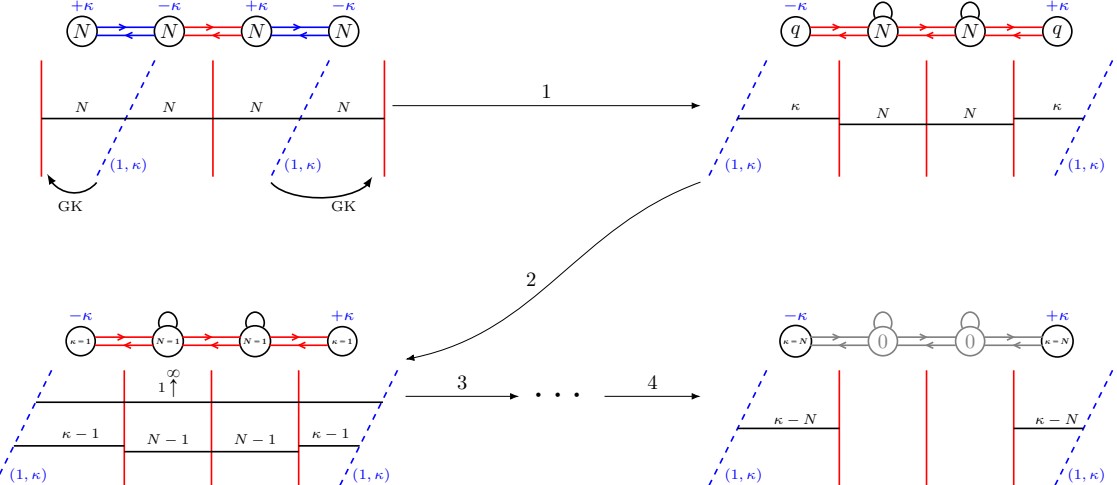

**Figure 17:** B branch Higgsing of the CSM theory in Figure 13a.

$$\text{Tr}\left[Q_{1,2}\widetilde{Q}_{1,2}\right], \quad \text{Tr}\left[Q_{3,4}\widetilde{Q}_{3,4}\right]. \tag{3.6d}$$

Here, the first three lines are the 3 positive roots (along with the 3 negative roots), while the last line displays the 2 Cartan elements.

In the case of an $a_1$ Higgsing, there are 3 operators at order $xt^2$ forming the adjoint representation of $\mathfrak{su}(2)$:

$$\mathfrak{M}^{(++00)}(\widetilde{Q}_{1,2})^{\kappa}, \quad \mathfrak{M}^{(--00)}(Q_{1,2})^{\kappa}, \quad \text{Tr}\left[Q_{1,2}\widetilde{Q}_{1,2}\right], \tag{3.7}$$

or analogously

$$\mathfrak{M}^{(00++)}(\widetilde{Q}_{3,4})^{\kappa}, \quad \mathfrak{M}^{(00--)}(Q_{3,4})^{\kappa}, \quad \text{Tr}\left[Q_{3,4}\widetilde{Q}_{3,4}\right]. \tag{3.8}$$

These operator taking a VEV is then chosen from the $\mathfrak{su}(2)$ moment map.

If the transition is a Kleinian transition, the operator acquiring a VEV is either $\mathfrak{M}^{(++00)}(\widetilde{Q}_{1,2})^{\kappa}$ (or $\mathfrak{M}^{(--00)}(Q_{1,2})^{\kappa}$) or $\mathfrak{M}^{(00++)}(\widetilde{Q}_{3,4})^{\kappa}$ (or $\mathfrak{M}^{(00--)}(Q_{3,4})^{\kappa}$).

(ii) $N < \kappa \leqslant 2N$: one can expect $(N+1)^2$ many Higgsings, some of the Kleinian type (i.e. $A_\#$) and some enhanced to $a_1$ when one of the two plateaux becomes balanced. The operators taking a VEV are $\mathfrak{M}^{(++00)}(\widetilde{Q}_{1,2})^{\kappa}$ (or $\mathfrak{M}^{(--00)}(Q_{1,2})^{\kappa}$) or $\mathfrak{M}^{(00++)}(\widetilde{Q}_{3,4})^{\kappa}$ (or $\mathfrak{M}^{(00--)}(Q_{3,4})^{\kappa}$).

(iii) $\kappa > 2N$: one can expect $(N+1)^2$ many Higgsings, all of the Kleinian type (i.e. $A_\#$) since none of the two plateaux can be balanced. The operators taking a VEV are $\mathfrak{M}^{(++00)}(\widetilde{Q}_{1,2})^{\kappa}$ (or $\mathfrak{M}^{(--00)}(Q_{1,2})^{\kappa}$) or $\mathfrak{M}^{(00++)}(\widetilde{Q}_{3,4})^{\kappa}$ (or $\mathfrak{M}^{(00--)}(Q_{3,4})^{\kappa}$).

This general discussion is illustrated and validated in Appendix B.1. Additional examples are provided in Appendices B.2, B.3, and B.4, further supporting the analysis and confirming that the A and B branch limits of the CSM theory index match the Coulomb branch Hilbert series of the corresponding magnetic quivers.

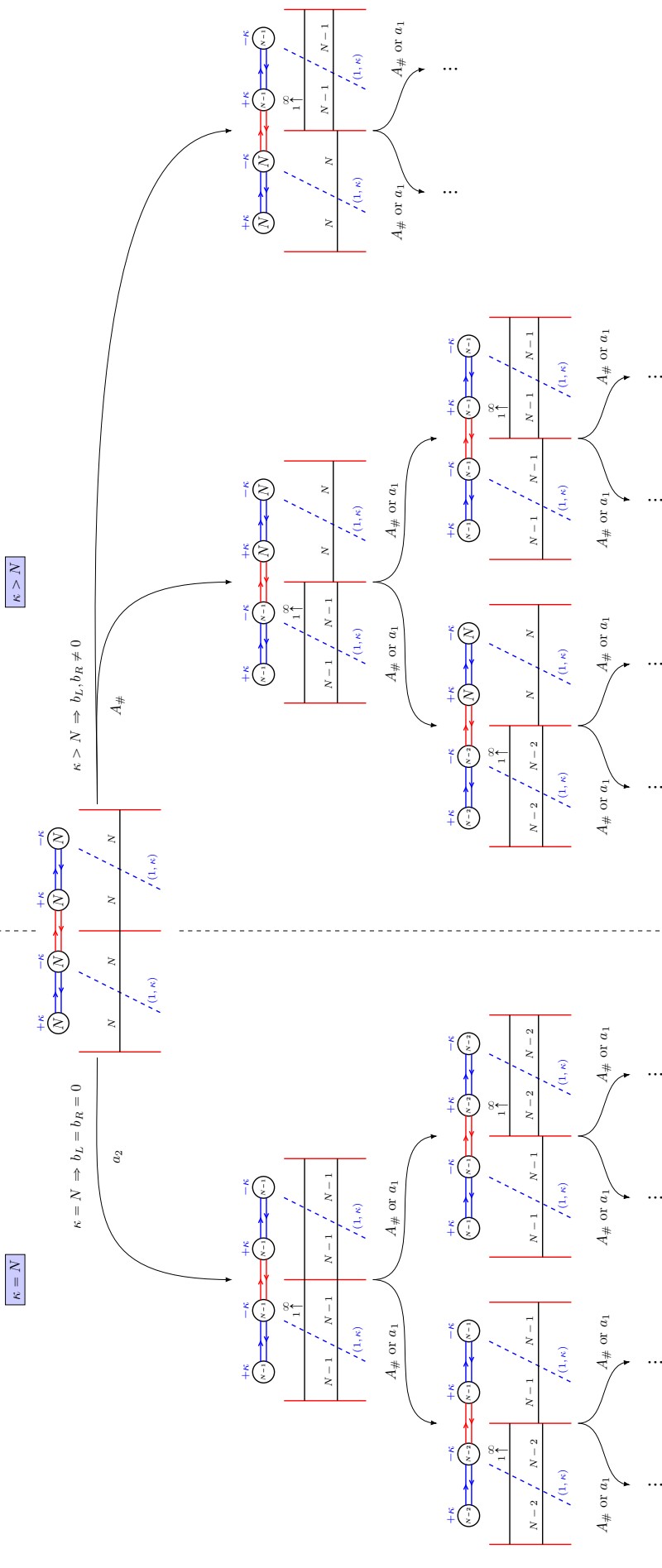

**Figure 18:** A branch Higgsing of the CSM theory in Figure 13a. The leftmost part of the figure depicts the case $q = N$, in which the two blue plateaux at the beginning (of balancing $b_L$ and $b_R$ respectively) are both balanced, namely $b_L = b_R = 0$, which leads to an enhanced $\alpha_2$ transition. The following $N^2$ transitions are in general of the $A_\#$ type, but can enhance to $\alpha_1$ if one of the two blue plateaux becomes balanced. On the other hand, the rightmost part of the figure depicts the case $q > N$, in which the two blue plateaux at the beginning are not balanced, meaning that the first Higgsing is of the $A_\#$ type. The following transitions are in general of the $A_\#$ type, but can enhance to $\alpha_1$ if one of the two blue plateaux becomes balanced. The latter, however, never happens if $q > 2N$.

## 3.2 Generalisations

The magnetic quiver proposal allows one to access the maximal branches of the CSM quiver's moduli even for $|\kappa| > 1$. The immediate advantages, for instance, include: (i) A straightforward access to a definition of a *good* CSM theory, based on $\mathsf{MQ}_{A/B}$. (ii) A unified and systematic approach to the maximal branches that does just result in a A/B branch limit of an index, but rather gives an entire arsenal of tools to study the branch geometry and quantum relations. (iii) An algorithmic procedure to trace out the RG-flow Hasse diagram.

In order to discuss such feature more systematically for $\text{CSM}_{\kappa>1}$ theories, two basic families are introduced below. These are the $\kappa > 1$ versions of the two families discussed in Section 2.3 for $|\kappa| = 1$, but with a choice of ranks that allows a convenient analysis.

### 3.2.1 Family I

The first $\text{CSM}_{\kappa>1}$ quiver family is shown in Figure 19a. It consists of a plateau of $n + 1$ nodes of rank $2N$, where only the external nodes carry a CS-level $\pm\kappa$, connected by (blue) hypermultiplets. Adjacent to the plateau there are two extra nodes of rank $N$ connected to the plateau via (red) twisted hypermultiplets.

The magnetic quiver $\mathsf{MQ}_A$ is shown in Figure 19b and is good provided that

$$\kappa\, n \geqslant 2N \,. \tag{3.9}$$

On the other hand, for the magnetic quiver $\mathsf{MQ}_B$ one has to consider three distinct cases: (i) for $\kappa \geqslant N$, (ii) for $\kappa < N = l\kappa + l_0$ (with $l, l_0 \in \mathbb{N}$), and (iii) for $\kappa < N = l\kappa$ (with $l \in \mathbb{N}$) the magnetic quiver is shown in Figure 19c, 19d, and 19e, respectively.

### 3.2.2 Family II

The second $\text{CSM}_{\kappa>1}$ quiver family is shown in Figure 20a: a CSM theory with a set of $n+1$ nodes, where only the external nodes carry a CS-level $\pm\kappa$, connected by (red) twisted hypermultiplets. On their left and on their right there are two extra nodes with CS-level $\pm\kappa$ and of rank $M_L$ and $M_R$ respectively, connected to the rest of the theory via (blue) hypermultiplets. The second $\text{CSM}_{\kappa>1}$ quiver family, shown in Figure 20a, consists of a chain of (red) twisted hypermultiplets connecting $n+1$ nodes, flanked by two sets of (blue) hypermultiplets that link plateaux of $M_L + 1$ and $M_R + 1$ nodes, respectively, with CS-levels assigned only to the two external nodes. Its magnetic quiver $\mathsf{MQ}_A$ can be derived in a uniform manner and is shown in Figure 20b. It is good provided that

$$M_L \geqslant N \,, \quad M_R \geqslant N \,. \tag{3.10}$$

Again, the $\mathsf{MQ}_B$ derivation requires a case distinction, which is detailed in Figures 20c–20e.

Given $\mathsf{MQ}_{A/B}$, one can work out the global symmetries and Higgsing transitions of these theories in the exact same way as in Section 2.3. The only difference is that here, instead of the $\mathcal{T}^T$-dual, the the magnetic quivers $\mathsf{MQ}_A$ and $\mathsf{MQ}_B$ are used to describe the A/B branch of the starting theory. Moreover, in order to account for the CS-level being greater than 1, the plateau balancing (2.21) has to be redefined as follows:

$$b = |\kappa| n + M_L + M_R - 2N \,. \tag{3.11}$$

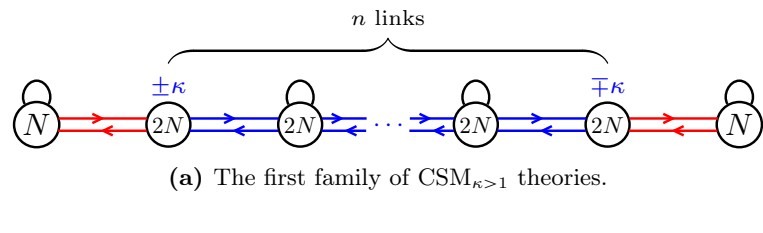

**(a)** The first family of $CSM_{\kappa>1}$ theories.

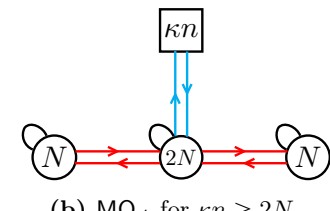

**(b)** $MQ_A$ for $\kappa n \geqslant 2N$.

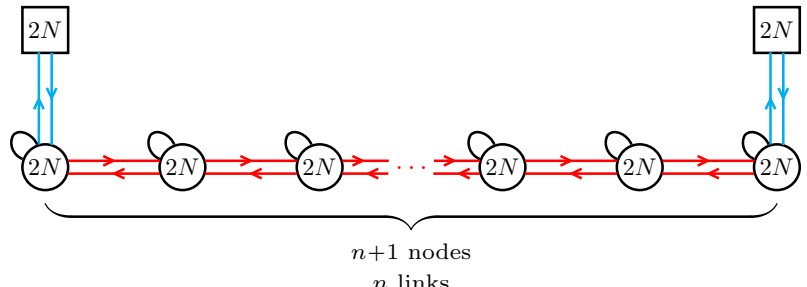

**(c)** $MQ_B$ for case $\kappa \geqslant N$ (which needs to be combined with $\kappa n \geqslant 2N$).

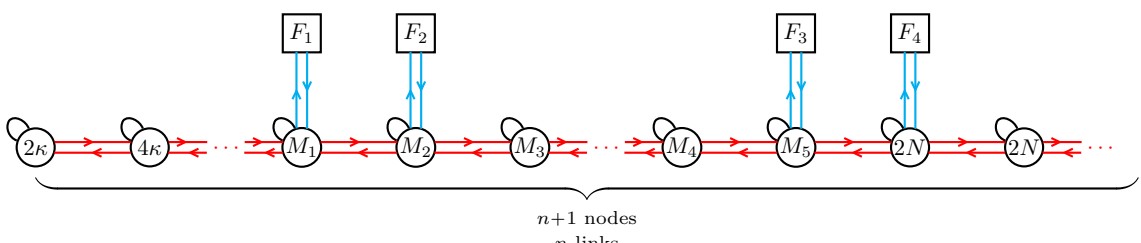

**(d)** $MQ_B$ for case $\kappa < N = l\kappa + l_0$ (with $l, l_0 \in \mathbb{N}$) (and supplemented by $\kappa n \geqslant 2N$).

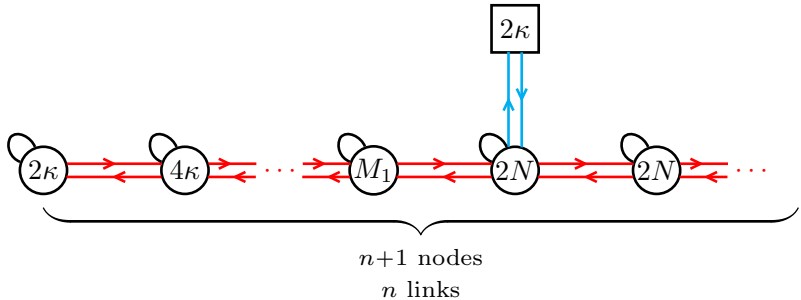

**(e)** $MQ_B$ for case $\kappa < N = l\kappa$ (with $l \in \mathbb{N}$) (and supplemented by $\kappa n \geqslant 2N$).

**Figure 19:** The first family of $CSM_{\kappa>1}$ theories and its magnetic quivers $MQ_{A/B}$. (a): The linear $CSM_{\kappa>1}$ quiver. (b): The magnetic quiver $MQ_A$, imposing the condition $\kappa n \geqslant 2N$. (c): $MQ_B$ for $\kappa \geqslant N$. (d): $MQ_B$ for $\kappa < N = l\kappa + l_0$ with $l, l_0 \in \mathbb{N}$. Only its left part is represented, as the quiver is symmetrical. In particular, introducing a parameter $s \in \mathbb{N}$ and setting $\kappa(s - l) \geqslant l_0$ to define where the leftmost flavour nodes are located, one has $M_1 = 2(s - 1)\kappa$, $M_2 = N + s\kappa$, $M_3 = N + (s + 1)\kappa$, $M_4 = N + (l - 1)\kappa$, $M_5 = N + l\kappa$ and $F_1 = s\kappa - N$, $F_2 = \kappa - (s\kappa - N)$, $F_3 = \kappa - l_0$, $F_4 = l_0$. (e): $MQ_B$ for $\kappa < N = l\kappa$ with $l \in \mathbb{N}$. Only its left part is represented, as the quiver is symmetrical. In particular, $M_1 = 2(l - 1)\kappa$.

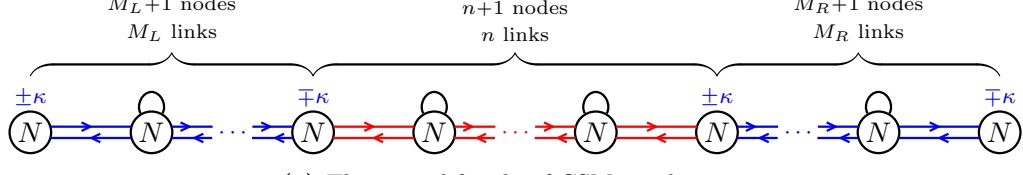

**(a)** The second family of $\mathrm{CSM}_{\kappa>1}$ theories.

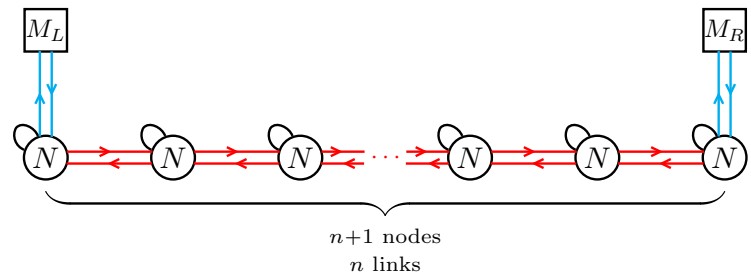

**(b)** $\mathsf{MQ}_A$ for $M_L \geqslant N$ and $M_R \geqslant N$.

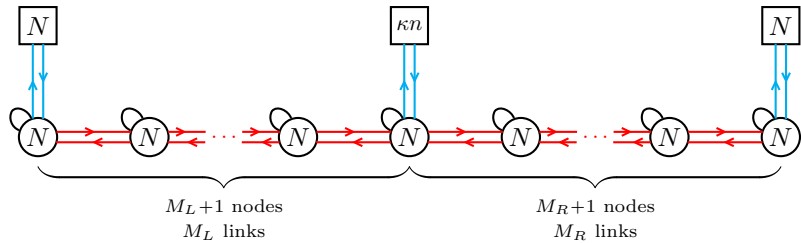

**(c)** $\mathsf{MQ}_B$ for case $\kappa \geqslant N$ (which needs to be combined with $M_{L/R} \geqslant N$).

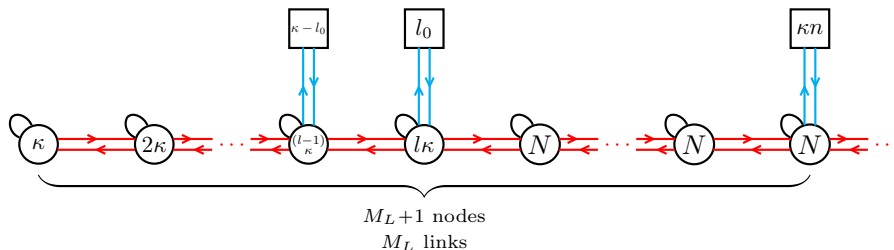

**(d)** $\mathsf{MQ}_B$ for case $\kappa < N = l\kappa + l_0 \leqslant M_L \kappa$ (with $l, l_0 \in \mathbb{N}$) (supplemented by $M_{L/R} \geqslant N$).

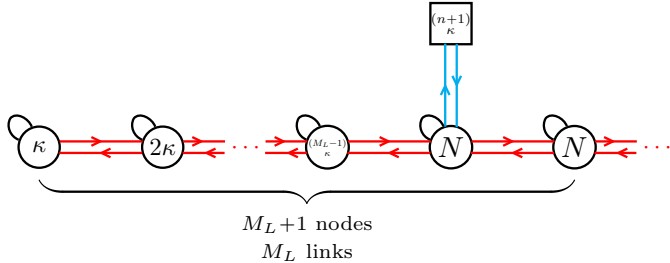

**(e)** $\mathsf{MQ}_B$ for case $\kappa < N = l\kappa = M_L \kappa$ (with $l \in \mathbb{N}$) (supplemented by $M_{L/R} \geqslant N$).

**Figure 20:** The second family of $\mathrm{CSM}_{\kappa>1}$ quivers and their magnetic quivers $\mathsf{MQ}_{A/B}$. (a): The linear $\mathrm{CSM}_{\kappa>1}$ quiver. (b): The magnetic quiver $\mathsf{MQ}_A$, imposing the conditions $M_{L/R} \geqslant N$. (c): $\mathsf{MQ}_B$ for $\kappa \geqslant N$. (d): $\mathsf{MQ}_B$ for $\kappa < N = l\kappa + l_0 \leqslant \kappa M_L$ with $l, l_0 \in \mathbb{N}$. Only the left part is shown, due to symmetry. (e): $\mathsf{MQ}_B$ for $\kappa < N = l\kappa = \kappa M_L$ with $l \in \mathbb{N}$. Again, only the left part is shown.

# 4 Circular $\mathcal{N}$=4 Chern–Simons Matter theories

So far, linear $\mathcal{N} = 4$ CSM quiver theories were considered that are realised on D3-NS5-$(1,\kappa)$ brane systems. A rather natural generalisation is to put the brane system on a circle which results in circular CSM quivers [13, 18, 21] whose $\mathcal{N} = 4$ supersymmetry has been analysed in [5, 57]. Besides the Type IIB realisation (see also Appendix A.2), circular CSM theories are related to M2 branes on complex 4 dimensional orbifold singularities [6–8, 13, 49]: for a system composed of $p$ NS5s, $q$ $(1,\kappa)$ 5-branes, the relevant M-theory singularity is $(\mathbb{C}^2/\mathbb{Z}_p \times \mathbb{C}^2/\mathbb{Z}_q)/\mathbb{Z}_\kappa$, see [13].

The maximal branches of such circular CSM theories were studied, for example, in [15, 47, 49, 58]. As for linear brane systems, at $\kappa = 1$ an $\mathrm{SL}(2,\mathbb{Z})$ transformation yields a D3-NS5-D5 system with a 3d $\mathcal{N} = 4$ Lagrangian quiver, see e.g. [24, 25]. The computation can be performed via the dualization algorithm, whose extension to circular quivers is currently under development [59], allowing for the mapping of the fugacities across the duality.

The purpose of this section is to show that the magnetic quiver proposal is readily applied to these setups and provides straightforward access to the maximal branch. Thus, it is a proof of concept, rather than an exhaustive analysis.

**Example 1: circular $\mathbf{U(N)_{+\kappa} \times U(N)_{-\kappa} \times \cdots \times U(N)_{+\kappa} \times U(N)_{-\kappa}}$.** To begin with, consider the brane systems in Figure 21a of $N$ D3 branes intersected by $n$ NS5s and $n$ $(1,\kappa)$ 5-branes that are placed alternately. Hence, all $\ell = 2n$ nodes in the CSM quiver have non-trivial CS-levels $\pm\kappa$. For concreteness, $\kappa > 0$.

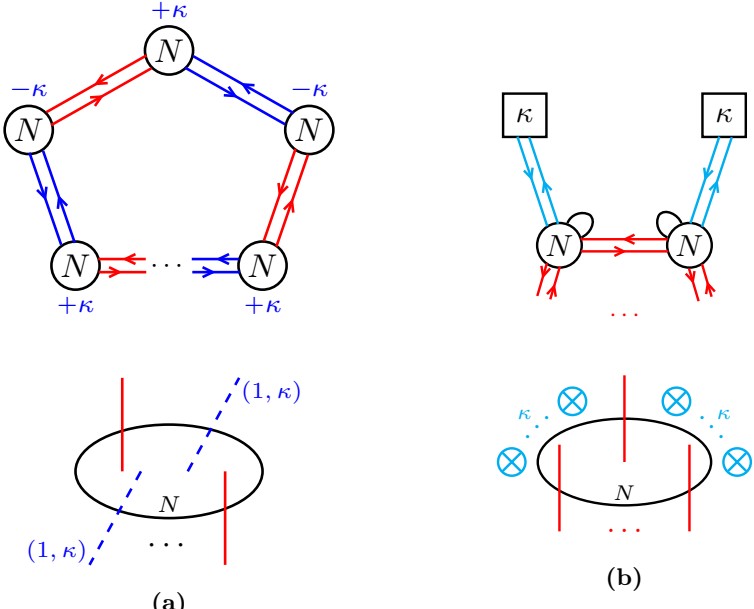

**Figure 21:** (a) The circular CSM quiver living in the world-volume of $N$ D3 intersected by $n$ NS5s and $n$ $(1,\kappa)$ 5-branes, which alternate. (b) The magnetic quiver $\mathsf{MQ}_{A/B}$, which are identical here.

Next, one readily derives the magnetic quivers $\mathsf{MQ}_{A/B}$, shown in Figure 21b. Due to the symmetry of the brane system, the A and B branch are isomorphic, and so are $\mathsf{MQ}_{A/B}$. The quivers $\mathsf{MQ}_{A/B}$ are known as Kronheimer-Nakajima quivers [60], whose Higgs and Coulomb branches have been studied extensively in the past[15]. The Coulomb branch of $\mathsf{MQ}_{A/B}$ is the moduli space of $N$ $\mathrm{SU}(n)$ instantons on $\mathbb{C}^2/\mathbb{Z}_{n\cdot\kappa}$ with framing

---

[15]The reader is referred to [61, 62], the review [63], and references therein.

$(0^{\kappa-1}, 1, 0^{\kappa-1}, 1, \ldots, 0^{\kappa-1}, 1)$. This agrees with [49, eq. (5.14)] where one branch geometry was extracted via Hilbert series.

As a remark, the A and B branch are isomorphic because of the symmetric arrangement of 5-branes. It is straightforward to show that, firstly, $\mathsf{MQ}_{A/B}$ depend on the brane arrangement up to GK-duality[16]; For instances, for fixed number of D3s, starting with a different arrangement of the 5-branes leads to generically different theories and the $\mathsf{MQ}_{A/B}$ reflect this properly. Secondly, that $\mathsf{MQ}_A$ and $\mathsf{MQ}_B$ can be distinct. See, for instance, Appendices B.5 and B.6. Also, for $\kappa = 1$, $\mathsf{MQ}_A$ is the $\mathcal{T}^T$-dual of the CSM theory, and $\mathsf{MQ}_B$ is the $\mathcal{S}$-dual of $\mathsf{MQ}_A$. Hence, the theory is self-mirror.

**Example 2: circular $\mathbf{U(N)_{+\kappa} \times U(N)^{\otimes \ell - 2} \times U(N)_{-\kappa}}$**   Next, consider the brane system in Figure 22 of $N$ D3 brane intersected by one NS5 and $\ell - 1$ many $(1, \kappa)$ 5-branes. As a result, the circular CSM quiver has two CS and $\ell - 2$ non-CS nodes. Again, $\kappa > 0$.

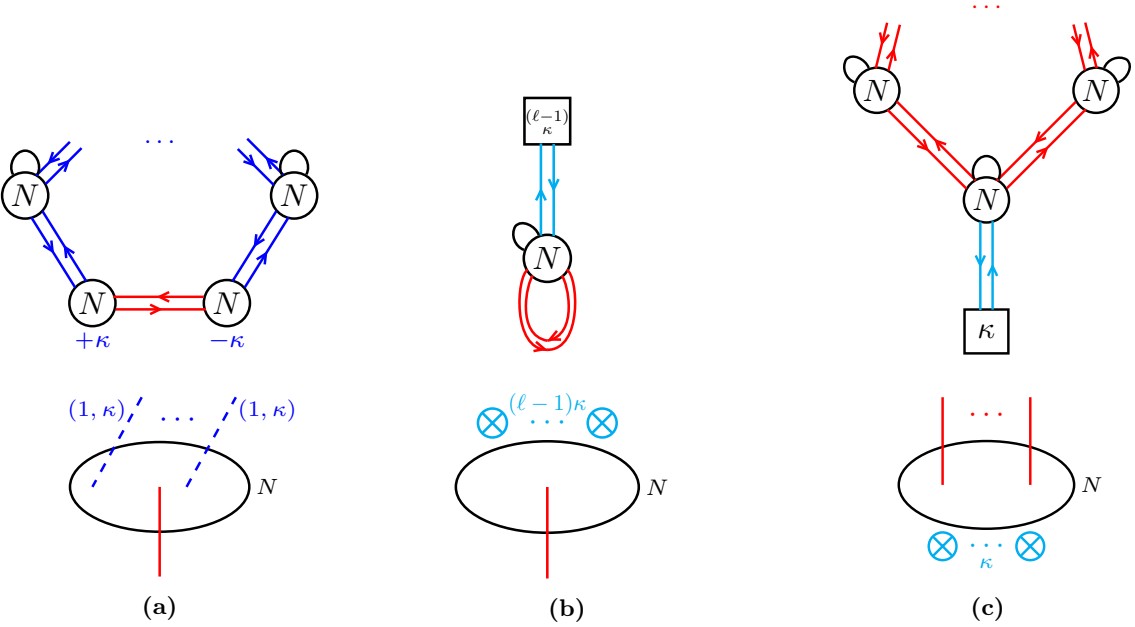

**Figure 22:** (a) The circular CSM theory living in the world-volume of $N$ D3s intersected by one NS5 and $(\ell - 1)$ $(1, \kappa)$ 5-branes. (b) The magnetic quiver $\mathsf{MQ}_A$. (c) The magnetic quiver $\mathsf{MQ}_B$, which has $(\ell - 1)$ many U($N$) gauge nodes.

Due to the apparent asymmetry of the brane system, the magnetic quivers are distinct. One recognises $\mathsf{MQ}_A$ as the A-type ADHM quiver, whose Coulomb branch is $\mathrm{Sym}^N(\mathbb{C}^2/\mathbb{Z}_{(\ell-1)\kappa})$ [61, 62]. This also agrees with the limit of the index in [58, eq. (7.27)]. On the other hand, $\mathsf{MQ}_B$ is again a Kronheimer-Nakajima quivers, whose Coulomb branches describes $\kappa$ SU($N$) instantons on $\mathbb{C}^2/\mathbb{Z}_\kappa$ with framing $(\kappa, 0^{\ell-2})$. This agrees with the Hilbert series derivation of [49, eq. (5.9)]. See Appendix B.7 for examples. Note that for $\kappa = 1$, $\mathsf{MQ}_A$ is the $\mathcal{T}^T$-dual of the CSM theory, while $\mathsf{MQ}_B$ is the mirror of the $\mathcal{T}^T$-dual.

**Symmetries, Hasse diagrams, and Higgsing.**   As the class of quivers that appears as $\mathsf{MQ}_{A/B}$ for circular $\mathcal{N} = 4$ CSM theories is within the scope of developed techniques, one can repeat the analysis of symmetries and the minimal transitions along A and B branch as for any linear quiver. One can either choose to work directly on the brane system or, equivalently, use quiver algorithms [40, 41] on $\mathsf{MQ}_{A/B}$ to deduce the residual theories after Higgsing as well as the transition geometries.

---

[16]Note that dualities like [64] circular   $\mathrm{U}(N)_\kappa \times \mathrm{U}(N)_{-\kappa} \times \mathrm{U}(N)_\kappa \times \mathrm{U}(N)_{-\kappa} \longleftrightarrow$ circular   $\mathrm{U}(N + |\kappa|)_\kappa \times \mathrm{U}(N)_0 \times \mathrm{U}(N)_{-\kappa} \times \mathrm{U}(N)_0$, come up naturally as the action of the GK duality in the brane system.

# 5 $\mathcal{N} = 3$ Chern–Simons Matter theories

A natural extension of the theories considered so far is realised by brane systems with at least three distinct types of 5-branes, which has $\mathcal{N} = 3$ supersymmetry. Despite the reduction of supersymmetry, the moduli space of vacua is still composed of branches. The maximal branches are again identified in the Type IIB brane systems as motions of D3-segments between distinct types of 5-branes; implying that there are as many maximal branches as there are distinct types of $(p, q)$ 5-branes. In fact, as emphasised in [15] already, the maximal branches are all hyper-Kähler by virtue of the SU(2) R-symmetry.

Here, the magnetic quiver proposal is extended to the maximal branches of $\mathcal{N} = 3$ CSM. This constitutes a non-trivial consistency check of the proposal beyond the realm of $\mathcal{N} = 4$ theories. Prior, these branches have only been scarcely analysed [15, 49].

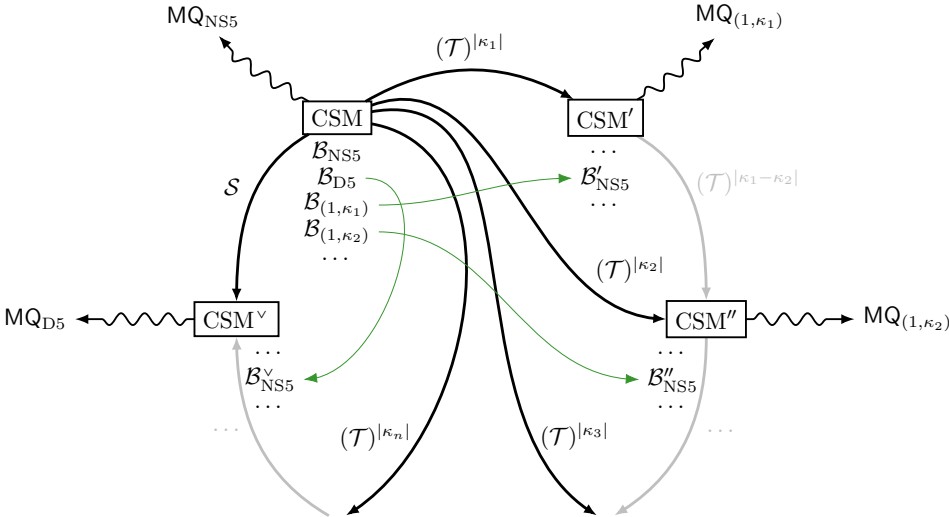

**Figure 23:** The strategy for the magnetic quiver derivation for the maximal branches $\mathcal{B}$ of a $\mathcal{N} = 3$ CSM quiver. To derive $\mathsf{MQ}_{(1,\kappa)}$ (resp. $\mathsf{MQ}_{\mathrm{D5}}$) for $\mathcal{B}_{(1,\kappa)}$ (resp. $\mathcal{B}_{\mathrm{D5}}$), one utilises a $(\mathcal{T})^{|\kappa|} \in \mathrm{SL}(2, \mathbb{Z})$ (resp. $\mathcal{S}$) duality that swaps $\mathcal{B}_{(1,\kappa)}$ (resp. $\mathcal{B}_{\mathrm{D5}}$) for $\mathcal{B}'_{\mathrm{NS5}}$ (resp. $\mathcal{B}^{\vee}_{\mathrm{NS5}}$) of the dual theory.

Before going into the specifics, the logic of the magnetic quiver derivation is spelled out, cf. Figure 23. Consider a $\mathcal{N} = 3$ CSM theory coming from a brane system with NS5s, D5s, and some $(1, \kappa_i)$ 5-branes. Generalising the proposal of Section 3, the magnetic quivers are derived as follows:

- Branch $\mathcal{B}_{\mathrm{NS5}}$ between NS5s: Up to boundary conditions (see Figure 11), the D3 segments between the NS5 branes give rise to the magnetic vector multiplets, while adjacent D3 segments induce bifundamental magnetic hypermultiplets. The other types of 5-brane contribute with the D5-brane charge in the respective NS5 interval.

- Branch $\mathcal{B}_{\mathrm{D5}}$ between D5s: One performs an $\mathcal{S} \in \mathrm{SL}(2, \mathbb{Z})$ transformation on the starting CSM system, which maps $(p, q) \to (q, -p)$. From this dual configuration $\mathrm{CSM}^{\vee}$ one reads the magnetic quiver between the NS5 branes; as this captures the D3 segments between D5s in the original CSM system, see Figure 11.

- Branch $\mathcal{B}_{(1,\kappa_i)}$ between $(1, \kappa_i)$ 5-branes (fixed $i$): One performs a $(\mathcal{T})^{|\kappa_i|} \in \mathrm{SL}(2, \mathbb{Z})$ transformation on the starting brane system. This swaps NS5 and $(1, \kappa_i)$, leaves the D5s invariant, and maps $(1, \kappa_j) \to (1, \kappa_j - \kappa_i)$. The magnetic quiver for this branch is read off from the D3 segments between NS5s in the dual brane system.

In fact, this formalism generalises to theories involving general $(p, q)$ 5-branes, as discussed in Section 5.2. This opens the door to analyse non-Lagrangian CSM quivers.

## 5.1 $\mathcal{N} = 3$ CSM theories with NS5, D5, and $(1, \kappa_i)$ 5-branes

Here, $\mathcal{N} = 3$ CSM theories with fundamental flavours are considered.

### 5.1.1 Abelian

As a proof of principle, the magnetic quiver proposal for the maximal branches of $\mathcal{N} = 3$ CSM theories is illustrated in abelian cases and validated against known results.

**First example.** Concretely, consider the abelian $\mathcal{N} = 3$ theory $T[3]$, realised as the world-volume theory on a single D3 branes intersecting two NS5, two D5, and two $(1, \kappa)$ 5-branes placed alternately in Type IIB, see Figure 24. The maximal branches have been

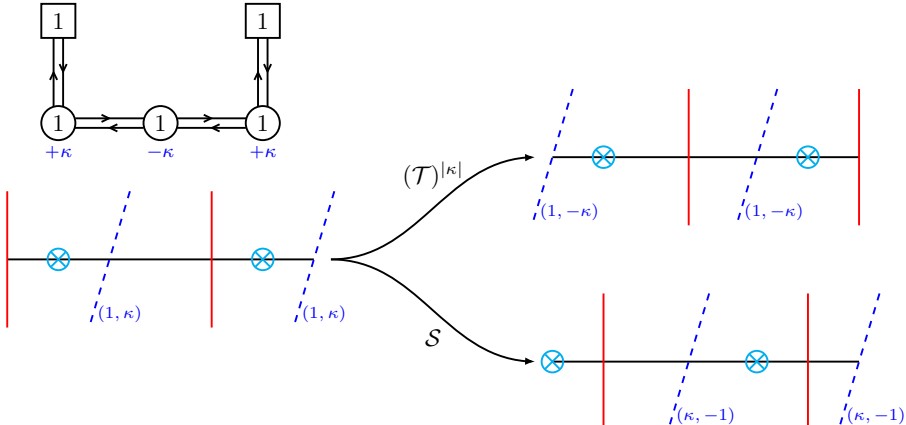

**Figure 24:** The $\mathcal{N} = 3$ CSM quiver theory containing fundamental flavours due to the inclusion of D5 branes. The SL$(2, \mathbb{Z})$ transformations $\mathcal{T}^{|\kappa|}$ and $\mathcal{S}$ map the original brane system into the dual brane systems that allow to derive the magnetic quiver for each maximal branch. In the quiver diagram, the colour coding that has been used so far for the fields is not employed any-more as the setup is now $\mathcal{N} = 3$ and hence there is no axial symmetry to refine the hypermultiplets.

identified in [15, eq. (4.20)] through a different method. Here, the maximal branches are directly and independently analysed via the explicit construction of the magnetic quivers. Step-by-step, one finds:

- Branch $\mathcal{B}_{\text{NS5}}$ — D3 segments between NS5s: From the brane system, one derives

$$\mathsf{MQ}_{\text{NS5}} : \quad \begin{array}{c} \square \ |\kappa| + 2 \\ | \\ \bigcirc \ 1 \end{array} \quad \text{with } \mathcal{C}(\mathsf{MQ}_{\text{NS5}}) \cong A_{|\kappa|+1} \text{ singularity.} \qquad (5.1)$$

  where the number of (magnetic) flavours stems from two sources: (i) the D5-brane charges of the D5 and $(1, \kappa)$ brane within the NS5 interval, and (ii) the D3 segment that connects the right-most NS5 with the right-most D5.

- Branch $\mathcal{B}_{(1,\kappa)}$ — D3 segments between $(1, \kappa)$s: To transition to the dual brane system, one applies $(\mathcal{T})^{|\kappa|}$ to all branes, noting that the D5s remain invariant. Then, by analogous arguments, the magnetic quiver reads

$$\mathsf{MQ}_{(1,\kappa)} : \quad \begin{array}{c} \square \ |\kappa| + 2 \\ | \\ \bigcirc \ 1 \end{array} \qquad (5.2)$$

suggesting that branch $\mathcal{B}_{\mathrm{NS5}}$ and $\mathcal{B}_{(1,\kappa)}$ are isomorphic. This has to hold, as $(\mathcal{T})^{|\kappa|}$ duality exchanges NS5 and $(1,\kappa)$ branes and the brane system in Figure 24 is symmetric with respect to those to 5-branes.

- Branch $\mathcal{B}_{\mathrm{D5}}$ — D3 segments between D5s: Here, the transition to an dual brane system is achieved via the $\mathcal{S}$ transformations. This swaps D5 $\leftrightarrow$ NS5s, and replaces $(1,\kappa)$ by $(\kappa,-1)$ 5-branes. Therefore, from this dual brane system, the magnetic quiver is read to be

$$\mathsf{MQ}_{\mathrm{D5}} : \quad \begin{array}{c} \square\, 4 \\ | \\ \bigcirc\, 1 \end{array} \,. \tag{5.3}$$

Consequently, all maximal branches are correctly and efficiently reproduced.

**Second example.** Consider the $T[4]$ theory of [25, Sec. 5.1], realised via a single D3 intersected by two NS5s, three D5s, four $(1,\kappa_1)$, and two $(1,\kappa_2)$ 5-branes (with $\kappa_1 \neq \kappa_2$, $\kappa_1\kappa_2 \neq 0$), see Figure 25. For each of the four maximal branches, one derives the magnetic

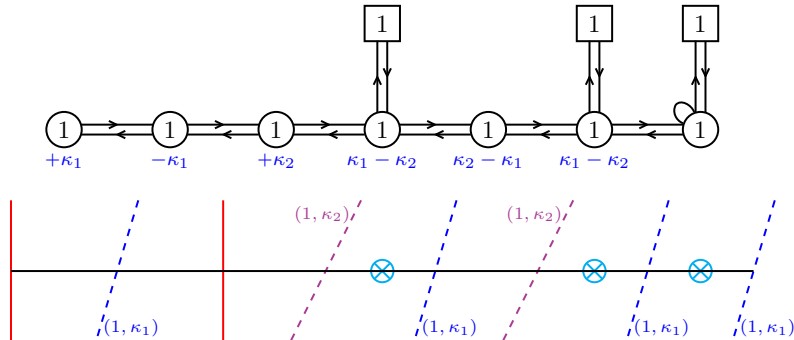

**Figure 25:** The $T[4]$ CSM theory and the associate brane system.

quiver straightforwardly (assuming $\kappa_2 > \kappa_1 > 0$ for concreteness):

- Branch $\mathcal{B}_{\mathrm{NS5}}$ — D3 segments between NS5s: From the CSM brane system, one finds:

$$\mathsf{MQ}_{\mathrm{NS5}} : \quad \begin{array}{c} \square\, \kappa_1 + 1 \\ | \\ \bigcirc\, 1 \end{array} \tag{5.4}$$

where the number of flavours is given by the D5-brane charge of the $(1,\kappa_1)$ brane in between the NS5s plus the extra single D3 segment on the right-hand-side of the second NS-brane. This independent derivation agrees with [15, eq. (5.9)], where this was derived via a different method.

- Branch $\mathcal{B}_{(1,\kappa_2)}$ — D3 segments between $(1,\kappa_2)$s: the useful SL$(2,\mathbb{Z})$ transformation is found by swapping NS5 $\leftrightarrow (1,\kappa_2)$. From the dual brane system, one finds:

$$\mathsf{MQ}_{(1,\kappa_2)} : \quad \begin{array}{c} \square\, |\kappa_1 - \kappa_2| + 3 \\ | \\ \bigcirc\, 1 \end{array} \tag{5.5}$$

to be explicit, the number of flavours stems from four contributions: (i) the D5-brane charge of the dualised $(1,\kappa_1)$ brane, which becomes a $(1,\kappa_1 - \kappa_2)$. (ii) the single D5 in between the $(1,\kappa_2)$ branes, noting that D5 is invariant. (iii) the single D3 brane on the left-hand side of the $(1,\kappa_2)$ interval. (iv) the D3 on the right-hand side of the $(1,\kappa_2)$ interval. Again, this correctly reproduces earlier results [15, eq. (5.13)].

- Branch $\mathcal{B}_{\mathrm{D5}}$ — D3 segments between D5s: here, one applies the $\mathcal{S}$ transformation to read off the magnetic quiver:

$$\mathsf{MQ}_{\mathrm{D5}} : \qquad \overset{\overset{3}{\square} \quad \overset{2}{\square}}{\underset{1 \quad 1}{\bigcirc - \bigcirc}} \qquad \text{with} \qquad \begin{array}{c} \bullet \\ A_1 \diagup \quad \diagdown A_2 \\ \bullet \qquad \bullet \\ A_3 \diagdown \quad \diagup A_2 \\ \bullet \end{array} \tag{5.6}$$

wherein the Hasse diagram confirms the result on the hyper-Kähler subspaces [15, eq. (5.20)]. In contrast, the $\mathsf{MQ}_{\mathrm{D5}}$ captures the entire maximal branch easily.

- Branch $\mathcal{B}_{(1,\kappa_1)}$ — D3 segments between $(1,\kappa_1)$s: the required $\mathrm{SL}(2,\mathbb{Z})$ transformation is defined by swapping $\mathrm{NS5} \leftrightarrow (1,\kappa_1)$. One derives:

$$\mathsf{MQ}_{(1,\kappa_1)} : \qquad \begin{array}{ccc} \overset{\kappa_2+2}{\square} & \overset{\kappa_2-\kappa_1+1}{\square} & \overset{1}{\square} \\ | & | & | \\ \underset{1}{\bigcirc} - \underset{1}{\bigcirc} - \underset{1}{\bigcirc} \end{array} \tag{5.7}$$

where the number of (magnetic) flavours is computed from the D5-brane charges in the dual brane system. This provides a simpler derivation and a more complete picture of the branch geometry than [15, eq. (5.24)].

### 5.1.2 Non-Abelian

Next, the magnetic quiver proposal is demonstrated on a non-abelian $\mathcal{N} = 3$ CSM quiver. Consider the flavoured $\mathcal{N} = 3$ CSM quiver of Figure 26, which comes from a brane system of $N$ D3 branes in between four distinct types of 5-branes. For each of the four maximal

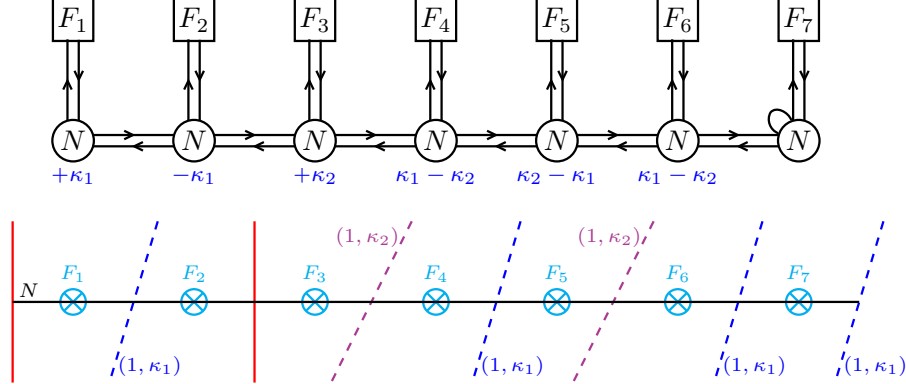

**Figure 26:** A Type IIB brane systems that gives rise to a non-Abelian CSM theory with fundamental flavours; for concreteness, $F_i > N$ for all $i \in \{1, \ldots, 7\}$ and $\kappa_1 \neq \kappa_2$, $\kappa_1\kappa_2 \neq 0$ is assumed. This theory can be thought of as a generalisation of the $T[4]$ CSM theory of Figure 25.

branches, one derives the magnetic quiver (assuming $\kappa_2 > \kappa_1 > 0$ for concreteness):

- Branch $\mathcal{B}_{\mathrm{NS5}}$ — D3 segments between NS5s: from the brane systems one reads off

$$\mathsf{MQ}_{\mathrm{NS5}} : \qquad \begin{array}{l} \overset{\square}{|} \ \kappa_1 + F_1 + F_2 + N \\ \underset{\bigcirc}{} \ N \end{array} \tag{5.8}$$

where the number of fundamental flavours is determined from: (i) the D5-brane charges of the 5-branes in between the two NS branes, plus (ii) the $N$ D3 branes that end on the stack of $F_2 > N$ D5 branes on the right, see also Figure 11.

- Branch $\mathcal{B}_{(1,\kappa_2)}$ — D3 segments between $(1,\kappa_2)$s: The $(\mathcal{T})^{|\kappa_2|}$-dual brane system allows to read off the following:

$$\mathsf{MQ}_{(1,\kappa_2)}: \quad \underset{\bigcirc\ N}{\overset{\square\ |\kappa_1 - \kappa_2| + F_4 + F_5 + 2N}{\vert}} \tag{5.9}$$

to be explicit, the number of flavours is determined as follows: (i) the D5-brane charge of the $(\mathcal{T})^{|\kappa_2|}$ transformed $(1,\kappa_1)$ 5-brane, which is $|\kappa_1 - \kappa_2|$. (ii) The two stacks of $F_4$ and $F_5$ D5s remain D5s; hence, contribute $F_4 + F_5$. (iii) The $N$ D3 branes on the left (resp. right) side of the $(1,\kappa_2)$ contribute another $N$, because they can end on the stack of $F_3$ (resp. $F_6$) D5 branes via the boundary conditions of Figure 11.

- Branch $\mathcal{B}_{\mathrm{D5}}$ — D3 segments between D5s: Using the $\mathcal{S}$-dual brane configuration, one reads off the following $\mathsf{MQ}_{\mathrm{D5}}$:

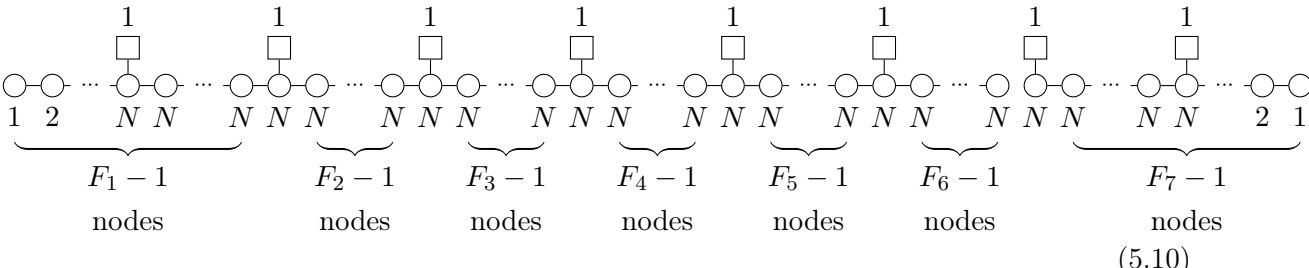

$$\tag{5.10}$$

where the seven different subsets of balanced nodes reflect the non-abelian isometry factors $\oplus_{i=1}^{7}\mathfrak{su}(F_i)$ on $\mathcal{B}_{\mathrm{D5}}$, while additional six $\mathfrak{u}(1)$ factors stem from the remaining six over-balanced gauge nodes. This matches the expected isometry algebra $(\oplus_{i=1}^{7}\mathfrak{u}(F_i))/\mathfrak{u}(1)_{\mathrm{diag}}$ of $\mathcal{B}_{\mathrm{D5}}$.

- Branch $\mathcal{B}_{(1,\kappa_1)}$ — D3 segments between $(1,\kappa_1)$s: Using the $(\mathcal{T})^{|\kappa_1|}$ dual brane system, one reads off

$$\mathsf{MQ}_{(1,\kappa_1)}: \quad \overset{\square\ M_1 \quad \square\ M_2 \quad \square\ M_3}{\underset{\bigcirc\ N \quad \bigcirc\ N \quad \bigcirc\ N}{\vert\quad\vert\quad\vert}} \quad \text{with} \quad \begin{cases} M_1 &= \kappa_2 + F_2 + F_3 + F_4 + N \\ M_2 &= \kappa_2 - \kappa_1 + F_5 + F_6 \\ M_3 &= F_7 + N \end{cases} \tag{5.11}$$

and the flavour ranks are derived from the D5-brane charges in the dual configuration.

**Comments.** The magnetic quivers for each maximal branch $\mathcal{B}$ of the moduli space allow to analyse the Higgsing RG-flows triggered by a $\mathcal{B}$ branch operator. Hence, this proposal offers predictions for general non-Abelian $\mathcal{N} = 3$ CSM quiver theories. An interesting direction for future work is to explore whether the predictions of our proposal can be corroborated by alternative approaches.

## 5.2 $\mathcal{N} = 3$ CSM theories with NS5, D5, and $(p,q)$ 5-branes

Next, the setup is generalised to brane-systems with arbitrary $(p,q)$ 5-branes (with $p > 0$, $q \neq 0$, and $p,q$ coprime integers). The SCFTs realised with $(p,q)$ 5-branes have been argued to describe the fixed points of CS quivers interpolating $T[\mathrm{U}(N)]$ CFTs [24]. For $p > 1$, there is no known Lagrangian description. To begin with, an Abelian example is used to validate the magnetic quiver results against known results derived from different techniques. Thereafter, the proposal is applied to non-Abelian $(p,q)$ 5-brane theories; here, one finds new predictions to the previously inaccessible maximal branches of non-Lagrangian $\mathcal{N} \geqslant 3$ CSM theories.

### 5.2.1 Abelian

First, consider the generalisation to the $\mathcal{N} = 3$ theory $T_{(p,q)}[3]$, realised as the world-volume theory on a single D3 branes intersecting two NS5, two D5, and two $(p,q)$ 5-branes placed alternately in Type IIB (with $p > 0$, $q \neq 0$, and $p, q$ coprime integers), see Figure 27. The maximal branches have been identified in [15, eq. (7.1)]. Again, the magnetic quiver

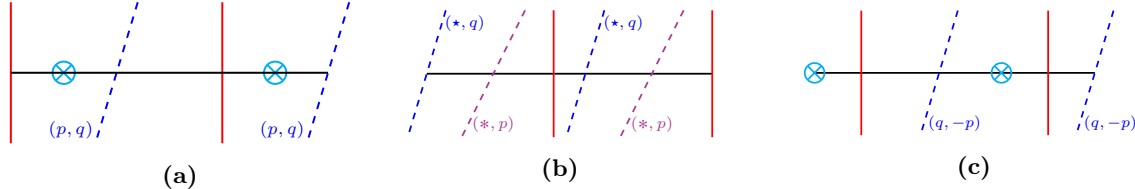

**(a)**          **(b)**          **(c)**

**Figure 27:** (a): The brane system of the $T_{(p,q)}[3]$ theory. (b): The dual brane system for branch $\mathcal{B}_{(p,q)}$ obtained via the transformation $\left(\begin{smallmatrix} p & -* \\ q & \star \end{smallmatrix}\right) \in \mathrm{SL}(2,\mathbb{Z})$ with two integers $*, \star$ that can be determined explicitly (cf. [25]), but are not relevant for the magnetic quiver derivation. (c): The dual brane system for branch $\mathcal{B}_{\mathrm{D5}}$ obtained via $\mathcal{S}$ transformation.

proposal allows to independently and straightforwardly derive these results:

- Branch $\mathcal{B}_{\mathrm{NS5}}$ — D3 segments between NS5s: From the brane system, one reads off

$$\mathsf{MQ}_{\mathrm{NS5}}: \quad \overset{\square\; |q| + 2}{\underset{\bigcirc\; 1}{\big|}} \qquad \mathcal{C}(\mathsf{MQ}_{\mathrm{NS5}}) = A_{|q|+1}\,. \tag{5.12}$$

- Branch $\mathcal{B}_{(p,q)}$ — D3 segments between $(p,q)$s: To transition to the dual brane system, one applies the $\mathrm{SL}(2,\mathbb{Z})$ transformation that swaps NS5 $\leftrightarrow (p,q)$, see Figure 27b. This affects the D5s, in contrast to the $T[3]$ case. One finds

$$\mathsf{MQ}_{(p,q)}: \quad \overset{\square\; |q| + p + 1}{\underset{\bigcirc\; 1}{\big|}} \qquad \mathcal{C}(\mathsf{MQ}_{(p,q)}) = A_{|q|+p}\,. \tag{5.13}$$

- Branch $\mathcal{B}_{\mathrm{D5}}$ — D3 segments between D5s: The $\mathcal{S}$ transformation is used, see Figure 27c, which swaps D5 $\leftrightarrow$ NS5s, and replaces $(p,q)$ by $(q,-p)$. This results in

$$\mathsf{MQ}_{\mathrm{D5}}: \quad \overset{\square\; p + 3}{\underset{\bigcirc\; 1}{\big|}} \qquad \mathcal{C}(\mathsf{MQ}_{\mathrm{D5}}) = A_{p+2}\,. \tag{5.14}$$

Consequently, all maximal branches are accurately and efficiently reproduced.

### 5.2.2 Non-Abelian

Consider a non-Abelian non-Lagrangian CSM theory derived from the world-volume theory of $N$ D3 branes in between two NS5, two $(p,q)$ 5-branes, and three stacks of D5s (labelled $F_i$); as in Figure 28. For definiteness, assume $F_i > N$ for all $i = 1, 2, 3$. The maximal branches are analysed as follows

- Branch $\mathcal{B}_{\mathrm{NS5}}$ — D3 segments between NS5s: From Figure 28a one reads

$$\mathsf{MQ}_{\mathrm{NS5}}: \quad \overset{\square\; F_1 + F_2 + |q| + N}{\underset{\bigcirc\; N}{\big|}} \tag{5.15}$$

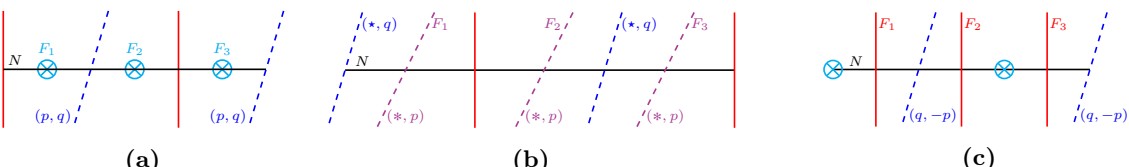

**Figure 28:** (a): The Type IIB brane setup for a non-Abelian non-Lagrangian CSM theory. (b): The dual brane system to analyse branch $\mathcal{B}_{(p,q)}$. The required $SL(2,\mathbb{Z})$ transformation is the same as in Figure 27. (c): The $\mathcal{S}$ dual brane system suitable for the $\mathcal{B}_{D5}$ branch.

where the number of fundamental flavours is composed of: (i) the D5-brane charges of the $F_1$ and $F_2$ stacks of D5s, in between the NS5 interval. (ii) The D5-brane charge of the $(p,q)$ 5-brane in the same NS5 interval. (iii) The $N$ D3 branes on the right of the interval that contribute via the boundary conditions on the $F_3 > N$ D5s.

- Branch $\mathcal{B}_{(p,q)}$ — D3 segments between $(p,q)$s: The dual brane system of Figure 28b gives rise to

$$
\mathsf{MQ}_{(p,q)}: \qquad \begin{array}{c} \square \\ | \\ \bigcirc \end{array} \begin{array}{l} (F_2 + F_3)p + |q| + N \\ \\ N \end{array} \tag{5.16}
$$

where the number of flavours follows from the relevant D5-brane charges.

- Branch $\mathcal{B}_{D5}$ — D3 segments between D5s: In the $\mathcal{S}$-dual brane system of Figure 28c, a case distinction is required: For $N \geqslant p$ with $N = j \cdot p + r$ such that $j \in \mathbb{N}_{>0}$ and $r \in \{0, 1, \ldots, p-1\}$, the magnetic quiver reads

$$
\mathsf{MQ}_{D5}: \qquad \underbrace{\overset{1}{\underset{1\ \ 2}{\bigcirc\!-\!\bigcirc}} \cdots \overset{}{\underset{N\ N}{\bigcirc\!-\!\bigcirc}} \cdots}_{\substack{F_1 - 1 \\ \text{nodes}}} \underbrace{\overset{p}{\underset{N\ N\ N}{\bigcirc\!-\!\bigcirc\!-\!\bigcirc}} \cdots}_{\substack{F_2 - 1 \\ \text{nodes}}} \underbrace{\overset{1}{\underset{N\ N\ N}{\bigcirc\!-\!\bigcirc\!-\!\bigcirc}} \cdots}_{} \underbrace{\overset{r}{\underset{N\ N\ j\cdot p}{\bigcirc\!-\!\bigcirc\!-\!\bigcirc}} \cdots \overset{}{\underset{2p\ \ p}{\bigcirc\!-\!\bigcirc}}}_{\substack{F_3 - 1 \\ \text{nodes}}} \tag{5.17a}
$$

The right-hand-side of the quiver follows from moving the $(q, -p)$ 5-brane through some of the $F_3$ many NS5s. Due to the intersection number $\left|\det \begin{pmatrix} 1 & q \\ 0 & -p \end{pmatrix}\right| = p$, each crossing creates/annihilates $p$ D3 branes.

For $N < p$, the magnetic quiver reads:

$$
\mathsf{MQ}_{D5}: \qquad \underbrace{\overset{1}{\underset{1\ \ 2}{\bigcirc\!-\!\bigcirc}} \cdots \overset{}{\underset{N\ N}{\bigcirc\!-\!\bigcirc}} \cdots}_{\substack{F_1 - 1 \\ \text{nodes}}} \underbrace{\overset{p}{\underset{N\ N\ N}{\bigcirc\!-\!\bigcirc\!-\!\bigcirc}} \cdots}_{\substack{F_2 - 1 \\ \text{nodes}}} \underbrace{\overset{1}{\underset{N\ N\ N}{\bigcirc\!-\!\bigcirc\!-\!\bigcirc}} \cdots \overset{}{\underset{N\ N}{\bigcirc\!-\!\bigcirc}} \cdots \overset{N}{\underset{N\ N}{\bigcirc\!-\!\bigcirc}}}_{\substack{F_3 - 1 \\ \text{nodes}}} \tag{5.17b}
$$

since $N < p$ the $(q, -p)$ 5-brane on the right-hand-side of Figure 28c cannot cross any of the $F_3$ many NS5s. Hence, the suspended D3s just contribute $N$ many fundamental flavours to the right-most (magnetic) gauge node.

# 6 Conclusion and outlook

In this paper, the symmetries, the moduli space of vacua, and certain RG-flows of linear and circular Chern–Simons Matter quiver theories with $\mathcal{N} = 4$ supersymmetry have been systematically analysed.

For CS-levels $\kappa_i = \pm\kappa$ with $|\kappa| = 1$, the main tool has been the explicit $\mathrm{SL}(2,\mathbb{Z})$ dualisation into a standard non-CS 3d $\mathcal{N} = 4$ quiver theory Q. For this, recently developed techniques such as Higgs branch subtraction and the decay and fission algorithm allow to trace out the Higgs/Coulomb branch Hasse diagram — or equivalently, the patter of RG-flows triggered by a VEV to a Higgs/Coulomb branch operator.

For CS-levels $\kappa_i = \pm\kappa$ with $|\kappa| > 1$, a dualisation into a non-CS Lagrangian $\mathcal{N} = 4$ theory is unavailable. Instead, a simple prescription has been devised to deduce two magnetic quivers $\mathsf{MQ}_{A/B}$ — one for the A and one for the B branch of the CSM theory. This proposal has passed a variety of consistency checks. For example, the A/B-branch limits of the CSM index match the Colomb branch Hilbert series of $\mathsf{MQ}_{A/B}$; moreover, the decay and fission predictions for $\mathsf{MQ}_{A/B}$ match the moduli of D3 brane segments in CSM brane configurations.

Building on these insights in the quantum moduli space branches of a CSM with emergent $\mathcal{N} = 4$ supersymmetry, the magnetic quiver proposal has been extended to the maximal branch of $\mathcal{N} = 3$ CSM theories from brane systems with several different $(p, q)$ 5-branes. For $p > 1$, these CSM theory have no known Lagrangian interpretation. This greatly improves the ease of derivation and the completeness of the branch geometry compared to previous studies.

**Other $\mathcal{N} = 4$ CSM.** Beyond the class of theories considered here, there are more 3d $\mathcal{N} = 4$ CSM theories, which one might group as follows: (i) CSM from the D3 world-volume theory in between NS5 and $(1, \kappa)$ 5-branes with orientifold 3-planes. This gives rise to linear orthosymplectic CSM quiver theories of the kind introduced in [4, 5]. (ii) Circular orthosymplectic CSM quiver theories, realised by above D3-NS5-$(1, \kappa)$ brane systems where one direction is a circle. (iii) CSM theories beyond standard Type IIB brane systems: for instance the CS theories constructed from the $T_N$ theories [17, 47, 65].

It is natural to expected that cases (i) and (ii) are within reach of the techniques developed in this paper. As of present, the $|\kappa| = 1$ case should admit a 3d $\mathcal{N} = 4$ Lagrangian $\mathrm{SL}(2,\mathbb{Z})$ dual; this is rather transparent from the brane system, even though the (field theoretic) dualisation algorithm is still under development for this setup. The magnetic quiver approach for $|\kappa| > 1$ is likely to be a adaptation/extension of the orthosymplectic magnetic quiver techniques introduced in [66] (and subsequent works). The exploration of the class (iii) is more speculative at this point, but an intriguing direction for the future.

**Non-Abelian $\mathcal{N} = 3$ CSM via $(p, q)$ 5-branes.** Another natural direction is the exploration of the maximal branches of non-Abelian $\mathcal{N} = 3$ CSM theories realised via $(p, q)$ 5-brane [24]. The proof of concept has been given in Section 5.2, but a more systematic analysis is left for future work.

# Acknowledgments

We would like to thank Benjamin Assel, Riccardo Comi, Amihay Hanany, Noppadol Mekareeya, and Sara Pasquetti for useful discussions. The work of FM and MS is supported by the Austrian Science Fund (FWF), START project "Phases of quantum field theories: symmetries and vacua" STA 73-N [grant DOI: 10.55776/STA73]. FM and MS also acknowledge support from the Faculty of Physics, University of Vienna.

# A  3d $\mathcal{N} = 4$ Chern–Simons matter theories

## A.1  Field theory

Gauge theories in 3d allow for a supersymmetric Chern–Simons term, with suitably quantised Chern–Simons level $\kappa$, that preserve $\mathcal{N} = 3$ supersymmetry [3]. Starting from an 3d $\mathcal{N} = 4$ Lagrangian $G$ gauge theory with a $\mathcal{N} = 4$ $G$ vector multiplet and hypermultiplets in a representation of $G$, the $\mathcal{N} = 4$ action is complemented by a CS-action for the 3d $\mathcal{N} = 2$ vector multiplet plus a superpotential type term $\sim \kappa \operatorname{Tr}[\Phi^2]$ for the adjoint $\mathcal{N} = 2$ chiral multiplet $\Phi$ inside the $\mathcal{N} = 4$ vector. The included CS terms generate masses for the vector multiplet fields; hence, the vector multiplet fermions and scalar can be integrated out as auxiliary fields at low energies. This results in a pure Chern–Simons theory with a new quartic superpotential for the hypermultiplets.

Starting from this, the manifest $\mathcal{N} = 3$ supersymmetry can enhance to $\mathcal{N} = 4$ at the conformal fixed point if the CS-levels and the matter content are suitably chosen, as first observed in [4, 5] and recently discussed, for example, in [17, 65].

## A.2  Brane realisation

The brane realisation of 3d $\mathcal{N} = 4$ CSM SCFTs in Type IIB superstring theory involves D3 branes, NS5 branes, and $(1, \kappa)$ 5-branes; see for instance [5, 7, 25].

- D3s extend along directions 0123, with the 3 direction compact.

- NS5s span directions 012789.

- $(1, \kappa)$ 5-branes span direction $012[4,7]_\theta[5,8]_\theta[6,9]_\theta$, with $[i,j]_\theta$ denotes the tilted direction $\cos\theta x_i + \sin\theta x_j$ in the $(x_i, x_j)$ plane. The angle $\theta$ is a fixed function of $\kappa$ to preserve $\mathcal{N} = 3$ supersymmetry [18, 21]: $\tan\theta = \kappa$.

| brane | 0 | 1 | 2 | 3 | 4 | 5 | 6 | 7 | 8 | 9 |
|---|---|---|---|---|---|---|---|---|---|---|
| D3 ( $-$ ) | $\times$ | $\times$ | $\times$ | $\times$ | | | | | | |
| NS5 ( $\vert$ ) | $\times$ | $\times$ | $\times$ | | $\times$ | $\times$ | $\times$ | | | |
| $(1,\kappa)$-5 ( $\diagup$ ) | $\times$ | $\times$ | $\times$ | | $[4,7]_\theta$ | | $[5,8]_\theta$ | | $[6,9]_\theta$ | |

**Table 3:** Space-time occupation and notation for CSM brane systems. For linear CSM quiver theories, the $x^3$ direction is non-compact, but the D3 segments are finite in $x^3$ as they end on the 5-branes. For circular CSM quiver theories, the $x^3$ direction is compactified to a circle.

Next, the CMS quiver theory to a brane configuration of a sequence of NS5s and $(1, \kappa)$ 5-branes along the $x^3$ direction is specified as shown in Table 4.

If there is a extra $U(1)_{\text{axial}}$ symmetry (i.e. an SCFT with $\mathcal{N} = 4$), then one can distinguish twisted and untwisted hypermultiplets, see Footnote 3.

When matching the index $\mathcal{I}$ of two $SL(2, \mathbb{Z})$ dual theories $Q$ and $Q^\vee$, it is crucial to remember that the equality holds provided that the transformation of the $U(1)_{\text{axial}}$ fugacity $t$ is taken into account:

$$Q \xrightarrow{\mathcal{S}} Q^\vee : \quad \mathcal{I}_Q(t) = \mathcal{I}_{Q^\vee}(t^{-1}), \tag{A.1a}$$

$$Q \xrightarrow{\mathcal{T}} Q^\vee : \quad \mathcal{I}_Q(t) = \mathcal{I}_{Q^\vee}(t^{-1}), \tag{A.1b}$$

$$Q \xrightarrow{\mathcal{T}^T} Q^\vee : \quad \mathcal{I}_Q(t) = \mathcal{I}_{Q^\vee}(t). \tag{A.1c}$$

This is already incorporated in the dualisation algorithm, as reviewed in Appendix C.

| Brane configuration | Supermultiplet | Quiver |
|---|---|---|
|  | $\mathcal{N} = 4$ vector multiplet |  |
|  | $\mathcal{N} = 2$ vector multiplet with CS-level $+\kappa$ |  |
|  | $\mathcal{N} = 2$ vector multiplet with CS-level $-\kappa$ |  |
|  | $\mathcal{N} = 4$ twisted vector multiplet |  |
|  | $\mathcal{N} = 4$ bifundamental hypermultiplet |  |
|  | $\mathcal{N} = 4$ bifundamental twisted hypermultiplet |  |
|  | $\mathcal{N} = 4$ fundamental hypermultiplet |  |
|  | $\mathcal{N} = 4$ fundamental twisted hypermultiplet |  |

**Table 4:** Conventions for the 3d multiplets that compose the D3 world-volume theory of the Type IIB brane systems involving D3 brane in between 5-branes. The branes follow the convention of Table 3. Here, the black horizontal lines are assumed to be a stack of $N$ D3s branes, while the 5-branes appear as single branes. The wiggly lines represent fundamental strings, from which the world-volume theory is deduced. The quiver notation is given with respect to 3d $\mathcal{N} = 2$. All the gauge (round nodes) groups are unitary.

# B  Index and Hilbert series checks for CSM theories

This appendix verifies the magnetic quiver proposal of Sections 3 and 4 by matching the index's A/B branch limits with the Coulomb branch Hilbert series of $\mathsf{MQ}_{A/B}$.

## B.1  Linear $\mathbf{U(2)_2 \times U(2)_{-2} \times U(2)_2 \times U(2)_{-2}}$

For the CSM theory of Section 3.1, consider $\kappa = N = 2$ and validate the proposal.

**Index.**  The global symmetry algebra (3.5) in this $\kappa = N$ case reads $[\mathfrak{su}(3)_u]_A \oplus [\mathfrak{u}(1)_q]_B$. Using this fugacity convention, the index expansion of the $\mathrm{CSM}_2$ theory reads

$$
\begin{aligned}
\mathcal{I} = 1 &+ x\left(t^2 + t^{-2}[1,1]_u\right) \tag{B.1} \\
&+ x^2\left(t^4(q+q^{-1}+2) + t^{-4}\left([2,2]_u + 2[1,1]_u + 1\right) - 2\right) \\
&+ x^3\left(2t^6\left(q+q^{-1}+1\right) + t^2\left(([1,1]_u - 1)(q+q^{-1}) - 2\right)\right. \\
&\qquad -t^{-2}\left([3,0]_u + [0,3]_u + 4[1,1]_u + 1\right) \\
&\qquad \left.+t^{-6}\left([3,3]_u + 2[3,0]_u + 2[0,3]_u + 2[2,2]_u + 2[1,1]_u\right)\right) + O(x^3)\,,
\end{aligned}
$$

and the map between the highest weight fugacities and those in Figure 13a is given by

$$
Y_1/Y_3 = \frac{u_1^2}{u_2}\,, \qquad Y_2/Y_4 = q\,, \qquad Y_3/Y_5 = \frac{u_2^2}{u_1}\,. \tag{B.2a}
$$

In view of the $\mathsf{MQ}_{A/B}$ proposal, consider the projections of the index (B.1) onto the A/B branch. This is realised by redefining the $x$ and $t$ fugacities as [45]

$$
x \to (a\,b)^{\frac{1}{2}}\,, \quad t \to (b/a)^{\frac{1}{4}}\,. \tag{B.3}
$$

The Hilbert series on the A branch is obtained by turning off $b$ (i.e. $b \to 0$) and expanding around $a$. The result matches with the Coulomb branch Hilbert series of $\mathsf{MQ}_A$:

$$
\begin{aligned}
\mathrm{HS}_{\mathcal{C}}(\mathsf{MQ}_A) = 1 &+ a\left([1,1]_u\right) + a^2\left([2,2]_u + 2[1,1]_u + 1\right) \tag{B.4} \\
&+ a^3\left([3,3]_u + 2[3,0]_u + 2[0,3]_u + 2[2,2]_u + 2[1,1]_u\right) + O\left(a^4\right)\,,
\end{aligned}
$$

where the map between these global symmetry fugacities and those used in Figure 13c is

$$
W_1/W_2 = \frac{u_1^2}{u_2}\,, \qquad W_2/W_3 = \frac{u_2^2}{u_1}\,. \tag{B.5}
$$

On the other hand, the Hilbert series on the B branch is obtained by turning off $a$ (i.e. $a \to 0$) and expanding around $b$. The result matches with the Coulomb branch Hilbert series of $\mathsf{MQ}_B$:

$$
\mathrm{HS}_{\mathcal{C}}(\mathsf{MQ}_B) = 1 + b + b^2\left(q + q^{-1} + 2\right) + 2b^3\left(q + q^{-1} + 1\right) + O\left(b^4\right)\,,
$$

where the map between these global symmetry fugacities and those used in Figure 13d is

$$
\widetilde{W}_1/\widetilde{W}_2 = q\,. \tag{B.6}
$$

**Higgsing pattern.**  Figure 29 shows the A branch Hasse diagram of the CSM theory and the Coulomb branch Hasse diagram of the magnetic quiver $\mathsf{MQ}_A$. Analogously, Figure 30 displays the B branch Hasse diagram of the CSM theory and the Coulomb branch Hasse diagram of the magnetic quiver $\mathsf{MQ}_B$. In the latter, the GK dual of $\mathrm{CSM}_2$ has been used for convenience. Both the A/B branch Higgsings confirm the pattern that has been presented in Section 3.1 from the brane perspective.

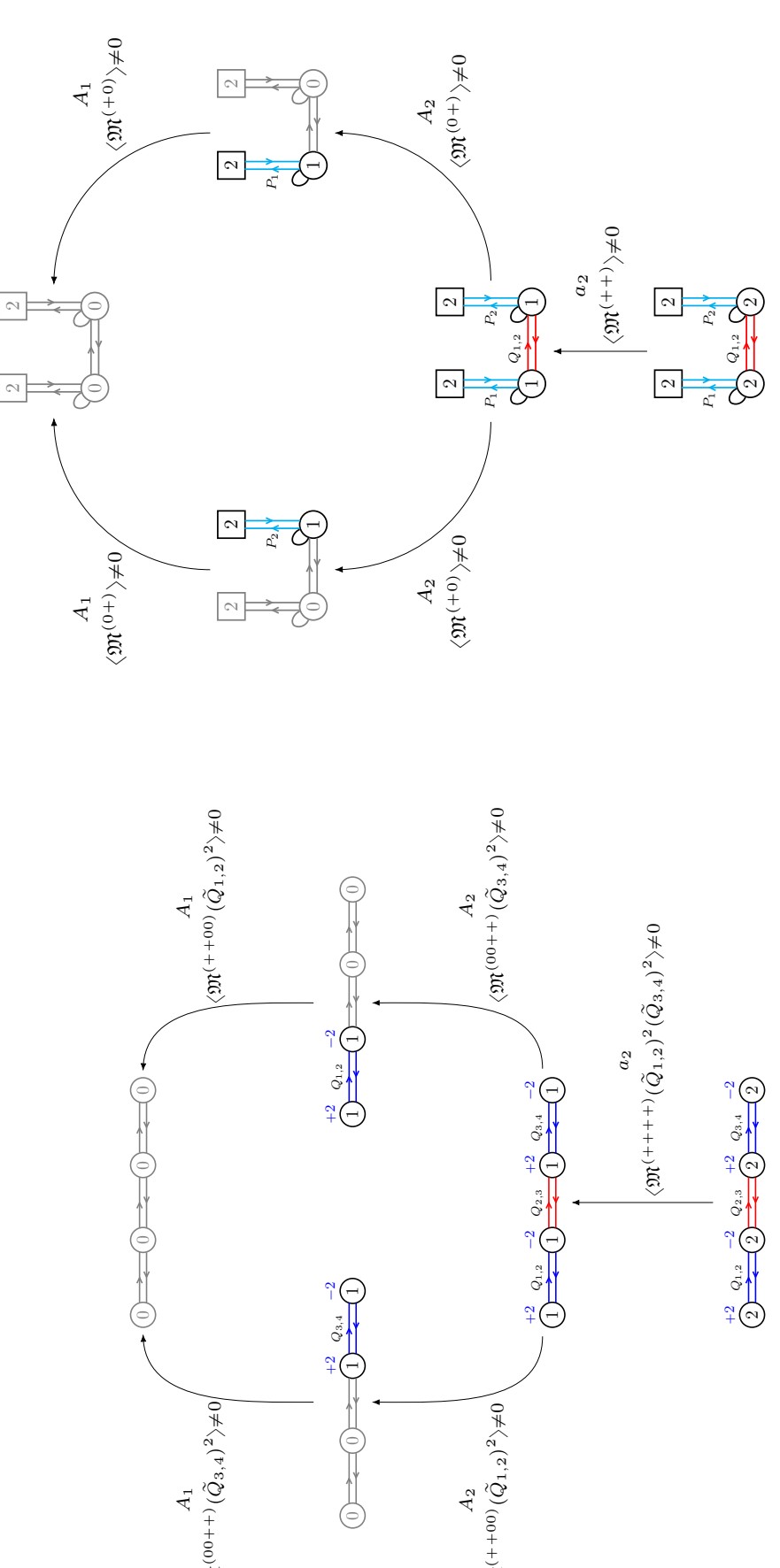

**(a)** A branch Higgsing of the CSM$_2$ theory.

**(b)** Coulomb branch Higgsing of the magnetic quiver MQ$_A$.

**Figure 29:** Hasse diagrams of the A/Coulomb branches of CSM$_2$/MQ$_A$, respectively. In each step the names assigned to the fields have noting to do with the names of the fields of the prior/subsequent step. Close to the arrows representing the Higgsing, both the geometric transition and the operator taking VEV are written. The grey pieces are the trivial parts of the quiver.

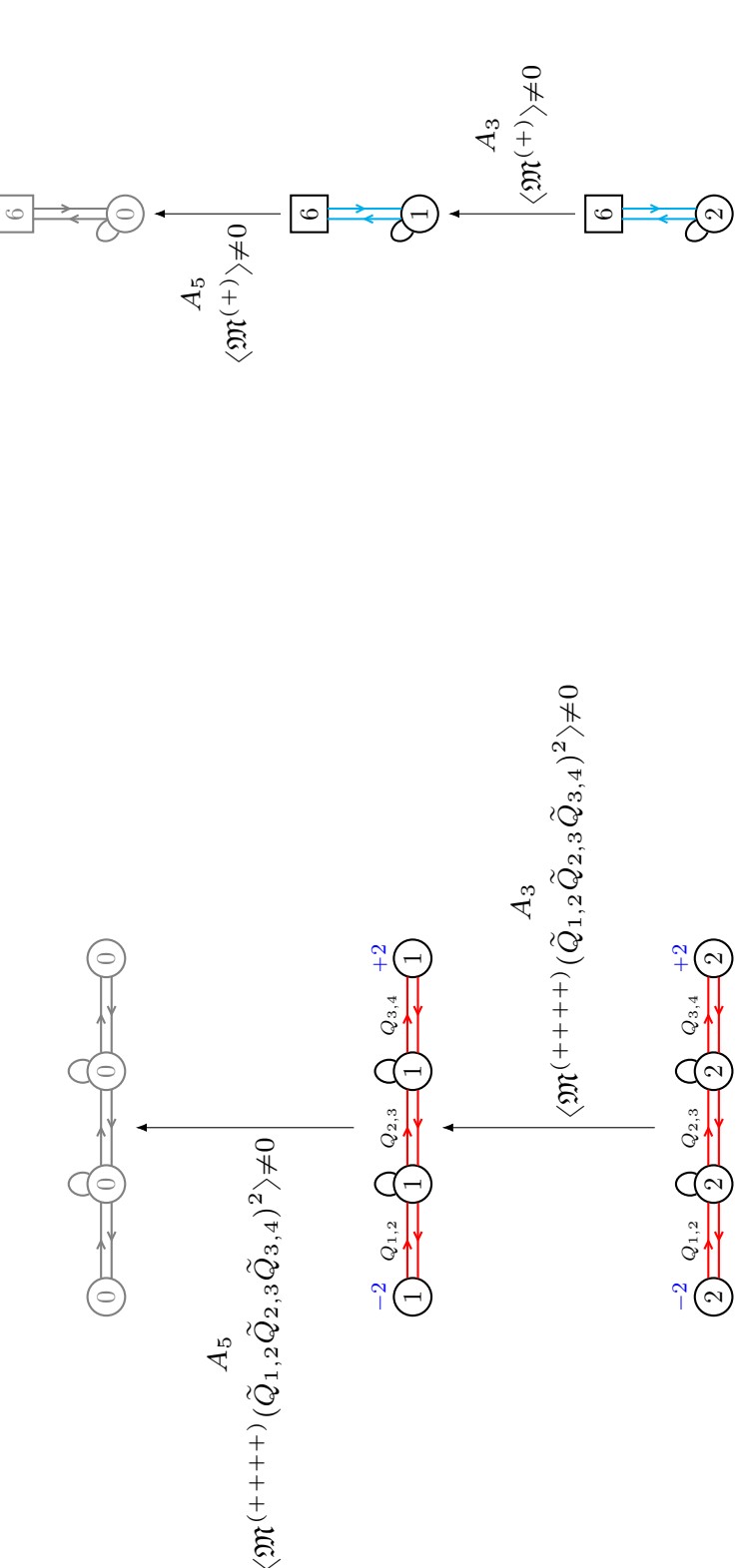

**(a)** B branch Higgsing of the $CSM_2$ theory.

**(b)** Coulomb branch Higgsing of the magnetic quiver $MQ_B$.

**Figure 30:** Hasse diagrams of the B/Coulomb branches of $GK(CSM_2)/MQ_B$, respectively. In particular, on the CSM side the GK-dual version of $CSM_2$ has been employed. In each step the names assigned to the fields have noting to do with the names of the fields of the prior/subsequent step. Close to the arrows representing the Higgsing, both the geometric transition and the operator taking VEV are written. The grey pieces are the trivial parts of the quiver.

## B.2 Linear $U(1)_2 \times U(2)_{-2} \times U(1)_2 \times U(2)_{-2}$

Recall that the example of Section 3.1 is only a special case of the 4 node non-Abelian CSM quiver shown in Table 1b. Here, a different parameter region is explored by considering the CSM quiver in Figure 31a. Among others, this parameter choice implies that a GK-duality is applied before the magnetic quivers $\mathsf{MQ}_{A/B}$ are read off, see Figures 31b and 31c. This serves to show that the magnetic quivers can be derived for any (good) parameter values.

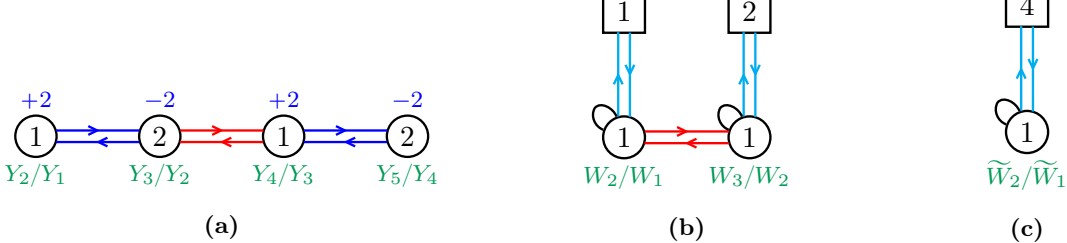

**(a)**               **(b)**               **(c)**

**Figure 31:** (a) The starting $\mathsf{CSM}_2$ theory. (b) The magnetic quiver $\mathsf{MQ}_A$. (c) The magnetic quiver $\mathsf{MQ}_B$. The parameters of the magnetic quivers have different names with respect to those of the CSM theories because they are not related by a duality, so no parameters map can be established.

The global symmetry algebra of the CSM theory in Figure 31a is

$$\left[\mathfrak{su}(2)_v \oplus \mathfrak{u}(1)_w\right]_A \oplus \left[\mathfrak{u}(1)_q\right]_B . \tag{B.7}$$

and its refined index expansion is perturbatively computed to read

$$
\begin{aligned}
\mathcal{I} = 1 &+ x(t^{-2}([2]_v + 1) + t^2) + x^{3/2}(t^{-3}([1]_v(w + w^{-1}))) \\
&+ x^2(t^{-4}([4]_v + [2]_v + 1) + t^4(q + q^{-1} + 1) - 3) \\
&+ x^{5/2}(t^{-5}([3]_v(w + w^{-1}) + [1]_v(w + w^{-1})) - t^{-1}([1]_v(w + w^{-1}))) \\
&+ x^3(t^6(q + q^{-1} + 1)) + t^2(([2]_v - 1)(q + q^{-1}) - 1) - 4t^{-2}[2]_v \\
&+ t^{-6}([6]_v + [4]_v + [2]_v(w^2 + w^{-2} + 1) + 1)) + O(x^{7/2}) .
\end{aligned}
\tag{B.8}
$$

The map between the algebra fugacities and those used in Figure 31a is

$$Y_1/Y_3 = v^2 , \qquad Y_5/Y_3 = w\,v , \qquad Y_2/Y_4 = q . \tag{B.9}$$

From the index expansion (B.8) one can inspect the theory analogously to Section 2.1. Importantly, one can project the index (B.8) onto the A/B branch, using (B.3) and the series expansions thereby described. The result is that the A branch index matches with the Coulomb branch Hilbert series of $\mathsf{MQ}_A$:

$$
\begin{aligned}
\mathrm{HS}_{\mathcal{C}}(\mathsf{MQ}_A) = 1 &+ a([2]_v + 1) + a^{3/2}([1]_v(w + w^{-1})) \\
&+ a^2([4]_v + [2]_v + 1) + a^{5/2}([3]_v(w + w^{-1}) + [1]_v(w + w^{-1})) \\
&+ a^3([6]_v + [4]_v + [2]_v(w^2 + w^{-2} + 1) + 1) + O(a^{7/2}) ,
\end{aligned}
$$

where the map between these global symmetry fugacities and those used in Figure 31b is

$$W_1/W_2 = v^2 , \qquad W_3/W_2 = w\,v . \tag{B.10}$$

Likewise, the B branch limit matches the Coulomb branch Hilbert series of $\mathsf{MQ}_B$:

$$\mathrm{HS}_{\mathcal{C}}(\mathsf{MQ}_B) = 1 + b + b^2\left(q + q^{-1} + 1\right) + b^3\left(q + q^{-1} + 1\right) + O(b^4) ,$$

where the map between these global symmetry fugacities and those used in Figure 31c is

$$\widetilde{W}_1/\widetilde{W}_2 = q . \tag{B.11}$$

## B.3 Linear $U(2)_2 \times U(2)_0 \times U(2)_{-2} \times U(1)_2 \times U(1)_{-2}$

Consider the 5-nodes CSM theory in Figure 32a. Its magnetic quivers $\mathsf{MQ}_{A/B}$ are shown in Figures 32b and 32c, respectively.

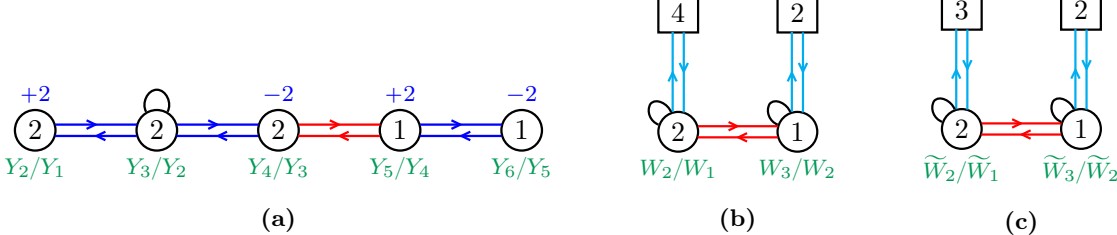

**(a)**         **(b)**        **(c)**

**Figure 32:** (a) The starting $\mathsf{CSM}_2$ theory. (b) The magnetic quiver $\mathsf{MQ}_A$. (c) The magnetic quiver $\mathsf{MQ}_B$. The parameters of the magnetic quivers have different names with respect to those of the CSM theories because they are not related by a duality, so no parameters map can be established.

The global symmetry algebra of the CSM theory in Figure 32a is

$$\left[\mathfrak{u}(1)_v \oplus \mathfrak{u}(1)_w\right]_A \oplus \left[\mathfrak{su}(2)_u \oplus \mathfrak{u}(1)_q\right]_B . \tag{B.12}$$

and its refined index expansion is perturbatively computed to read

$$\begin{aligned}
\mathcal{I} =\ & 1 + x\left(t^2([2]_u + 1) + 2t^{-2}\right) + x^{3/2}\left(t^{-3}\left(v + v^{-1}\right)\right) \tag{B.13}\\
& + x^2\left(t^4([4]_u + 2[2]_u + [1]_u(q + q^{-1}) + 2) + t^{-4}\left(w + w^{-1} + 4\right) + [2]_u - 3\right) \\
& + x^{5/2}\left(t^{-1}([2]_u(v + [2]_u v^{-1}) + t^{-5}\left(vw^{-1} + v^{-1}w + 3(v + v^{-1})\right)\right) \\
& + x^3(t^6([6]_u + 2[4]_u + [3]_u(q + q^{-1}) + 4[2]_u + 2[1]_u(q + q^{-1}) + 2) \\
& \quad + t^2([4]_u - 4[2]_u - 1]) + t^{-2}([2]_u(w + w^{-1} + 1) - (w + w^{-1}) - 5) \\
& \quad + t^{-6}(v^2 + v^{-2} + 2(w + w^{-1}) + 7)) + O(x^{7/2}) .
\end{aligned}$$

The map between the algebra fugacities and those used in Figure 32a is

$$Y_1/Y_4 = v\,, \qquad Y_2/Y_3 = u^2\,, \qquad Y_6/Y_4 = w\,, \qquad Y_5/Y_3 = q\,u\,. \tag{B.14}$$

Importantly, one can project the index (B.13) onto the A/B branch, using (B.3) and the series expansions thereby described. The result is that the A branch index matches with the Coulomb branch Hilbert series of $\mathsf{MQ}_A$:

$$\begin{aligned}
\mathrm{HS}_\mathcal{C}(\mathsf{MQ}_A) =\ & 1 + 2a + a^{3/2}\left(v + v^{-1}\right) + a^2\left(w + w^{-1} + 4\right) \tag{B.15}\\
& + a^{5/2}\left(vw^{-1} + v^{-1}w + 3(v + v^{-1})\right) \\
& + a^3\left(v^2 + v^{-2} + 2(w + w^{-1}) + 7\right) + O(a^{7/2})\,,
\end{aligned}$$

where the map between these global symmetry fugacities and those used in Figure 32b is

$$W_2/W_1 = v\,, \qquad W_3/W_1 = v\,w\,. \tag{B.16}$$

Likewise, the B branch limit matches the Coulomb branch Hilbert series of $\mathsf{MQ}_B$:

$$\begin{aligned}
\mathrm{HS}_\mathcal{C}(\mathsf{MQ}_B) =\ & 1 + b\,([2]_u + 1) + b^2\left([4]_u + 2[2]_u + [1]_u(q + q^{-1}) + 2\right) \tag{B.17}\\
& + b^3\left([6]_u + 2[4]_u + [3]_u(q + q^{-1}) + 4[2]_u + 2[1]_u(q + q^{-1}) + 2\right) + O(b^4)\,,
\end{aligned}$$

where the map between these global symmetry fugacities and those used in Figure 32c is

$$\widetilde{W}_1/\widetilde{W}_2 = u^2\,, \qquad \widetilde{W}_3/\widetilde{W}_2 = q\,u\,. \tag{B.18}$$

## B.4 Linear $\mathbf{U(1)_2 \times U(1)_0 \times U(1)_0 \times U(1)_{-2} \times U(1)_2 \times U(1)_{-2}}$

Consider the 6-nodes Abelian CSM theory in Figure 33a; whose magnetic quivers $\mathsf{MQ}_{A/B}$ are shown in Figures 33b and 33c, respectively.

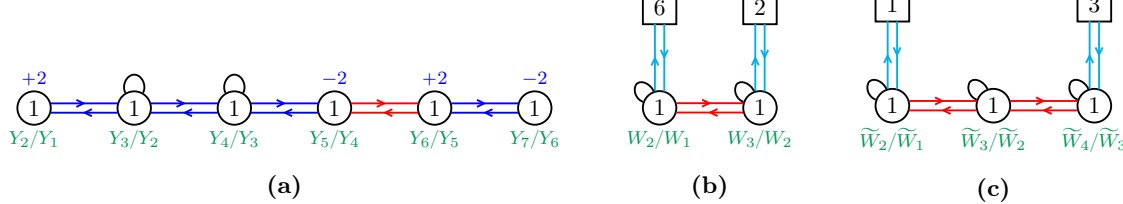

(a)  (b)  (c)

**Figure 33:** (a) The starting $\mathsf{CSM}_2$ theory. (b) The magnetic quiver $\mathsf{MQ}_A$. (c) The magnetic quiver $\mathsf{MQ}_B$. The parameters of the magnetic quivers have different names with respect to those of the CSM theories because they are not related by a duality, so no parameters map can be established.

The global symmetry algebra of the CSM theory in Figure 33a is

$$[\mathfrak{u}(1)_v \oplus \mathfrak{u}(1)_w]_A \oplus [\mathfrak{su}(3)_u \oplus \mathfrak{u}(1)_q]_B \,. \tag{B.19}$$

and its refined index expansion is evaluate to read

$$
\begin{aligned}
\mathcal{I} = {} & 1 + x \left( t^2([1,1]_u + 1) + 2t^{-2} \right) + x^{3/2} \left( t^{-3} \left( w + w^{-1} \right) \right) \\
& + x^2 \left( t^4([0,1]_u q + [1,0]_u q^{-1} + [2,2]_u + [1,1]_u + 1) + 3t^{-4} - 4 \right) \\
& + x^{5/2} \left( t^{-1}(([1,1]_u - 1) \left( w + w^{-1} \right)) + 2t^{-5} \left( w + w^{-1} \right) \right) \\
& + x^3 \left( t^6([1,2]_u q + [2,1]_u q^{-1} + [0,1]q + [1,0]_u q^{-1} + [3,3]_u + [2,2]_u + [1,1]_u + 1) \right. \\
& \qquad - t^2([1,0]_u q^{-1} + [0,1]_u q + [3,0]_u + [0,3]_u + 4[1,1]_u) \\
& \qquad \left. + t^{-6} \left( w^2 + w^{-2} + 4 \right) - 4t^{-2} \right) \\
& + x^{7/2}(t^1(([2,2]_u - [1,1]_u) \left( w - w^{-1} \right)) - 4t^{-3} \left( w + w^{-1} \right) \\
& \qquad + t^{-7} \left( v + v^{-1} + 3 \left( w + w^{-1} \right) \right)) \\
& + x^4 \left( t^8 \left( [0,2]_u q^2 + [2,0]_u q^{-2} + [2,3]_u q + [3,2]_u q^{-1} + [1,2]_u q + [2,1]_u q^{-1} \right. \right. \\
& \qquad\qquad + [0,1]_u q + [1,0]_u q^{-1} + [4,4]_u + [3,3]_u + [2,2]_u + [1,1]_u + 1) \\
& \qquad - t^4 \left( [2,0]_u q + [0,2]_u q^{-1} + 2[1,2]_u q + 2[2,1]_u q^{-1} + 2[0,1]_u q + 2[1,0]_u q^{-1} \right. \\
& \qquad\qquad + [4,1]_u + [1,4]_u + 4[2,2]_u) \\
& \qquad + t^{-4}((w^2 + w^{-2}) ([1,1]_u - 1) - 4) \\
& \qquad \left. + t^{-8} \left( vw^{-1} + v^{-1}w + 2 \left( w^2 + w^{-2} \right) + 5 \right) + 5[1,1]_u + 5 \right) + O(x^{9/2}) \,.
\end{aligned}
\tag{B.20}
$$

The map between the symmetry fugacities in (B.19) and those used in Figure 33a is

$$Y_1/Y_5 = v \,, \qquad Y_2/Y_3 = \frac{u_1^2}{u_2} \,, \qquad Y_3/Y_4 = \frac{u_2^2}{u_1} \,, \qquad Y_7/Y_5 = w \,, \qquad Y_6/Y_3 = q\frac{u_1}{u_2} \,. \tag{B.21}$$

To validate the $\mathsf{MQ}_{A/B}$ proposal, one projects the index (B.13) on the A/ B branch, using (B.3). Again, the A branch limit matches the Coulomb branch Hilbert series of $\mathsf{MQ}_A$:

$$
\begin{aligned}
\mathrm{HS}_{\mathcal{C}}(\mathsf{MQ}_A) = {} & 1 + 2a + a^{3/2} \left( w + w^{-1} \right) + 3a^2 + a^{5/2} \left( 2(w + w^{-1}) \right) \\
& + a^3 \left( w^2 + w^{-2} + 4 \right) + a^{7/2} \left( v + v^{-1} + 3(w + w^{-1}) \right) \\
& + a^4 \left( vw^{-1} + v^{-1}w + 2(w^2 + w^{-2}) + 5 \right) + O(a^{9/2}) \,,
\end{aligned}
\tag{B.22}
$$

where the map between these global symmetry fugacities and those used in Figure 33b is

$$W_1/W_2 = v\,, \qquad W_3/W_2 = w\,. \tag{B.23}$$

Likewise, the B branch limit matches the Coulomb branch Hilbert series of $\mathsf{MQ}_B$:

$$
\begin{aligned}
\mathrm{HS}_\mathcal{C}(\mathsf{MQ}_B) = {}& 1 + b\left([1,1]_u + 1\right) + b^2\left([0,1]_u q + [1,0]_u q^{-1} + [2,2]_u + [1,1]_u + 1\right) \\
& + b^3\left([1,2]_u q + [2,1]_u q^{-1} + [0,1]_u q + [1,0]_u q^{-1}\right. \\
& \qquad\left. +[3,3]_u + [2,2]_u + [1,1]_u + 1\right) \\
& + b^4\left([0,2]_u q^2 + [2,0]_u q^{-2} + [2,3]_u q + [3,2]_u q^{-1} + [1,2]_u q + [2,1]_u q^{-1}\right. \\
& \qquad\left. +[0,1]_u q + [1,0]_u q^{-1} + [4,4]_u + [3,3]_u + [2,2]_u + [1,1]_u + 1\right) \\
& + O\left(b^5\right)\,, \tag{B.24}
\end{aligned}
$$

where the map between these global symmetry fugacities and those used in Figure 33c is

$$\widetilde{W}_2/\widetilde{W}_1 = \frac{u_1^2}{u_2}\,, \qquad \widetilde{W}_1/\widetilde{W}_3 = \frac{u_2^2}{u_1}\,, \qquad \widetilde{W}_4/\widetilde{W}_1 = q\frac{u_1}{u_2}\,. \tag{B.25}$$

## B.5  Circular $\mathbf{U(N)}_{+\boldsymbol{\kappa}} \times \mathbf{U(N)}_{-\boldsymbol{\kappa}} \times \mathbf{U(N)}_{+\boldsymbol{\kappa}} \times \mathbf{U(N)}_{-\boldsymbol{\kappa}}$

Consider the 4-nodes circular $\mathrm{CSM}_{\kappa>0}$ theory in Figure 34a; whose magnetic quiver $\mathsf{MQ}_A \equiv \mathsf{MQ}_B$ is shown in Figure 34b (for $\kappa = 1$, $\mathsf{MQ}_{A/B}$ is the $\mathcal{T}^T$-dual of the CSM theory; note that it is a self-mirror theory). It is a specific instance of the generic example 1 discussed in Section 4. The Coulomb branch of $\mathsf{MQ}_{A/B}$ is the moduli space of $N$ SU(2) instantons on $\mathbb{C}^2/\mathbb{Z}_{2\kappa}$ with framing $(0^{\kappa-1}, 1, 0^{\kappa-1}, 1)$. The index vs Hilbert series check is shown in Table 5. When $\kappa = 1$ the check is performed on the index itself as the $\mathcal{T}^T$-duality holds, so no Hilbert series are written.

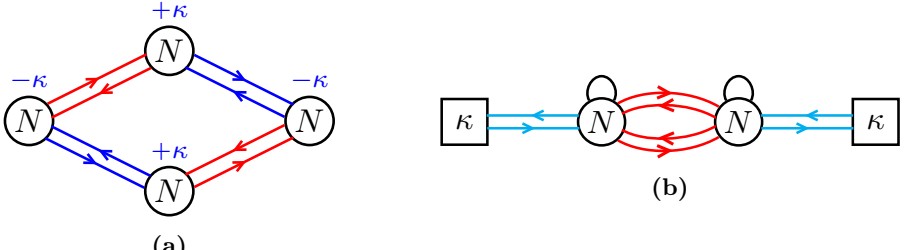

**(a)**

**(b)**

**Figure 34:** (a) The starting $\mathrm{CSM}_{\kappa>0}$ circular theory. (b) The magnetic quiver $\mathsf{MQ}_{A/B}$. For $\kappa = 1$, $\mathsf{MQ}_{A/B}$ is the $\mathcal{T}^T$-dual of the CSM theory.

| $N$ | $\kappa$ | Index and Hilbert series |
|---|---|---|
| 1 | 1 | $\mathcal{I}(\mathrm{CSM}) = 1 + \left(4t^2 + 4t^{-2}\right)x + \left(4t^3 + 4t^{-3}\right)x^{3/2} + \left(9t^4 + 9t^{-4}\right)x^2$ $+ \left(12t^5 + 12t^{-5} - 4t - 4t^{-1}\right)x^{5/2} + \left(22t^6 + 22t^{-6} - 8t^2 - 8t^{-2}\right)x^3$ $+ \left(24t^7 + 24t^{-7} - 24t^3 - 24t^{-3}\right)x^{7/2} + \left(41t^8 + 41t^{-8} - 24t^4 - 24t^{-4} + 29\right)x^4$ $+ \left(48t^9 + 48t^{-9} - 40t^5 - 40t^{-5} + 32t + 32t^{-1}\right)x^{9/2}$ $+ \left(66t^{10} + 66t^{-10} - 56t^6 - 56t^{-6} + 10t^2 + 10t^{-2} + 16\right)x^5$ $+ \left(107t^{12} + 107t^{-12} - 80t^8 - 80t^{-8} + 8t^4 + 8t^{-4} - 16t^2 - 16t^{-2} - 32\right)x^6$ $+ O\left(x^{13/2}\right)$ |

| $N$ | $\kappa$ | |
|---|---|---|
| 1 | 2 | $\mathcal{I}(\text{CSM}) = 1 + \left(2t^2 + 2t^{-2}\right)x + \left(9t^4 + 9t^{-4}\right)x^2$<br>$\quad + \left(16t^6 + 16t^{-6} - 8t^2 - 8t^{-2}\right)x^3 + \left(35t^8 + 35t^{-8} - 24t^4 - 24t^{-4} + 13\right)x^4$<br>$\quad + \left(54t^{10} + 54t^{-10} - 44t^6 - 44t^{-6} + 24t^2 + 24t^{-2}\right)x^5$<br>$\quad + \left(91t^{12} + 91t^{-12} - 72t^8 - 72t^{-8} - 2t^4 - 2t^{-4} - 48\right)x^6 + O\left(x^7\right)$<br><br>$\text{HS}_{\mathcal{C}}(\text{MQ}_{A/B}) = 1 + 2a + 9a^2 + 16a^3 + 35a^4 + 54a^5 + 91a^6 + O(a^7)$<br>$\quad = \text{HS}_{A/B}(\text{CSM})$ |
| 2 | 1 | $\mathcal{I}(\text{CSM}) = 1 + \left(4t^2 + 4t^{-2}\right)x + \left(4t^3 + 4t^{-3}\right)x^{3/2}$<br>$\quad + \left(18t^4 + 18t^{-4} + 16\right)x^2 + \left(24t^5 + 24t^{-5} + 12t + 12t^{-1}\right)x^{5/2}$<br>$\quad + \left(58t^6 + 58t^{-6} + 36t^2 + 36t^{-2} + 16\right)x^3 + O\left(x^{7/2}\right)$ |
| 2 | 2 | $\mathcal{I}(\text{CSM}) = 1 + \left(2t^2 + 2t^{-2}\right)x + \left(11t^4 + 11t^{-4} + 4\right)x^2$<br>$\quad + \left(28t^6 + 28t^{-6} + 14t^2 + 14t^{-2}\right)x^3 + O\left(x^4\right)$<br><br>$\text{HS}_{\mathcal{C}}(\text{MQ}_{A/B}) = 1 + 2a + 11a^2 + 28a^3 + O(a^4)$<br>$\quad = \text{HS}_{A/B}(\text{CSM})$ |

**Table 5:** Index and Hilbert series for some values of $N$ and $\kappa > 0$ for the example in Figure 34.

## B.6 Circular $U(N)_{+\kappa} \times U(N)_0 \times U(N)_{-\kappa} \times U(N)_0$

Consider the 4-nodes circular $\text{CSM}_{\kappa>0}$ theory in Figure 35a; whose magnetic quiver $\text{MQ}_A \equiv \text{MQ}_B$ is shown in Figure 35b (for $\kappa = 1$, $\text{MQ}_{A/B}$ is the $\mathcal{T}^T$-dual of the CSM theory, which is self-mirror). The Coulomb branch of $\text{MQ}_{A/B}$ is the moduli space of $N$ $\text{SU}(2)$ instantons on $\mathbb{C}^2/\mathbb{Z}_{2\kappa}$ with framing $(0^{2\kappa-1}, 2)$: i.e. a different framing compared to Appendix B.5. The index vs Hilbert series check is shown in Table 6. When $\kappa = 1$ the check is performed on the index itself as the $\mathcal{T}^T$-duality holds, so no Hilbert series are written.

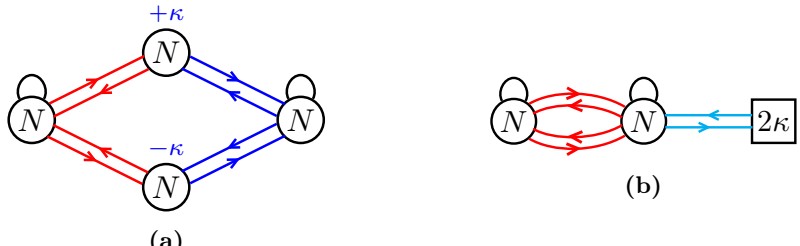

**Figure 35:** (a) The starting $\text{CSM}_{\kappa>0}$ circular theory. (b) The magnetic quiver $\text{MQ}_{A/B}$. For $\kappa = 1$, $\text{MQ}_{A/B}$ is the $\mathcal{T}^T$-dual of the CSM theory.

| $N$ | $\kappa$ | Index and Hilbert series |
|---|---|---|
| 1 | 1 | $\mathcal{I}(\text{CSM}) = 1 + \left(6t^2 + 6t^{-2}\right)x + \left(19t^4 + 19t^{-4} + 4\right)x^2$<br>$\quad + \left(44t^6 + 44t^{-6} - 20t^2 - 20t^{-2}\right)x^3 + \left(85t^8 + 85t^{-8} - 60t^4 - 60t^{-4} + 65\right)x^4$<br>$\quad + \left(146t^{10} + 146t^{-10} - 116t^6 - 116t^{-6} + 64t^2 + 64t^{-2}\right)x^5$<br>$\quad + \left(231t^{12} + 231t^{-12} - 188t^8 - 188t^{-8} + 16t^4 + 16t^{-4} - 188\right)x^6 + O\left(x^7\right)$ |

| N | κ | |
|---|---|---|
| 1 | 2 | $\mathcal{I}(\text{CSM}) = 1 + \left(4t^2 + 4t^{-2}\right)x + \left(11t^4 + 11t^{-4} + 4\right)x^2$ $\quad + \left(24t^6 + 24t^{-6} - 8t^2 - 8t^{-2}\right)x^3 + \left(45t^8 + 45t^{-8} - 28t^4 - 28t^{-4} + 33\right)x^4$ $\quad + \left(76t^{10} + 76t^{-10} - 56t^6 - 56t^{-6} + 32t^2 + 32t^{-2}\right)x^5$ $\quad + \left(119t^{12} + 119t^{-12} - 92t^8 - 92t^{-8} + 8t^4 + 8t^{-4} - 92\right)x^6 + O\left(x^7\right)$ <br><br> $\text{HS}_{A/B}(\text{CSM}) = 1 + 4a + 11a^2 + 24a^3 + 45a^4 + 76a^5 + 119a^6 + O(a^7)$ $= \text{HS}_{\mathcal{C}}(\text{MQ}_{A/B})$ |
| 2 | 1 | $\mathcal{I}(\text{CSM}) = 1 + \left(6t^2 + 6t^{-2}\right)x + \left(35t^4 + 35t^{-4} + 32\right)x^2$ $\quad + \left(131t^6 + 131t^{-6} + 113t^2 + 113t^{-2}\right)x^3 + O\left(x^4\right)$ |
| 2 | 2 | $\mathcal{I}(\text{CSM}) = 1 + \left(4t^2 + 4t^{-2}\right)x + \left(16t^4 + 16t^{-4} + 12\right)x^2$ $\quad + \left(51t^6 + 51t^{-6} + 39t^2 + 39t^{-2}\right)x^3 + O\left(x^4\right)$ <br><br> $\text{HS}_{A/B}(\text{CSM}) = 1 + 4a + 16a^2 + 51a^3 + O(a^4)$ $= \text{HS}_{\mathcal{C}}(\text{MQ}_{A/B})$ |

**Table 6:** Index and Hilbert series for some values of $N$ and $\kappa > 0$ for the example in Figure 35.

## B.7 Circular $\mathbf{U}(N)_{+\kappa} \times \mathbf{U}(N)_0 \times \mathbf{U}(N)_{-\kappa}$

Consider the 3-nodes circular $\text{CSM}_{\kappa>0}$ theory in Figure 36a; whose magnetic quivers $\text{MQ}_A$ and $\text{MQ}_B$ are shown in Figures 36b and 36c respectively (for $\kappa = 1$, $\text{MQ}_{A/B}$ is the $\mathcal{T}^T$-dual of the CSM theory, while $\text{MQ}_B$ is the mirror of the $\mathcal{T}^T$-dual.). It is a specific instance of the generic example 2 discussed in Section 4. The Coulomb branch of $\text{MQ}_A$ is $\text{Sym}^N(\mathbb{C}^2/\mathbb{Z}_{2\kappa})$; while the Coulomb branch of $\text{MQ}_B$ is the moduli space of $N$ SU(2) instantons on $\mathbb{C}^2/\mathbb{Z}_\kappa$ with framing $(0^{\kappa-1}, 2)$. The index vs Hilbert series check is shown in Table 7. When $\kappa = 1$ the check is performed on the index itself as the $\mathcal{T}^T$-duality holds, so no Hilbert series are written.

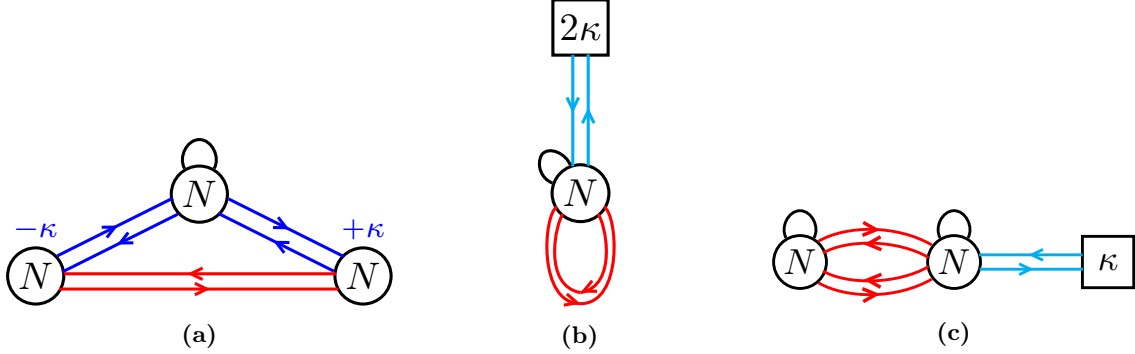

**Figure 36:** (a) The starting $\text{CSM}_\kappa$ circular theory. (b) The magnetic quiver $\text{MQ}_A$. (c) The magnetic quiver $\text{MQ}_B$. For $\kappa = 1$, $\text{MQ}_{A/B}$ is the $\mathcal{T}^T$-dual of the CSM theory, while $\text{MQ}_B$ is the mirror of the $\mathcal{T}^T$-dual.

| N | κ | Index and Hilbert series |
|---|---|---|
| 1 | 1 | $\mathcal{I}(\text{CSM}) = 1 + 2x^{1/2}t + \left(6t^2 + 3t^{-2}\right)x + \left(10t^3 + 4t^{-1}\right)x^{3/2}$ $\quad + \left(19t^4 + 5t^{-4} - 2\right)x^2 + \left(28t^5 - 12t + 4t^{-3}\right)x^{5/2}$ $\quad + \left(44t^6 - 26t^2 + 7t^{-6}\right)x^3 + \left(60t^7 - 44t^3 + 14t^{-1} + 4t^{-5}\right)x^{7/2}$ $\quad + \left(85t^8 - 66t^4 + 9t^{-8} + 34\right)x^4 + O\left(x^{9/2}\right)$ |

$$\mathcal{I}(\mathsf{CSM}) = 1 + \left(6t^2 + t^{-2}\right)x + 4t^{-1}x^{3/2} + \left(19t^4 + 3t^{-4} - 6\right)x^2 + 4tx^{5/2}$$
$$+ \left(44t^6 + 3t^{-6} - 30t^2 + 4t^{-2}\right)x^3 + \left(4t^3 + 4t^{-5}\right)x^{7/2}$$
$$+ \left(85t^8 + 5t^{-8} - 70t^4 - 4t^{-4} + 24\right)x^4 + \left(4t^5 + 16t + 8t^{-3}\right)x^{9/2}$$
$$+ \left(146t^{10} + 5t^{-10} - 126t^6 + 4t^{-6} + 14t^2 - 9t^{-2}\right)x^5$$
$$+ \left(4t^7 + 16t^3 + 4t^{-9} - 36t^{-1}\right)x^{11/2}$$
$$+ \left(231t^{12} + 7t^{-12} - 198t^8 - 4t^{-8} - 34t^4 - 2t^{-4} + 34\right)x^6 + O\left(x^{13/2}\right)$$

$$\mathrm{HS}_A(\mathsf{CSM}) = 1 + a + 3a^2 + 3a^3 + 5a^4 + 5a^5 + 7a^6 + O(a^7)$$
$$= \mathrm{HS}_{\mathcal{C}}(\mathsf{MQ}_A)$$

$$\mathrm{HS}_B(\mathsf{CSM}) = 1 + 6b + 19b^2 + 44b^3 + 85b^4 + 146b^5 + 231b^6 + O(b^7)$$
$$= \mathrm{HS}_{\mathcal{C}}(\mathsf{MQ}_B)$$

(row: 1    2)

---

(row: 2    1)

$$\mathcal{I}(\mathsf{CSM}) = 1 + 2x^{1/2}t + \left(9t^2 + 3t^{-2}\right)x + \left(22t^3 + 10t^{-1}\right)x^{3/2}$$
$$+ \left(55t^4 + 11t^{-4} + 25\right)x^2 + \left(116t^5 + 46t + 26t^{-3}\right)x^{5/2}$$
$$+ \left(242t^6 + 22t^{-6} + 60t^2 + 44t^{-2}\right)x^3 + \left(448t^7 + 34t^3 + 32t^{-1} + 50t^{-5}\right)x^{7/2}$$
$$+ \left(820t^8 - 83t^4 + 58t^{-4} + 45t^{-8} + 14\right)x^4 + O\left(x^{9/2}\right)$$

---

(row: 2    2)

$$\mathcal{I}(\mathsf{CSM}) = 1 + \left(6t^2 + t^{-2}\right)x + 4t^{-1}x^{3/2} + \left(35t^4 + 4t^{-4} + 1\right)x^2$$
$$+ \left(28t + 4t^{-3}\right)x^{5/2} + \left(131t^6 - 22t^2 + 26t^{-2} + 6t^{-6}\right)x^3 + O\left(x^{7/2}\right)$$

$$\mathrm{HS}_A(\mathsf{CSM}) = 1 + a + 4a^2 + 6a^3 + O(a^4)$$
$$= \mathrm{HS}_{\mathcal{C}}(\mathsf{MQ}_A)$$

$$\mathrm{HS}_B(\mathsf{CSM}) = 1 + 6b + 35b^2 + 131b^3 + O(b^4)$$
$$= \mathrm{HS}_{\mathcal{C}}(\mathsf{MQ}_B)$$

**Table 7:** Index and Hilbert series for some values of $N$ and $\kappa > 0$ for the example in Figure 36.

## B.8   Mixing of fugacities

It is now worth analysing a phenomenon which occurred in some of the above studied examples (see Section 2.1 and 2.2, and Appendix B.2, B.3 and B.4). Fugacity maps can and have to mix Abelian and non-Abelian fugacities, see for instance [67]. To appreciate this fact, consider the example in Figure 37, which has the following Coulomb branch isometry algebra

$$\begin{cases} \mathfrak{su}(n-1) \oplus \mathfrak{u}(1), & b \geqslant 1. \\ \mathfrak{su}(n), & b = 0. \end{cases} \tag{B.26}$$

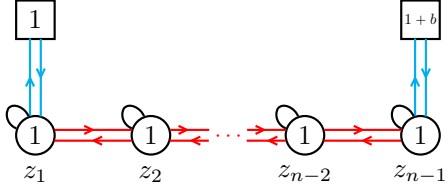

**Figure 37:** Example theory for the mixing between Abelian and non-Abelian fugacities. Here $b \in \mathbb{N}_0$. The Cartans have been written under the respective gauge nodes.

To make that symmetry manifest in the index (or the Coulomb branch Hilbert series),

the following fugacity map is required:

$$\text{for } j \in \{1, \ldots, n-2\}: \quad z_j = \prod_{i=1}^{n-2} x_i^{C_{j,i}^{A_{n-2}}} = \begin{cases} \dfrac{x_1^2}{x_2}, & j = 1 \\[2mm] \dfrac{x_j^2}{x_{j-1}x_{j+1}}, & j \in \{2, \ldots, n-3\} \\[2mm] \dfrac{x_{n-2}^2}{x_{n-3}}, & j = n-2 \end{cases} \tag{B.27}$$

where $C_{j,i}^{A_{n-2}}$ is the Cartan matrix of $\mathfrak{su}(n-1)$. There is, however, still a fugacity map required for $z_{n-1}$. The reason for this is simple: the $b \geqslant 1$ case has to follow from the decomposition of $\mathfrak{su}(n) \rightarrow \mathfrak{su}(n-1) \oplus \mathfrak{u}(1)$

$$[1, 0, \ldots, 0, 1]_{A_{n-1}} \rightarrow [1, 0, \ldots, 0, 1]_{A_{n-2}} \oplus [0, 0, \ldots, 0, 0]_{A_{n-2}} \tag{B.28}$$
$$\oplus [1, 0, \ldots, 0, 0]_{A_{n-2}} \cdot q \oplus [0, 0, \ldots, 0, 1]_{A_{n-2}} \cdot q^{-1}$$

The first line of (B.28) shows that the adjoint of $\mathfrak{su}(n-1)$ appear together with the singlet that gives rise to the Cartan generator of $\mathfrak{u}(1)$. These adjoint representations are realised by monopole operators in the standard fashion [24]. The $\mathfrak{u}(1)$ Cartan comes from the dressing factor (i.e. trivial monopole operator dressed by U(1) Casimir invariants) at the $(n-1)$st gauge node. Likewise, the Cartan elements for $\mathfrak{su}(n-1)$ are realised by the $n-2$ dressing factors of the first $n-2$ gauge nodes. The positive roots of $\mathfrak{su}(n-1)$ are then realised by monopole operators with non-trivial fluxes of the form $\vec{m} \equiv (m_1, m_2, \ldots, m_{n-2}, m_{n-1}) \in \Phi_+$ with $\Phi_+ = \{(1, 0, \ldots, 0, 0), (1, 1, \ldots, 0, 0), \ldots, (1, 1, \ldots, 1, 0)\}$; likewise, the negative roots by $\Phi_- = \{-\vec{m} \mid \vec{m} \in \Phi_+\}$. It is straightforward to verify that all these monopole operators have conformal dimension[17] $2\Delta = 2$.

The bifundamental representations in the second line of (B.28) are realised by monopole operators in the quiver in Figure 37 that have conformal dimension $2\Delta = b + 1$. In fact, their fluxes are $\Sigma_\pm = \{\pm(0, 0, \ldots, 0, 0, 1), \pm(0, 0, \ldots, 0, 1, 1), \ldots, \pm(0, 1, \ldots, 1, 1, 1)\}$, which give rise to exactly $n-1$ non-trivial monopole operators charged as $z_{n-1}^{+1}$ and $n-1$ monopole operators charged as $z_{n-1}^{-1}$ at order $2\Delta = b + 1$. Since the case $b = 0$ leads to symmetry enhancement $\mathfrak{su}(n-1) \oplus \mathfrak{u}(1) \rightarrow \mathfrak{su}(n)$, one knows that the fugacity map for $z_{n-1}$ needs to be given by the Cartan matrix of $A_{n-1}$. This leads to

$$z_{n-1} \rightarrow \frac{q}{x_{n-2}}, \tag{B.29}$$

which is a mixing of non-Abelian and Abelian fugacities. For the case $b = 0$, one simply replaces $q \rightarrow x_{n-1}^2$ and finds representations of $\mathfrak{su}(n)$.

## C   The dualisation algorithm

This appendix provides the machinery needed to perform all the dualisations described in the main text. Such a tool goes under the name of *dualisation algorithm* [26–28] and it constitutes a purely field theoretic approach to SL(2, $\mathbb{Z}$) dualities for 3d $\mathcal{N} = 4$ theories. The basic idea of the algorithm goes as follows. First freeze the gauge integration of the nodes so that the theory factorises into its field theory constituents, dubbed as *QFT blocks*. Then dualise each block by means of a set of basic duality moves and glue back the pieces[18].

---

[17]Compared to [38], where $\Delta$ is half-integer valued (as it labels $\mathfrak{su}(2)$ representations), it is convenient to work with the integer valued $2 \cdot \Delta$.

[18]Gluing two U($N$) flavour nodes means to gauge a diagonal combination thereof. Moreover, for the new gauge node thus generated one has to add a $\mathcal{N} = 2$ adjoint chiral field which couples in the superpotential with the moment map operators of the glued blocks.

In the following, the ingredients to perform the algorithm are introduced. However, many subtleties are not included (such as contact terms and singlets, which are omitted as they do not play any role in this paper). For all the details, the reader is referred to [28].

As a final comment, the dualisation algorithm presented here acts on $3d$ $\mathcal{N} = 4$ good theories. However, it was originally introduced in the context of $4d$ $\mathcal{N} = 1$ theories [26–28]. Moreover recently it has been enlarged to $3d$ $\mathcal{N} = 4$ and $4d$ $\mathcal{N} = 1$ bad theories [68, 69] and to $3d$ $\mathcal{N} = 2$ dualities as well [70, 71].

## C.1 $\mathrm{SL}(2, \mathbb{Z})$ operators

The standard generators for the $\mathrm{SL}(2, \mathbb{Z})$ duality group are called $\mathcal{S}$ and $\mathcal{T}$. The operator $\mathcal{S}$ satisfies

$$\mathcal{S}^{-1} = -\mathcal{S} \,. \tag{C.1}$$

and together with $\mathcal{T}$ it also enjoys the property

$$(\mathcal{S}\mathcal{T})^3 = \mathbb{1} \,. \tag{C.2}$$

Therefore $\mathcal{T}$ can be rewritten as

$$\mathcal{T} = (\mathcal{S}\mathcal{T}\mathcal{S}\mathcal{T}\mathcal{S})^{-1} \,. \tag{C.3}$$

Moreover one can write the transpose of $\mathcal{T}$ as

$$\mathcal{T}^T = \mathcal{T}\mathcal{S}^{-1}\mathcal{T} = -\mathcal{S}\mathcal{T}^{-1}\mathcal{S} \,, \qquad (\mathcal{T}^T)^{-1} = \mathcal{T}^{-1}\mathcal{S}\mathcal{T}^{-1} = -\mathcal{S}\mathcal{T}\mathcal{S} \,. \tag{C.4}$$

In terms of the D3 brane world-volume theory's coupling constant $\tau$, the $\mathrm{SL}(2, \mathbb{Z})$ generators act as

$$\mathcal{S} : \tau \to \frac{1}{\tau} \,, \qquad \mathcal{T} : \tau \to \tau - 1 \,, \tag{C.5}$$

and therefore $\mathcal{T}^T$ acts as

$$\mathcal{T}^T : \tau \to \frac{\tau}{1 - \tau} \,. \tag{C.6}$$

One can also provide a QFT realisation of these operators as duality walls. In particular, $\mathcal{S}$ is realised by the $FT[\mathrm{U}(N)]$ theory [72][19], represented on the left of Figure 38 as a dashed line, which displays on its two sides its enhanced global symmetry $\mathrm{U}(N)_X \times \mathrm{U}(N)_Y$ (the sign $\pm$ distinguishes between $\mathcal{S}$ and $\mathcal{S}^{-1}$)[20]. This object, often referred to as the $\mathcal{S}$-wall, can be also made asymmetric by breaking one of its two $\mathrm{U}(N)$'s down to $\mathrm{U}(M) \times \mathrm{U}(1)_V$ with $M < N$ [26, 28, 73]: this asymmetric $\mathcal{S}$-wall is represented on the right of Figure 38.

**Figure 38:** The $\mathcal{S}^{\pm 1}$ operator on the left and its asymmetric version on the right.

---

[19]The $FT[\mathrm{U}(N)]$ theory is the $T[\mathrm{U}(N)]$ theory [24] with the addition of some extra singlets [72].
[20]If one indicates the enhanced global symmetry $\mathrm{U}(N)_X \times \mathrm{U}(N)_Y$ Cartans' dependence of the corresponding partition functions, the correct definition of the $\mathcal{S}$-wall is

$$\mathcal{S}(\vec{X}, \vec{Y}) = FT[\mathrm{U}(N)](\vec{X}, -\vec{Y}) = FT[\mathrm{U}(N)](-\vec{X}, \vec{Y}) \,, \qquad \mathcal{S}^{-1}(\vec{X}, \vec{Y}) = \mathcal{S}(\vec{X}, -\vec{Y}) = \mathcal{S}(-\vec{X}, \vec{Y}) \,. \tag{C.7}$$

A remarkable property of the $\mathcal{S}$-wall comes from (C.1) together with $\mathcal{S}\mathcal{S}^{-1} = \mathbb{1}$: When gluing together two copies of this object (of opposite sign) one gets a theory whose partition function is a Dirac delta, identifying the Cartans of the external flavour groups. This object, known as *Identity-wall* $\mathbb{I}$ [26], is depicted in Figure 39 together with its asymmetric version, which identifies $M$ Cartans $W_{1,\ldots,M}$ of the smaller flavour group $\mathrm{U}(M)$ with $M$ Cartans $Z_{1,\ldots,M}$ of the larger $\mathrm{U}(N)$, while the remaining $N-M$ Cartans $Z_{M+1,\ldots,N}$ of $\mathrm{U}(N)$ are specialised to the fugacity $V$ of the $\mathrm{U}(1)_V$. For the details, the reader is referred to [28].

**Figure 39:** The symmetric Identity-wall on the left and its asymmetric version on the right.

On the other hand, the QFT realisation of the $\mathcal{T}$-wall is the insertion of a CS-level, which can be represented as in Figure 40. Its inverse, not drawn in the picture, can be constructed inverting the property (C.3).

Finally, the $\mathcal{T}^T$-wall can be realised in QFT by combining the above introduced $\mathcal{S}$- and $\mathcal{T}$-wall as dictated by the property (C.4).

**Figure 40:** The $\mathcal{T}$ operator.

## C.2 QFT blocks

The dualisation algorithm relies on the fact that, by freezing the gauge integrations, one can break the field content of a theory down to a collection of basic contributions, known as QFT blocks. As the family of $3d$ $\mathcal{N} = 4$ theories on which this paper focuses can be realised by Hanany–Witten brane setups, one can put these QFT blocks in a one-to-one correspondence with the elements of the brane system. In particular one can define the following QFT blocks (see Figure 41):

- the bifundamental block, realised by a NS5-brane ( | );

- the fundamental flavour block, realised by a D5-brane ( $\otimes$ );

- the bifundamental block with CS-level $\kappa \in \mathbb{Z}$, realised by a $(1, \kappa)$ 5-brane ( ⫶ ).

In principle, when reading off the quiver from a brane system, the QFT blocks of which it is composed have axial charge as dictated by the conventions of Table 4. However, the aim of this section is to write exact IR dualities, and therefore the fields have twisted or not-twisted axial charge depending on which side of the equation they appear. Therefore the axial charge in this section is written close to the fields, indicating with h. and tw.h. the hypermultiplets and twisted hypermultiplets respectively.

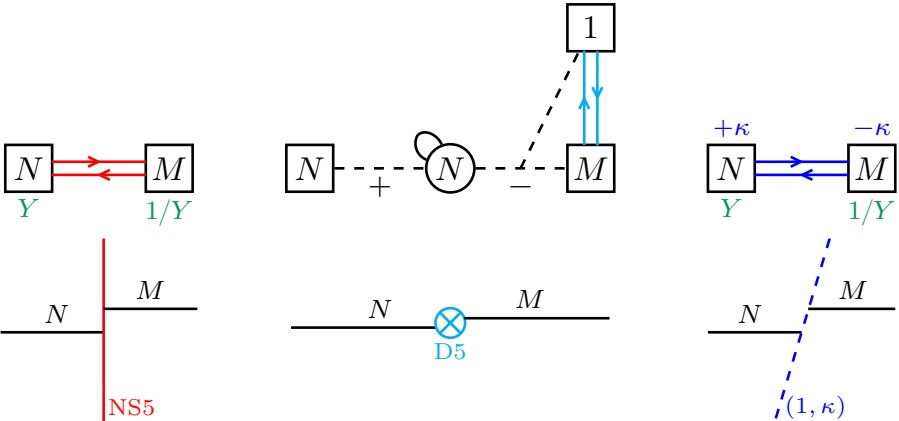

**Figure 41:** From left to right: the bifundamental block, the fundamental flavour block and the bifundamental block with CS-level $\kappa \in \mathbb{Z}$, together with their color coded 5-brane realisation. D3 branes are represented as black horizontal segments with a label denoting their multiplicity. The topological fugacities have been written in green under the nodes while the CS-levels are written in blue over the nodes. As per the fundamental flavour block in the middle, the most generic case is depicted, namely when $N \neq M$ (in particular, $N > M$), hence the asymmetric Identity-wall. When instead $N = M$, one has the same structure but with a symmetric Identity-wall.

## C.3   The strategy

The idea underlying the dualisation algorithm is that the action of an $\mathrm{SL}(2, \mathbb{Z})$ transformation can be implemented on a theory in a local manner, mimicking the local effect that the $\mathrm{SL}(2, \mathbb{Z})$ S-duality group has on type IIB branes. Thus one can consider the QFT blocks and dualise each one of them locally using a set of duality moves[21], which are presented in Figure 43. The algorithm then reads as follows:

1. Given a theory, break it down into the QFT blocks in Figure 41.

2. Dualise each block by means of the basic duality moves in Figure 43.

3. Glue back the dualised pieces.

4. If any VEV is turned on, follow the RG-flow by applying the duality in Figure 42, which is the QFT realisation of the Hanany–Witten (HW) transition [22].

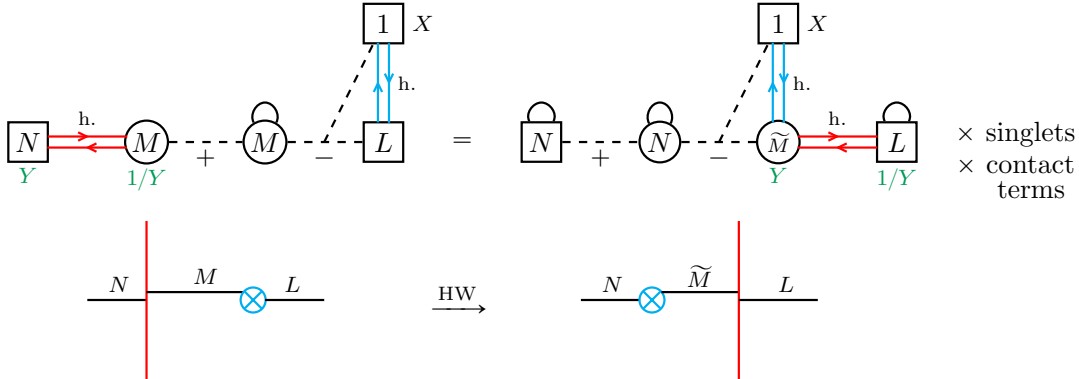

**Figure 42:** The duality realising the HW transition: a bifundamental block is swapped with a fundamental flavour block. This affects the rank $M$, which becomes $\widetilde{M} = N + L - M + 1$, with $N \geqslant \widetilde{M} \geqslant 0$. In the lower part of the figure the HW move is written in the brane language.

---

[21]Their proof boils down to an iterative application of Aharony duality [74]. See [28] for more details.

**$\mathcal{S}$-duality moves**

**(a)** A bifundamental block is $\mathcal{S}$-dualised into a fundamental flavour block.

**(b)** A flavour block is $\mathcal{S}$-dualised into a bifundamental block.

**(c)** A bifundamental block with CS-levels $\kappa_i = \pm 1$ is $\mathcal{S}$-dualised into itself but with swapped CS-levels (namely $\kappa_i = \mp 1$).

**$\mathcal{T}$-duality moves**

**(d)** A bifundamental block is $\mathcal{T}$-dualised into a bifundamental block with CS-levels $\kappa_i = \mp 1$.

**(e)** A flavour block is $\mathcal{T}$-dualised into itself.

**(f)** A bifundamental block with CS-levels $\pm \kappa$ is $\mathcal{T}$-dualised into a bifundamental block with CS-levels $\pm(\kappa-1)$. If $|\kappa| = 1$, the result of the $\mathcal{T}$-dualisation is a bifundamental block without any CS-level.

**$\mathcal{T}^T$-duality moves**

**(g)** A bifundamental block is $\mathcal{T}^T$-dualised into itself.

**(h)** A flavour block is $\mathcal{T}^T$-dualised into a bifundamental block with CS-levels $\kappa_i = \mp 1$.

**(i)** A bifundamental block with CS-levels $\kappa_i = \pm 1$ is $\mathcal{T}^T$-dualised into a flavour block.

**Figure 43:** The $\mathcal{S}$, $\mathcal{T}$ and $\mathcal{T}^T$ basic duality moves.

In the following pages the basic duality moves for the $\mathcal{S}$, $\mathcal{T}$ and $\mathcal{T}^T$ operators are explained, together with a dualisation example for each $SL(2,\mathbb{Z})$ transformation.

As a final remark, the above algorithm works for good theories [24] only. Its generalisation to bad theories has been discussed in [68, 69], but it is not reviewed here as it is not the scope of the present paper to work with this kind of models.

## C.4   The $\mathcal{S}$-dualisation

The action of $\mathcal{S}$-duality on the blocks is essentially to swap the bifundamental and the fundamental flavour blocks, while mapping the CS $|\kappa| = 1$ bifundamental block into itself. The basic $\mathcal{S}$-duality moves are depicted in Figures 43a, 43b and 43c, where the $\mathcal{S}$ operator is written just as a symbol and not as its QFT realisation to make the pictures possibly more readable. Moreover, singlets and contact terms are not shown explicitly but just written as words. All the details can be found in [28].

An explicit example of $\mathcal{S}$-duality is presented in Figure 44 and derived in Figure 45 (in a slightly schematic way, as singlets and contact terms are ignored).

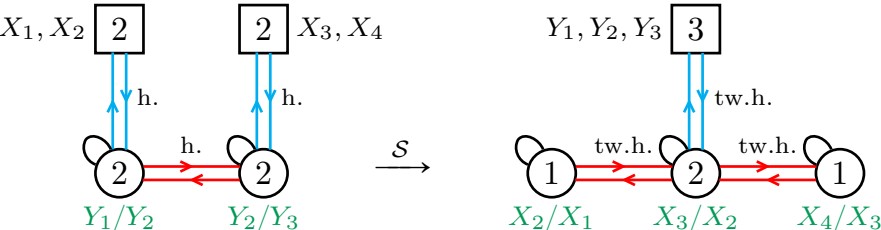

**Figure 44:** The $\mathcal{S}$-dual pair.

The algorithmic dualisation goes as follows.

0. Starting from the theory on top of Figure 45, first add two trivial flavour nodes on the sides with the corresponding bifundamentals (which are realised by the NS5 branes on the extremities of the brane system).

1. Then, freeze the gauge integrations and break the quiver down into basic QFT blocks.

2. Now apply the $\mathcal{S}$ duality moves 43a and 43b to the blocks. Notice that the fourth dualised block, which is a fundamental flavour, comes with a symmetric Identity-wall which has been written just as $\mathbb{I}$ for convenience.

3. Now glue back together the dualised pieces, noticing that the external $\mathcal{S}$ and $\mathcal{S}^{-1}$ operators are trivial, and that inside the quiver the combinations $\mathcal{S}\mathcal{S}^{-1} = \mathbb{1}$ appear.

4. Now, in order to get rid of the asymmetric Identity-walls on the sides (as they signal the presence of a VEV which has to be extinguished in order to reach the end of the RG flow), apply the HW move 42 on the sides of the quiver. Two iterations (steps 4.1 and 4.2 in Figure 45) are needed before the asymmetric Identity-walls turn into symmetric ones and thus by definition collapse into a single U(2) node with three flavours.

5. After removing the trivial fields on the sides of the quiver, the last quiver in Figure 45 is reached. The final outcome is indeed the duality in Figure 44.

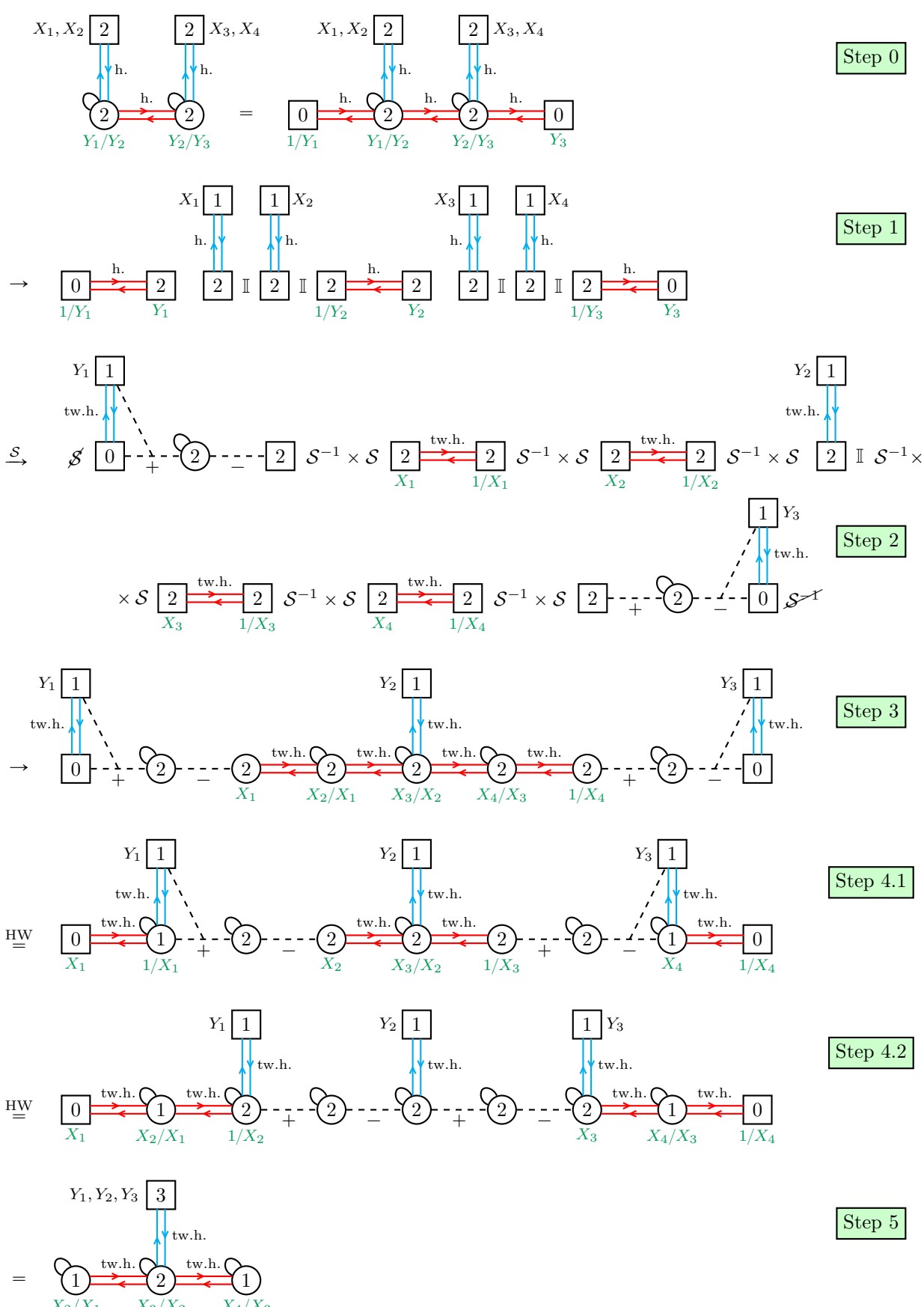

**Figure 45:** The $\mathcal{S}$-dualisation algorithm at work.

## C.5  The $\mathcal{T}$-dualisation

The action of $\mathcal{T}$-duality on the blocks is essentially to add a CS-level to the bifundamental block and to decrease by 1 the level $\kappa$ of the CS bifundamental block, while mapping the flavour block into itself. The basic $\mathcal{T}$-duality moves are depicted in Figures 43d, 43e and 43f, where the $\mathcal{T}$ operator is written just as a symbol and not as its QFT realisation to make the pictures possibly more readable. Moreover, singlets and contact terms are not shown explicitly but just written as words. All the details can be found in [28].

An explicit example of $\mathcal{T}$-duality is presented in Figure 46 and derived in Figure 47 (in a slightly schematic way, as singlets and contact terms are ignored). In particular, the dual theories considered are transformed one into the other by applying $\mathcal{T}$ twice, namely by the operator $(\mathcal{T})^2$. The reason why this example has been chosen is that it is of particular interest for the main text (see Appendix B.1).

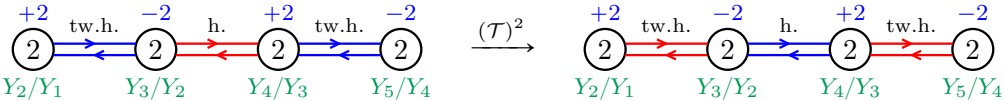

**Figure 46:** The $(\mathcal{T})^2$-dual pair.

**Figure 47:** The $\mathcal{T}$-dualisation algorithm at work.

The algorithmic dualisation goes as follows.

0. Starting from the theory on top of Figure 47, first add two trivial flavour nodes on the sides with the corresponding bifundamentals (which are realised by the NS5 branes on the extremities of the brane system).

1. Then, freeze the gauge integrations and break the quiver down into basic QFT blocks.

2. Now apply the $\mathcal{T}$ duality moves 43d and 43f to the blocks. Since the transformation to be performed is $(\mathcal{T})^2$, iterate this procedure twice (steps 2.1 and 2.2 in Figure 47).

3. Now glue back together the dualised pieces, noticing that the external $\mathcal{T}$ and $(\mathcal{T})^{-1}$ operators are trivial, and that inside the quiver the combinations $\mathcal{T}(\mathcal{T})^{-1} = \mathbb{1}$ appear.

4. At this point no VEVs are turned on, so one has already obtained the result. The final outcome is the duality in Figure 46.

## C.6   The $\mathcal{T}^T$-dualisation

The reason why the operator $\mathcal{T}^T$ is interesting is that it allows for the dualisation of a theory with CS-levels $\kappa_i = \pm 1$ into a theory without any CS-level. In particular, the action of $\mathcal{T}^T$-duality on the blocks is essentially to swap the CS $|\kappa| = 1$ bifundamental block and the fundamental flavour block, while mapping the bifundamental block into itself. The basic $\mathcal{T}^T$-duality moves are depicted in Figures 43g, 43h and 43i, where the $\mathcal{T}^T$ operator is written just as a symbol and not as its QFT realisation to make the pictures possibly more readable. Moreover, singlets and contact terms are not shown explicitly but just written as words. All the details can be found in [28].

An explicit example of $\mathcal{T}^T$-duality is presented in Figure 48 and derived in Figure 49 (in a slightly schematic way, as singlets and contact terms are ignored).

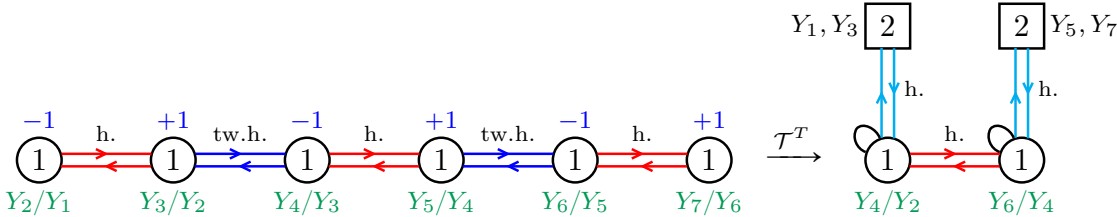

**Figure 48:** The $\mathcal{T}^T$-dual pair.

The algorithmic dualisation goes as follows.

0. Starting from the theory on top of Figure 49, first add two trivial flavour nodes on the sides with the corresponding bifundamentals (which are realised by the $(1,1)$ 5-branes on the extremities of the brane system).

1. Then, freeze the gauge integrations and break the quiver down into basic QFT blocks.

2. Now apply the $\mathcal{T}^T$ duality moves 43g and 43i to the blocks. Notice that the third and fifth dualised blocks, which are fundamental flavours, come with a symmetric Identity-wall which has been written just as $\mathbb{I}$ for convenience.

3. Now glue back together the dualised pieces, noticing that the external $\mathcal{T}^T$ and $(\mathcal{T}^T)^{-1}$ operators are trivial, and that inside the quiver the combinations $\mathcal{T}^T(\mathcal{T}^T)^{-1} = \mathbb{1}$ appear.

4. Now, in order to get rid of the asymmetric Identity-walls on the sides (as they signal the presence of a VEV which has to be extinguished in order to reach the end of the RG flow), apply the HW move 42 on the sides of the quiver. One iteration is needed to turn the asymmetric Identity-walls into symmetric ones.

5. After removing the trivial fields on the sides of the quiver, the last quiver in Figure 49 is reached. The final outcome is indeed the duality in Figure 48.

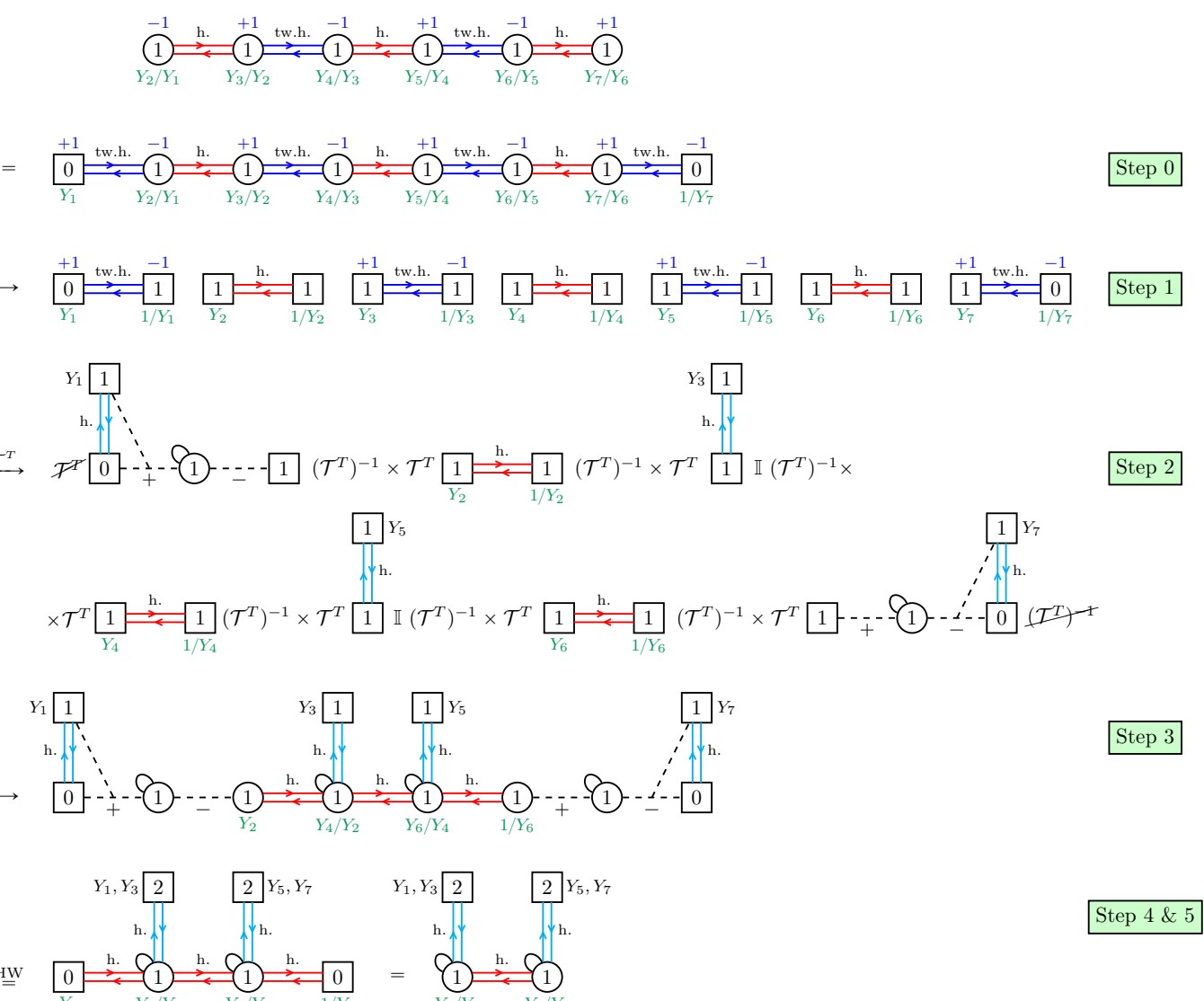

**Figure 49:** The $\mathcal{T}^T$-dualisation algorithm at work.

## C.7 CSM$_{|\kappa|>1}$ theories have no Lagrangian non-CS SL(2, $\mathbb{Z}$) dual

One might ask whether the $\mathcal{T}^T$ dualisation can provide a way to obtain a non-CS dual starting from a CS quiver having $|\kappa_i| = |\kappa| > 1$ ($\kappa \in \mathbb{Z}$). In this case the logic discussed so far can still be applied by looking at the CS-levels $\kappa_i$ as the insertion of $\kappa_i$ $\mathcal{T}$-walls for $\kappa_i > 1$ (or $\mathcal{T}^{-1}$-walls for $\kappa_i < 1$) in the theory. However, the result of the SL(2, $\mathbb{Z}$) dualisation is a quiver with some links given by $\mathcal{S}$-walls, which do not come in pairs and hence do not recombine into Identity-walls: this means that the dual is non-Lagrangian. The bottom line is that for CSM$_{|\kappa|>1}$ theories the duality group SL(2, $\mathbb{Z}$) cannot provide a Lagrangian dual having no CS-levels. This statement can be made more precise as follows.

**Proposition.** A Chern–Simons matter theory with levels $|\kappa_i| = |\kappa| > 1$ ($\kappa \in \mathbb{Z}$) cannot have a Lagrangian non-CS $\mathrm{SL}(2,\mathbb{Z})$-dual.

**Proof.** So far the field theory realisations of the $\mathrm{SL}(2,\mathbb{Z})$ operators has been considered. However, for this proof it is convenient to employ the language in which the $\mathrm{SL}(2,\mathbb{Z})$ transformation are written as matrices acting on column vectors representing the branes [18]. In particular, denoting the $(p,q)$-branes as $\binom{p}{q}$, one can write

$$
\mathrm{NS5} = \begin{pmatrix} \pm 1 \\ 0 \end{pmatrix}, \qquad \mathrm{D5} = \begin{pmatrix} 0 \\ \pm 1 \end{pmatrix}, \tag{C.8}
$$

where the $\pm$ is the sign of the FI and of the flavour Cartan, respectively, of the QFT blocks realised by these branes. Therefore, the $\mathrm{SL}(2,\mathbb{Z})$ operators

$$
\mathcal{S} : (p,q) \to (q,-p), \qquad \mathcal{T} : (p,q) \to (p, q-p), \qquad \mathcal{T}^T : (p,q) \to (p-q, q), \tag{C.9}
$$

can be represented as the following matrices:

$$
\mathcal{S} = \begin{pmatrix} 0 & -1 \\ 1 & 0 \end{pmatrix}, \qquad \mathcal{T} = \begin{pmatrix} 1 & 0 \\ 1 & 1 \end{pmatrix}, \qquad \mathcal{T}^T = \begin{pmatrix} 1 & 1 \\ 0 & 1 \end{pmatrix}. \tag{C.10}
$$

In this language, one says that the brane $X$ is dualised into the brane $Y$ by the transformation $\mathcal{V}$ if $X = \mathcal{V}Y$.

Now, coming back to the proof for the above proposition, consider a generic transformation $\mathcal{V} \in \mathrm{SL}(2,\mathbb{Z})$ parametrised as

$$
\mathcal{V} = \begin{pmatrix} a & b \\ c & d \end{pmatrix}, \quad \text{with } ad - bc = 1 \text{ and } a,b,c,d \in \mathbb{Z}. \tag{C.11}
$$

A CSM theory with levels $|\kappa_i| > 1$ is realised by NS5-branes and $(1, \kappa_i)$-branes. In order to have a $\mathcal{V}$-dual theory with no CS-levels one needs $\mathcal{V}$ to transform the branes of the starting setup into NS5 and/or D5. This is the case if one of the following situations occurs.

- $\mathcal{V}$ dualises $(1, \kappa_i)$-branes into NS5-branes and NS5-branes into NS5-branes, namely

$$
\begin{pmatrix} 1 \\ \kappa_i \end{pmatrix} = \mathcal{V} \begin{pmatrix} \pm 1 \\ 0 \end{pmatrix} \implies \begin{cases} a = \pm 1, \\ c = \pm \kappa_i, \\ \forall b, d, \end{cases} \qquad \begin{pmatrix} 1 \\ 0 \end{pmatrix} = \mathcal{V} \begin{pmatrix} \pm 1 \\ 0 \end{pmatrix} \implies \begin{cases} a = \pm 1, \\ c = 0, \\ \forall b, d. \end{cases} \tag{C.12}
$$

- $\mathcal{V}$ dualises $(1, \kappa_i)$-branes into NS5-branes and NS5-branes into D5-branes, namely

$$
\begin{pmatrix} 1 \\ \kappa_i \end{pmatrix} = \mathcal{V} \begin{pmatrix} \pm 1 \\ 0 \end{pmatrix} \implies \begin{cases} a = \pm 1, \\ c = \pm \kappa_i, \\ \forall b, d, \end{cases} \qquad \begin{pmatrix} 1 \\ 0 \end{pmatrix} = \mathcal{V} \begin{pmatrix} 0 \\ \pm 1 \end{pmatrix} \implies \begin{cases} b = \pm 1, \\ d = 0, \\ \forall a, c. \end{cases} \tag{C.13}
$$

- $\mathcal{V}$ dualises $(1, \kappa_i)$-branes into D5-branes and NS5-branes into NS5-branes, namely

$$
\begin{pmatrix} 1 \\ \kappa_i \end{pmatrix} = \mathcal{V} \begin{pmatrix} 0 \\ \pm 1 \end{pmatrix} \implies \begin{cases} b = \pm 1, \\ d = \pm \kappa_i, \\ \forall a, c, \end{cases} \qquad \begin{pmatrix} 1 \\ 0 \end{pmatrix} = \mathcal{V} \begin{pmatrix} \pm 1 \\ 0 \end{pmatrix} \implies \begin{cases} a = \pm 1, \\ c = 0, \\ \forall b, d. \end{cases} \tag{C.14}
$$

- $\mathcal{V}$ dualises $(1, \kappa_i)$-branes into D5-branes and NS5-branes into D5-branes, namely

$$
\begin{pmatrix} 1 \\ \kappa_i \end{pmatrix} = \mathcal{V} \begin{pmatrix} 0 \\ \pm 1 \end{pmatrix} \implies \begin{cases} b = \pm 1, \\ d = \pm \kappa_i, \\ \forall a, c, \end{cases} \qquad \begin{pmatrix} 1 \\ 0 \end{pmatrix} = \mathcal{V} \begin{pmatrix} 0 \\ \pm 1 \end{pmatrix} \implies \begin{cases} b = \pm 1, \\ d = 0, \\ \forall a, c. \end{cases} \tag{C.15}
$$

None of these systems of equations, together with the conditions in (C.11) and $|\kappa_i| > 1$, admit solutions. The same holds if one chooses to start from a $\left(\begin{smallmatrix} -1 \\ \kappa_i \end{smallmatrix}\right)$-brane and/or from a $\left(\begin{smallmatrix} -1 \\ 0 \end{smallmatrix}\right)$-brane. This concludes the proof of the proposition.

On the other hand, allowing for $|\kappa_i| = 1$, one finds that the systems of equations (C.13) have solutions, which are

$$\mathcal{V} = \pm \begin{pmatrix} -1 & -1 \\ +1 & 0 \end{pmatrix} = \pm(\mathcal{T}^T)^{-1}\mathcal{S}, \quad \text{for } \kappa_i = +1\,, \tag{C.16a}$$

$$\mathcal{V} = \pm \begin{pmatrix} +1 & -1 \\ +1 & 0 \end{pmatrix} = \pm\mathcal{T}^T\mathcal{S}, \qquad \text{for } \kappa_i = -1\,. \tag{C.16b}$$

Also the systems of equations (C.14) have solutions, which are

$$\mathcal{V} = \pm \begin{pmatrix} +1 & +1 \\ 0 & +1 \end{pmatrix} = \pm\mathcal{T}^T, \qquad \text{for } \kappa_i = +1\,, \tag{C.17a}$$

$$\mathcal{V} = \pm \begin{pmatrix} +1 & -1 \\ 0 & +1 \end{pmatrix} = \pm(\mathcal{T}^T)^{-1}, \quad \text{for } \kappa_i = -1\,. \tag{C.17b}$$

In particular, the solution (C.17a) explains the choice made throughout Section 2, namely to $\mathcal{T}^T$-dualise CSM theories realised by $(1,1)$ 5-branes: indeed it transforms $(1,1)$ 5-branes into D5-branes and it leaves NS5-branes untouched. On the other hand, the solution (C.16a) combines the action of $\mathcal{S}$ and $\mathcal{T}^T$ and indeed shows up in the SL$(2,\mathbb{Z})$ duality web in Figure 2.

# D  The Giveon–Kutasov dualisation

Taking into account again the dual pair in Figure 48, one can consider an alternative version of the electric theory on the left, which can be obtained by applying the Giveon–Kutasov (GK) duality [51] on the sides of the quiver. This duality, depicted in Figure 50 from the point of view of both QFT and the brane system, is the analogous of the Hanany–Witten transition for quivers with CS-levels, or equivalently for brane systems with $(1, \kappa > 0)$ 5-branes [18].

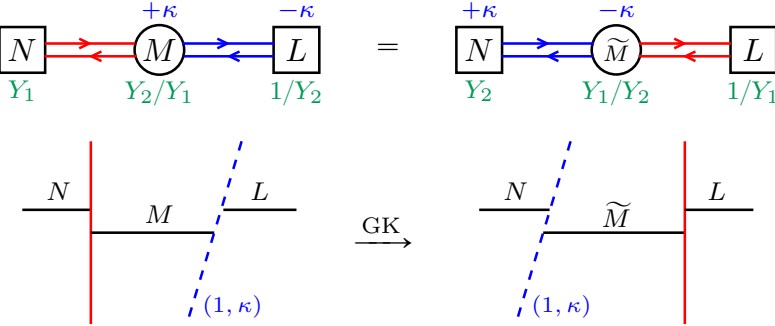

**Figure 50:** The GK transition. On top its QFT realisation is represented: a bifundamental block with CS-levels $\pm\kappa$ is swapped with a bifundamental block of twisted axial charge. Doing this swap, the central rank $M$ is changed to $\widetilde{M} = N + L - M + |k|$, with $N \geqslant \widetilde{M} \geqslant 0$. This is reflected by the brane system on the bottom.

Focusing on the $\kappa = 1$ case, the theory that one gets after applying GK to the electric quiver in Figure 48 is shown in Figure 51, and by computing with the dualisation algorithm its $\mathcal{T}^T$-dual one can find again the magnetic theory on the right of Figure 48. In particular, when $\mathcal{T}^T$-dualising the theory that has undergone GK transitions, one immediately finds

the correct dual without the need for any HW move. This is indeed clear by looking at the brane systems in Figure 52. Therefore, one can think of the GK transition for NS5 and $(1,1)$ 5-branes as the $\mathcal{T}^T$-analogous of the HW transition for NS5 and D5 branes.

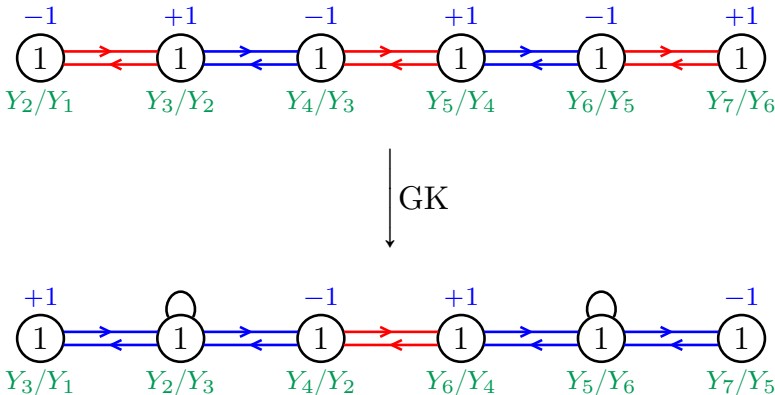

**Figure 51:** The GK move applied to the left quiver in Figure 48.

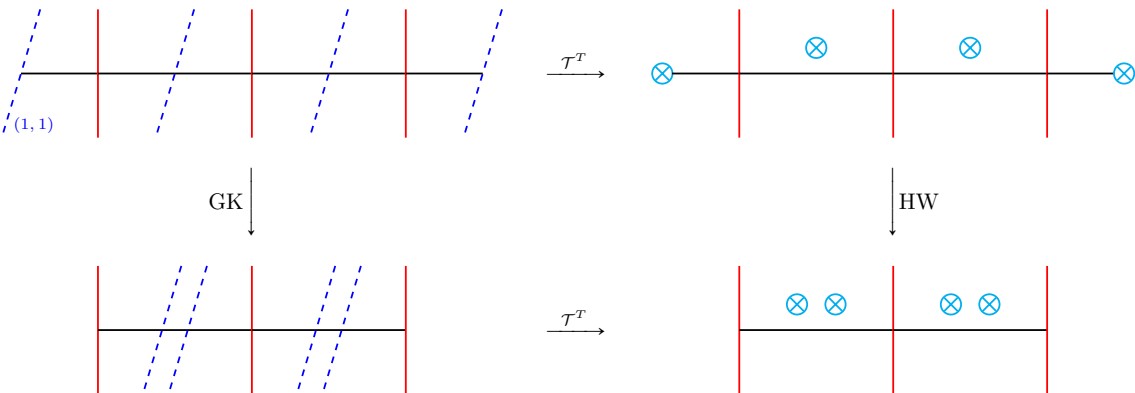

**Figure 52:** The GK transition for NS5 and $(1,1)$ 5-branes is the $\mathcal{T}^T$-analogous of the HW move.

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
