# Peer review of "Vacua, Symmetries, and Higgsing of Chern-Simons Matter Theories"

_SciPost Physics, doi:SciPost Phys. 18, 174 (2025)_

## Round 1 · Referee Report · Anonymous (Referee 1) · 2025-4-10

Report

The paper discusses a novel approach to characterize the maximal branches of the moduli spaces of vacua of 3d $\mathcal{N}=4$ Chern-Simons-matter theories.
Early examples where analyzed by Jafferis-Yin and Hosomichi-Lee-Lee-Lee-Park. However, due to the inherent difficulties of a pure field-theoretic analysis, most recent checks focused on the study of Hilbert series, plus (sporadic) examples of sphere partition functions for suitably chosen Chern-Simons levels.
The combination of brane setups, magnetic quiver and Hilbert series dualization employed in the paper allows for a more systematic approach. In particular, for theories that are dual to 3d N=4 quivers without CS couplings, the authors construct the Hasse diagram of the two maximal branches thanks to a detailed analysis of the duality map, combined with the more standard techniques to analyze the moduli spaces of vacua without CS terms. The authors observe from the brane setup that certain monopole operators must have equal fluxes w.r.t gauge nodes between the two NS5-branes, which is a neat and efficient way to uncover this feature, that would be hard to systematize by constructing gauge-invariant monopole operators ``by hand''.
For theories which do not admit a description without CS couplings, this work proposes a magnetic quiver. The proposal is motivated by a number of explicit examples. The authors further extend the magnetic quiver proposal to theories with only $\mathcal{N}=3$ supersymmetry, one magnetic quiver per maximal branch, and successfully check their proposal against earlier work of Assel. However, in this less supersymmetric case the authors only discusses Abelian examples.
The content is innovative and interesting for theoretical physicists working in the field; it may also be of interest to mathematicians working on the foundations of 3d $\mathcal{N}=4$ Coulomb branches. The paper is clearly written, the necessary steps are explained in detail but in a neat and not verbose manner. Besides the main claims about moduli spaces of vacua of CS-matter theories, the paper includes various additional interesting results, such as the explicit calculations in Appendix B and the no-go theorem in Appendix C.7.
Overall, I consider that the content and presentation of the work meet the acceptance criteria. I strongly recommend the paper be published in SciPost Physics, provided some minor clarifications are incorporated. Please refer to the attached pdf file for a list of small requested changes

Requested changes

Main requested changes: - In the computation of the superconformal index, it would be good to explain how the sum over magnetic fluxes is dealt with. For example in Eq.(2.6) the authors compute the index perturbatively in $x$, thus each term should include a sum over topological sectors from the value of the gauge curvature on $S^2$ (e.g. the sum over $l$ in [Ref. 52, Eq.(2.1)]). What is the fugacity associated to such summation (e.g. the analogue of the fugacity $w$ in [Ref. 52, Eq.(2.1)])? Are the results at each order in $x$ a resummation of the contribution from (infinitely many) topological sectors? Or do only finitely many values of the sum contribute at each order? The same applies to the other computations of the index. I would expect that, for $p$ unitary gauge nodes, each order in $x$ fixes a linear combination of the integers $l_i$ ($i=1, \dots, p$) and there are $p-1$ remaining series to resum.

  • Relatedly, the authors exchange the perturbative expansion in $x$ with the infinite sum over topological sectors. Do the authors also exchange the sum over topological sectors with the residue integral? If so, they should explain why this is justified.

For other minor corrections please refer to the attached pdf.

Attachment

Recommendation

Ask for minor revision

  • validity: top
  • significance: high
  • originality: good
  • clarity: top
  • formatting: excellent
  • grammar: perfect

Author:  Marcus Sperling  on 2025-04-25  [id 5415]

(in reply to Report 1 on 2025-04-10)

We thank the referee for thorough and careful reading of our manuscript and the precise queries.

We reply to the major points raised in turn: The superconformal index has a well-established representation as a sum over magnetic fluxes wherein the summand is a contour integral over the gauge variables. The integrand contains the perturbative contributions of the hypermultiplets and vector multiplets in the form of ratios of Pochhammer symbols. Such ratios are well-behave power series in $x$, hence no inverse powers of $x$ appear for good theories. The non-perturbative contributions of monopole operators are controlled by their conformal dimensions. For “good” theories, one can extract the order of the x contribution of a flux sector. Hence, one can estimate roughly as $x^{\text{(conformal dimension)}} * \text{(ratio of Pochhammers)}$. Hence, the conformal dimension of a given flux sector determines the lowest order in $x$. Thus, one can arrive at a perturbative expansion of the index by restricting the flux summation range to a finite set, and for each flux the integral is evaluated exactly.

To briefly address the comments regarding ref [52]: the mentioned $l$ is simply the magnetic flux and the fugacity $w$ is situation dependent, but generically the topological fugacities that are labelled in green as ration of $Y_j$.

We have addressed the minor points as follows: 1) Thank you for pointing this out. It is clarified now.

2) We have expanded on the conventions used in Appendix A. In light of the queries raised we have adjusted the charge of the U(1) axial symmetry to match with the more common conventions in the literature. Hence, the D3-D3 strings across an NS5 brane are now conventional untwisted hypermultiplets. (This, however, has no impact on the results and just offers clarifications.) Now, all quivers are read off by using the conventions of Table 4, while the index match requires to take inversions of the axial fugacity into account, see eqs. (A.1). In contrast, the dualisation algorithm already accounts for this by swapping twisted and untwisted multiplets.

3) The fugacity $x$ originates from [arXiv:1403.6107], where in the trace formulation of the index, the $x$ is exponentiated by $J+(H+C)/4$. With $H$ and $C$ the Cartan generators of the two SU(2) R-symmetries factors, and $J$ is the generator of the rotations on the 2-sphere.

4 - 6) Yes, thank you. We have corrected that.

7) We have added clarification to that admittedly vague statement. All we meant was that a different (non-symmetrical) starting arrangement of 5-branes gives rise to (generically) distinct theories, whose MQA/B are properly reflecting this fact.

8 -9) Thank you for spotting this! It is corrected.

10) We have clarified that sentence.

11) The appearing identify wall in Step 2 originates from the dualisation of the red bifundamental in the center of Step 1, see Fig 40(a) for the symmetric case $M=N$. In the first and last block of Step 2, the identity wall is written explicitly as two S-walls glued, because it is asymmetric (as in Fig. 40(a)). We write it as a symbol only if it is symmetric, which is precisely the middle case of Step 2.

12) The first paragraph of Section C.4 addresses the comment on the short-hand notation.

---

## Round 1 · Referee Report · Anonymous (Referee 3) · 2025-4-20

Report

This article examines the moduli space of certain classes of 3d $\mathcal{N}=3$ and $\mathcal{N}=4$ Chern-Simons-matter theories. The authors propose the use of magnetic quivers to describe the geometry of the so-called ``maximal branches" of the moduli spaces for these theories.

The results pertaining to the $\mathcal{N}=4$ theories are presented with sufficient clarity. Through the use of dualities, the authors demonstrate that the maximal branches can be identified with the Higgs and Coulomb branches of corresponding 3d $\mathcal{N}=4$ gauge theories with zero Chern-Simons levels. While explicit results are provided (e.g., Table 1 (b)), the referee does not find this aspect of the work particularly novel, as these results appear obtainable straightforwardly from well-established dualities. Nevertheless, the referee finds that the comparison between Higgsing performed via the Giveon-Kutasov duality and via the $\mathcal{T}^T$ duality, as presented for instance in Figure 4, adds scientific value to this part of the manuscript.

For the $\mathcal{N}=3$ theories, the authors limit their focus to abelian gauge theories. In the referee's opinion, the results and presentation in this section are less clear and require significant improvements.

This manuscript may be suitable for publication in SciPost, provided that the authors address the following comments satisfactorily.

Requested changes

The necessary improvements for the part concerning $\mathcal{N}=3$ Chern-Simons-matter theories include the following points:

  1. Clarity of Scope: The authors should clearly state from the beginning of the paper that their analysis for $\mathcal{N}=3$ theories is focused specifically on abelian gauge theories.

  2. Derivation Procedure: The procedure used to read off the magnetic quiver from the associated brane system is not clearly explained. For example, concerning the gauge theory presented in Eqs. (5.1)-(5.5), the authors need to provide a clear explanation of how the number of flavors in the magnetic quiver can be read off from the corresponding D3-brane segment. (As a side note, the reference to an $A_{|\kappa|-1}$ singularity below (5.1) should likely be corrected to $A_{|\kappa|+1}$ singularity).

  3. Related to the previous point, the referee finds the logical progression confusing. It is unclear whether the magnetic quiver was deduced from the known result for the maximal branch ($\mathbb{C}^2/\mathbb{Z}_{|\kappa|+2}$, as stated in Eq. (4.20) of Ref. [13], arXiv:1706.00793) or if it was derived independently from the brane system. The authors should clarify this derivation path. This clarification is needed for both subsections 5.1 and 5.2.

  4. Generalizability: If the magnetic quiver can indeed be read off directly from the brane system, it is not clear why the analysis was restricted to a single D3-brane (abelian case). The authors should comment on whether their method could be applied to $N$ D3-branes ($N>1$), which would correspond to non-abelian gauge theories and potentially lead to $U(N)$ gauge groups in the magnetic quivers (e.g., in Eqs. (5.1)-(5.10)). If the method is not applicable to the non-abelian case, a clear explanation for this limitation should be provided.

  5. Demonstration of novelty: If the authors' approach allows for the derivation of the magnetic quiver directly from the brane system, they should demonstrate its utility by providing further interesting examples that go beyond the results already contained in Ref. [13] (arXiv:1706.00793). This would help to establish that the proposed approach is genuinely new and capable of generating new results.

Recommendation

Ask for minor revision

  • validity: -
  • significance: -
  • originality: -
  • clarity: -
  • formatting: -
  • grammar: -

Author:  Marcus Sperling  on 2025-04-25  [id 5417]

(in reply to Report 3 on 2025-04-20)

We thank the referee for their thoughtful and detailed report. We appreciate the constructive feedback and the opportunity to improve the manuscript accordingly. Below, we address each point raised in the report.

General Comments on Novelty and Methodology We would like to clarify that to the best of our knowledge, there are currently no known SL(2,ℤ) duality transformations that map a 3d Lagrangian Chern-Simons-matter (CSM) theory with Chern-Simons level |κ| > 1 to a 3d Lagrangian theory without Chern-Simons terms. The approach proposed in this work - deriving magnetic quivers for each maximal branch - offers a novel and systematic procedure that does not rely on the existence of such dualities. In the special case of CS-level |κ| = 1, which is covered in Section 2, standard SL(2,ℤ) dualities apply. This section is included primarily for context and completeness and is not the focus of the paper.

We agree with the referee that the comparison of Higgsing via Giveon-Kutasov duality and $\mathcal{T}^T$-transformed theories (as shown, e.g., in Figure 4) provides additional insight.

Comments on the $\mathcal{N}=3$ Chern-Simons-matter Theories Section 1) Clarity of Scope: We agree with the referee’s suggestion and have made the scope of the analysis explicit at the beginning of the relevant section. The focus on abelian \mathcal{N}=3 theories was intended to serve as a proof of concept for the proposed method and to allow for a direct comparison with known results in the literature.

2) Derivation Procedure: We have revised the explanation of the derivation procedure in detail, in particular the method by which the number of flavors in the magnetic quiver is read off from the brane configuration. We hope the added clarifications now make the derivation more accessible to the reader. We have also corrected the typographical error concerning the ADE singularity reference below Eq. (5.1).

3) Logical Flow of Derivation: We have clarified the derivation path throughout Section 5 . Specifically, we emphasize that the magnetic quivers presented are derived independently from the brane setups. The known results from the literature, such as the geometry $\mathbb{C}^2/\mathbb{Z}_{|\kappa|+2}$ (Eq. (4.20) of Ref. [13]), were used solely for validation of our method.

4) Generalizability and Restriction to Abelian Case: We have extended the original manuscript to include several new non-Abelian $\mathcal{N}=3$ examples. These examples are now discussed in Sections 5.1.2 and 5.2.2, and demonstrate that our procedure is applicable beyond the Abelian case. We also provide an explanation of how the method generalizes to configurations involving multiple D3-branes and how this leads to non-Abelian gauge groups in the magnetic quiver.

5) Demonstration of Novelty: The newly added examples also serve to illustrate the novelty and broader applicability of our approach. These go beyond previously known results and, to the best of our knowledge, constitute new predictions. We have emphasized this contribution accordingly in the revised manuscript.

We thank the referee again for their insightful suggestions, which have helped improve the clarity and scope of the manuscript.

---

## Round 1 · Referee Report · Anonymous (Referee 2) · 2025-4-20

Strengths

Clearly organized.

Weaknesses

Extremely technical.

Report

The paper provides a systematic framework for analyzing the maximal branches of the moduli space of 3d supersymmetric Chern-Simons Matter (CSM) theories with N=4 or N=3 supersymmetry. While several previous results on the subject exist, the paper analyzes the problem using recent techinques, like magnetic quivers and dualization algorithms, and provides general results for the vacua and RG structure of 3d supersymmetric CSM theories in regimes previously hard to tackle, like CS levels greater than one or N=3 supersymmetry. The analysis is demonstrated with many examples and tables. This is an extremely technical paper addressed to a relatively small community of people working on similar topics, but it is scientifically sound and it contains useful information. I think that it meets the SciPost criteria for acceptance and I recommend it for publication.

Requested changes

It would be useful to have in the introduction a more expanded discussion about the motivations of this paper in particular about what we can learn from the maximal branch structure of the moduli space.

The paragraph starting with "In N =4 CSM theories, the presence of Chern-Simons couplings modifies this picture." in the introduction is a bit confusing. Most of the things said below are also true for N=4 theories without CS. Please reformulate it.

Recommendation

Ask for minor revision

  • validity: high
  • significance: good
  • originality: good
  • clarity: high
  • formatting: excellent
  • grammar: excellent

Author:  Marcus Sperling  on 2025-04-25  [id 5416]

(in reply to Report 2 on 2025-04-20)

We thank the referee for the comments and suggestions.

Regarding the requested changes, we have modified the manuscript as follows: 1) We have expanded on the motivation for this work in the introduction. 2) We have clarified the paragraph pointed out in the report.

---

## Round 2 · Referee Report · Anonymous (Referee 2) · 2025-4-25

Report

The authors addressed the comments. The paper can be published.

Recommendation

Publish (meets expectations and criteria for this Journal)

---

## Round 2 · Referee Report · Anonymous (Referee 3) · 2025-4-26

Report

The authors have addressed the referee's comments satisfactorily. The article deserves publication in the SciPost.

Recommendation

Publish (meets expectations and criteria for this Journal)

---

## Round 2 · Referee Report · Anonymous (Referee 1) · 2025-4-27

Report

The authors have satisfactorily addressed the questions and corrected the points raised in my previous review. Furthermore, following other reviewers's insight, they have clarified parts of the presentation and extended their analysis of $\mathcal{N}=3$ cases with non-Abelian examples, which appear to be both correct and novel.
As a side comment, the newly added find Table 4 very neat. It is certainly true that the previous conventions did not alter the final results, but I believe the current version using the most common convention for hypers vs twisted hypers will be of easier access for the readers.

Overall I consider the paper worth publishing in SciPost Physics in the current version.

Recommendation

Publish (easily meets expectations and criteria for this Journal; among top 50%)

---

## Round 2 · List of Changes

• clarified axial charge convention; added Table 4 to specify multiplets read off from brane intersections
  • improved presentation of Abelian N=3 Chern-Simons matter theories sections
  • added new sections on non-Abelian N=3 Chern-Simons matter theories
  • added Figure 23 to explain the strategy for deriving magnetic quivers for each maximal branch in N=3 theories
  • revised and expanded introduction, to emphasise motivation
  • fixed typos

---

## Editorial Decision

published